# Practical Mechanism for Fault-Tolerant Spiking Neural Networks via Simple Input Control Based on Learnable Fragmentation

**Hyun-Jong Lee** [1]   **Jae-Han Lim** [1]

## Abstract

Spiking Neural Networks (SNNs) are regarded as the third generation of neural networks, offering energy-efficient computing for neuromorphic devices. Despite this benefit, hardware-implemented SNNs are vulnerable to hardware faults, which severely degrade their performance. Previous approaches have required direct access to internal SNN circuits to modify weights or monitor internal states, limiting their practicality. Improving robustness to hardware faults without such access remains challenging. To overcome this challenge, we propose a fault-tolerant mechanism that operates only through input data control. Hardware faults reduce the usable learning capacity of SNNs, resulting in a mismatch between the instantaneous input load and the degraded network dynamics. Our mechanism mitigates this mismatch by dividing each input sample into multiple fragments, redistributing the input load via a learnable fragmentation strategy. The strategy learns two key fragmentation components: 1) division boundaries and 2) the number of fragments. To our knowledge, this is the first mechanism to improve the fault tolerance of SNNs without accessing the internal circuits. Experimental results demonstrate that our mechanism consistently outperforms previous methods in various SNN models, achieving these gains without direct access to internal circuits. Furthermore, we validate its effectiveness on SNNs implemented with a physical FPGA platform, confirming its practicality.

## 1. Introduction

Researchers have focused on Spiking Neural Networks (SNNs) to develop neuromorphic devices that target energy-constrained environments (Garaffa et al., 2021; Jeong et al., 2025; Venkatesha et al., 2021). As third-generation neural networks, SNNs process information using discrete spike events and temporal dynamics, enabling low-power operation and high biological plausibility compared to conventional artificial neural networks (Schuman et al., 2022; Pfeiffer & Pfeil, 2018). These properties make SNNs particularly attractive for hardware-implemented neuromorphic devices under limited power and resource budgets.

Although SNNs are essential for neuromorphic devices, hardware-implemented SNNs remain vulnerable to permanent hardware faults, which frequently occur in neuromorphic devices' electrical components and significantly impair SNNs' learning performance (Spyrou et al., 2021; Lee & Lim, 2023). This vulnerability stems from two factors: 1) the intrinsic instability of hardware components and 2) the sensitivity of SNN training dynamics to persistent parameter perturbations (Garaffa et al., 2021).

Previous approaches to enhance the tolerance of hardware-implemented SNNs against hardware faults require hardware reconfigurability to manage electronic components directly in neuromorphic devices or rely on complex algorithms to regulate abnormal neuronal activities (Vu et al., 2019; Putra et al., 2022). Although these approaches have improved fault tolerance, they exhibit the following problems that reduce their practicality.

***1. Requirement for direct access to internal SNN circuits:*** Existing approaches require direct access to internal circuits to modify synaptic weights or circumvent faulty synapses (Putra et al., 2022; Chen & Chakrabarty, 2021). However, these approaches rely on hardware reconfigurability, which incurs additional design complexity and implementation cost (Garaffa et al., 2021; Takano & Amano, 2022; Putra et al., 2023). In particular, the reconfigurable design is incompatible with off-the-shelf neuromorphic devices, where access to internal circuits is significantly limited.

***2. Algorithmic complexity in conventional mechanisms:*** Previous approaches rely on complex algorithms that are impractical for hardware implementation (Vu et al., 2019; Yang et al., 2022; Han et al., 2023). The algorithms often malfunction within neuromorphic devices due to their intrinsic instability (Liu et al., 2017; Rasch et al., 2023).

---

[1]Department of Software, Kwangwoon University, Seoul, South Korea. Correspondence to: Jae-Han Lim <ljhar@kw.ac.kr>.

*Proceedings of the 43rd International Conference on Machine Learning*, Seoul, South Korea. PMLR 306, 2026. Copyright 2026 by the author(s).

These limitations motivate the need for pragmatic fault-tolerant mechanisms that operate without accessing or modifying internal SNN circuits. To this end, we identify a fundamental performance-limiting phenomenon in faulty SNNs, called **the bottleneck problem**, where learning degrades when the effective information load exceeds the network's usable learning capacity. The bottleneck arises from two coupled effects. First, permanent synaptic faults cause affected weights to become fixed during training, reducing the number of learnable synapses. Second, the resulting fixed synaptic contributions bias membrane potential dynamics, shifting neuronal states toward the regions where gradients vanish, further reducing learning capacity.

The bottleneck problem is caused by a mismatch between input information load and usable learning capability, rather than by a lack of internal circuit reconfigurability. This implies that controlling input presentation becomes a direct means of mitigation without accessing or modifying internal circuits. Motivated by this insight and **flow control in computer networks** (Kurose & Ross, 2012), we propose a fault-tolerant SNN mechanism based solely on **input data control** via **learnable data fragmentation**. Our mechanism mitigates the bottleneck by temporally redistributing the information of a single input sample into smaller fragments. The fragmentation strategy is learned using simple training procedures that explicitly account for reduced usable capacity, improving learning performance without accessing or modifying internal SNN circuits.

Key differences from previous approaches are as follows. First, **our mechanism does not require hardware reconfigurability to modify synapses or access internal circuit states**, which would otherwise increase implementation cost and overhead. Second, our mechanism does not need complex algorithms to control neuronal activity. These novelties arise from the two core features of our approach.

***1. Division of a data sample into small fragments:*** Our mechanism divides a single input sample into multiple fragments, reducing the information load entering the network. This allows faulty SNNs to learn effectively despite reduced usable learning capacity.

***2. Fragmentation method based on training:*** To make fragmented inputs remain aligned with the network's learning dynamics, we develop a training-based fragmentation strategy that adapts the number of fragments and division lines to reduce the input load imposed on the reduced capacity.

With our mechanism, various SNN models achieve significantly higher classification accuracy than models using previous approaches under fault-injected conditions, while consuming less energy due to our mechanism's simplicity. We also conduct experiments with real hardware SNNs built in a Field-Programmable Gate Array (FPGA). Our work has the following contributions.

- We propose a practical mechanism to enhance the fault tolerance of hardware-implemented SNNs using only input-data control, without direct synapse modification or access to internal circuits, enabling deployment on non-reconfigurable neuromorphic hardware.

- We present a concrete theoretical basis for our mechanism by mathematically and experimentally investigating how synaptic faults degrade the usable learning capacity of SNN models in our motivation study to develop a theoretically sound fragmentation mechanism.

- We introduce a training-based data fragmentation method that adapts the number of fragments and division lines to reduce usable learning capacity, without relying on internal circuit monitoring.

- We provide a rich set of evaluation results in various scenarios, including hardware environments. The evaluation results demonstrate that ours, based on simple data fragmentation, improves the fault tolerance of SNNs more significantly than previous approaches.

## 2. Background

### 2.1. Spiking neural networks

In SNNs, spiking neurons fire and emit output spikes at each time step, the unit of time in spike occurrence. At each time step, a neuron updates its membrane potential based on incoming synaptic input and its past state, and it may emit an output spike depending on the potential. Various neuron models have been proposed for SNNs. Among them, the Leaky Integrate-and-Fire (LIF) model is widely used for its simplicity and effectiveness in capturing essential spiking dynamics (Moitra et al., 2023). In this paper, we adopt a discrete-time LIF formulation, which is commonly used in both software simulation and neuromorphic hardware implementations. The LIF neuron dynamics are given by Equation (1).

$$u_t = \alpha\, u_{t-1} + i_t - \beta s_{t-1}, \quad \alpha = e^{-\Delta t/\tau_m}$$
$$s_t = \text{Heaviside}(u_t - \vartheta) \tag{1}$$

where $u_t$ is the membrane potential at time $t$[1], and $i_t$ is the input current aggregated from pre-synaptic spikes. The decay factor $\alpha$ models the leakage of the membrane potential with time constant $\tau_m$ and time-step duration $\Delta t$. The term $\beta s_{t-1}$ represents the membrane reset effect induced by a spike at the previous time step, with reset strength $\beta$. The binary output spike $s_t \in \{0, 1\}$ is generated by the Heaviside step function $heavidside(u)$, which is one if $u > 0$ and zero otherwise. $\vartheta$ denotes the threshold.

---

[1]To simplify the notation, the layer and neuron indices are omitted from $u_t$.

Synaptic weights determine how spikes from pre-synaptic neurons influence the membrane potential of post-synaptic neurons (Venkatesha et al., 2021). SNNs can be trained through a variety of paradigms, including supervised and unsupervised rules. In practice, supervised rules commonly rely on surrogate-gradient-based optimization, enabling gradient-based training despite the non-differentiability of spikes, while unsupervised rules such as spike-timing-dependent plasticity (STDP) update weights based on the relative timing of pre- and post-synaptic activity.

SNNs target energy-efficient information processing in neuromorphic systems through event-driven dynamics and temporal sparsity. SNNs consume less energy than conventional neural networks for the following reasons. First, computation and communication are event-driven, as spike transmission and input integration occur only when spikes are present, inducing sparse activity (Lee & Lim, 2024). Second, neuromorphic hardware implementations replace dense multiply–accumulate (MAC) operations with simple accumulation (AC) operations, replacing multiplications between weights and binary spike inputs with accumulation operations triggered by spike events.

### 2.2. Synaptic faults

Synaptic faults refer to persistent or transient impairments in synaptic weights or connectivity caused by device variability, noise, or hardware degradation. They alter the effective synaptic influence on post-synaptic neurons and disrupt SNN learning dynamics by introducing bias or stochastic perturbations. A representative class of permanent faults is Stuck-At Faults (SAFs), in which synaptic weights become fixed at extreme values within their allowable range and no longer adapt during training (Vatajelu et al., 2019). SAFs are categorized as SA1 or SA0, corresponding to saturation at the upper or lower weight bound, respectively; these labels do not imply numerical values of 1 or 0. In contrast, Random Weight Faults (RWFs) are transient perturbations where synaptic weights fluctuate stochastically around their nominal values due to noise sources such as thermal effects (Vatajelu et al., 2019). RWFs introduce time-varying uncertainty in synaptic efficacy, effectively weakening or amplifying connectivity. Beyond weight corruption, Connectivity Error Faults (CEFs) affect the network topology rather than individual weight values, permanently altering intended connectivity patterns within SNNs.

## 3. State of the art

### 3.1. Analysis of faults in neuromorphic devices

Researchers have investigated the impact of hardware faults on neuromorphic devices. They inject faults into synapses and neurons in neuromorphic devices, and analyze the im-

pact of faults on the classification performance of SNNs (Vatajelu et al., 2019). The authors in (Lee & Lim, 2023) build a memristive neuromorphic simulator and analyze how faults affect classification accuracy. In this study, they observed that faults occurring in synapses associated with important input features tend to induce more severe performance degradation. In addition, various fault models have been explored to examine the detailed response of spiking neurons under the fault models (Ali El Sayed, 2021; Garaffa et al., 2021). However, previous work overlooks the excessive internal updates induced by hardware faults.

### 3.2. Mechanisms to improve fault tolerance of hardware-implemented SNNs

Conventional approaches to improving the fault tolerance of hardware-implemented SNNs in neuromorphic devices typically rely on modifying faulty components or implementing additional fault-mitigation architectures. Prior work has exploited the error-correction capability of binary codes in output decoding to enhance the robustness of neural networks (Liu et al., 2019; Yu et al., 2023). Other methods induce auxiliary spikes or reroute spike propagation to mitigate the adverse effects of hardware faults in neuromorphic SNNs (Vu et al., 2019; Yang et al., 2022). Fault mapping techniques have also been proposed to identify neurons that are severely affected by hardware faults (Putra et al., 2022; Wicaksana Putra et al., 2021; Yang et al., 2022). In addition, some approaches explicitly mask faulty elements by zeroing affected pre-trained weights and subsequently retraining the network with per-layer thresholds (Siddique & Hoque, 2023). Astrocyte-based self-recovery strategies have been introduced to strengthen fault tolerance in neuromorphic systems (Han et al., 2023; Varshika et al., 2023). Building upon these ideas, subsequent work augments SNNs with astrocyte-inspired leaky integrators to stabilize spiking dynamics and further improve robustness under hardware non-idealities (Yunusoglu et al., 2025). Lightweight approaches, such as suppressing abnormal pre-activation, removing fault-affected neurons, and tuning thresholds, have enhanced the fault tolerance (Saha et al., 2023; Spyrou et al., 2021; Saha et al., 2024). Despite their effectiveness, these methods require complex internal architectures and assume direct access to internal network states or parameters, while neglecting the limited reconfigurability of hardware components in practical neuromorphic devices.

## 4. Motivation study

To motivate our work, we investigate how synaptic faults lead to bottlenecks and degrade the learning ability of SNNs. For this purpose, we first show that membrane potentials often deviate far from the spiking threshold of neurons in faulty SNNs. We then demonstrate that this deviation causes

the surrogate gradient to collapse toward zero, thereby inducing the vanishing gradient problem. Finally, we show that input fragmentation reduces the likelihood of the vanishing-gradient problem, thereby mitigating the bottleneck and preserving SNNs' learning capacity.

### 4.1. Distribution of membrane potentials in faulty SNNs

To demonstrate that membrane potentials are likely to escape the surrogate gradient corridor in faulty SNNs, we measure the absolute deviation of the membrane potential from the spiking threshold (i.e., $|u_t - \vartheta|$), which directly determines corridor-escape. Here, the surrogate gradient corridor refers to the region where the surrogate derivative is non-negligible, and we denote its half-width by $\xi$; outside this corridor, $g(u_t - \vartheta) \approx 0$, where $g(\cdot)$ is the surrogate derivative function. In this experiment, we evaluate three SNN models: a Multi-Layered Perceptron (MLP) with 4 layers, VGG-7, and ResNet-18[2], on the MNIST, CIFAR-10, and CIFAR-100 datasets for the MLP, VGG-7, and ResNet-18. We inject SAFs ($SA0 : SA1 = 1.75 : 9.04$) (Chen et al., 2017) into 50% of synapses in the three SNN models during training. We train these SNN models using the Adam optimizer and adopt Root Mean Square Error (RMSE) as a loss function (Fang et al., 2023). To increase the statistical reliability, we repeat the experiments ten times and present the results with 95% confidence intervals.

*Table 1.* Average of the absolute deviation of membrane potential from threshold($\mathbb{E}[|u_t - \vartheta|]$) under SAF injection.

|          | MLP          | VGG-7        | RESNET-18    |
|----------|--------------|--------------|--------------|
| NOMINAL  | $0.25 \pm 0.05$ | $0.81 \pm 0.23$ | $0.39 \pm 0.11$ |
| SAF      | $80.18 \pm 6.48$ | $30.94 \pm 5.27$ | $2.41 \pm 0.38$ |

Table 1 demonstrates the average absolute deviation over all spiking neurons in two settings: 1) SNNs without SAF injection ("NOMINAL") and 2) SNNs with SAF injection ("SAF"). The experimental results show that the deviation in the SAF setting is much larger than that in the NOMINAL setting, exceeding the half-width of the surrogate gradient corridor in a large fraction of neurons, thus indicating that membrane potentials are more likely to lie outside the surrogate gradient corridor in the presence of synaptic faults.

We further visualize the distribution of the absolute deviation of potentials from the threshold (i.e., $|u_t - \vartheta|$) for the MLP cases in Figure 1. As shown in Figure 1, most membrane potentials are far from the threshold in the SAF setting, whereas a large fraction lie close to the threshold in the NOMINAL setting. This distributional shift directly increases the probability that the surrogate gradient evaluates to zero, explaining why synaptic faults induce severe gradient vanishing during training.

---

[2]We employ LIF neurons for MLP and VGG, and IF neurons for ResNet (Hu et al., 2023)

### 4.2. Gradient collapse from abnormal distribution of the membrane potential ($|u_t - \vartheta|$)

We next show that the gradients of non-faulty synapses can vanish under SAF-induced membrane-potential shifts

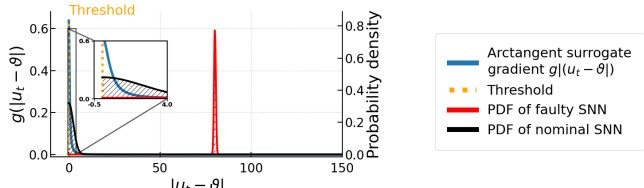

*Figure 1.* The surrogate gradient $g(u_t - \vartheta)$ (arctangent) and the probability distribution of $|u_t - \vartheta|$ in two settings: 1) SAF is not injected and 2) SAF is injected.

Figure 1 depicts the surrogate gradient function (blue) and the distribution of the absolute deviation from the threshold, $|u_t - \vartheta|$, in MLPs under two settings: SNN with SAFs (red) and SNN without SAFs (black). In SNNs with SAFs, a large fraction of membrane potentials escape the surrogate gradient corridor, leading to strong suppression of surrogate gradients in most cases. Considering the gradient equation of STBP, the gradient of a synaptic weight connected to such corridor-escaped neurons, $\nabla_w \mathcal{L}$, is strongly suppressed.

$$\nabla_w \mathcal{L} = \sum_{t=1}^{T} \nabla_{u_t} \mathcal{L} \cdot \frac{\partial u_t}{\partial w}$$
$$= \sum_{t=1}^{T} \nabla_{s_t} \mathcal{L} \cdot g(u_t - \vartheta) \frac{\partial u_t}{\partial w} \tag{2}$$

where $\mathcal{L}$ is the loss function and $s_t$ denotes the spike output connected to the synapse at time t. As surrogate gradients are more likely to fall below effective levels in faulty SNNs, $\nabla_w \mathcal{L}$ is more likely to approach zero in aggregate, thereby exacerbating the bottleneck problem.

To justify this claim, we measure the probability that the gradient is negligible, $Pr(|\nabla_w \mathcal{L}| < \sigma)$ with $\sigma = 0.0001$, across various SNN models and datasets (Pennington et al., 2017; Zhang et al., 2018). As shown in Table 2, these probabilities are significantly higher in SAF settings than in NOMINAL settings. This indicates that synaptic faults substantially increase the likelihood of gradient vanishing, preventing SNN models from learning input data samples.

*Table 2.* Probability that the absolute value of the gradient is below 0.0001 across all neurons $Pr(|\nabla_w \mathcal{L}| < \sigma)$, $\sigma = 0.0001$ under SAF with 50% fault ratio.

|          | MLP             | VGG-7           | RESNET-18       |
|----------|-----------------|-----------------|-----------------|
| NOMINAL  | $0.0092 \pm 0.0012$ | $0.019 \pm 0.0026$ | $0.011 \pm 0.0035$ |
| SAF      | $0.99 \pm 0.008$ | $0.87 \pm 0.033$ | $0.96 \pm 0.027$ |

### 4.3. Input fragmentation to mitigate the bottleneck problem and gradient vanishing

To demonstrate the effect of input fragmentation in mitigating the bottleneck problem in faulty SNNs, we adopt an

input fragmentation method in which each input sample is divided into four fragments using horizontal division lines at regular intervals. To show the benefit of fragmentation, we measure two performance metrics: 1) $E[|u_t - \vartheta|]$ and 2) $Pr(|\nabla_w \mathcal{L}| < \sigma)$ with $\sigma = 0.0001$.

*Table 3.* $\mathbb{E}[|u_t - \vartheta|]$, probability that a gradient is below 0.0001 ($Pr(|\nabla_w \mathcal{L}|) < \sigma$, $\sigma = 0.0001$), and classification accuracy of MLP, VGG-7, and ResNet-18 SNN models under SAF injection with 50% fault ratio.

| | MLP (MNIST) | VGG-7 (CIFAR-10) | ResNet-18 (CIFAR-100) |
|---|---|---|---|
| | | $\mathbb{E}[|u_t - \vartheta|]$ | |
| SAF | $80.18 \pm 6.48$ | $30.94 \pm 5.27$ | $2.41 \pm 0.38$ |
| SAF (FRAGS) | $18.64 \pm 3.41$ | $7.38 \pm 2.39$ | $1.37 \pm 0.83$ |
| | | $Pr(|\nabla_w \mathcal{L}|) < \sigma, \sigma = 0.0001$ | |
| SAF | $0.99 \pm 0.008$ | $0.87 \pm 0.033$ | $0.96 \pm 0.027$ |
| SAF (FRAGS) | $0.29 \pm 0.031$ | $0.35 \pm 0.037$ | $0.38 \pm 0.042$ |
| | | CLASSIFICATION ACCURACY | |
| SAF | $10.82 \pm 0\%$ | $21.43 \pm 5.69\%$ | $1.12 \pm 0.12\%$ |
| SAF (FRAGS) | $51.05 \pm 10.62\%$ | $37.29 \pm 6.81\%$ | $8.28 \pm 3.25\%$ |

Table 3 shows that $E[|u_t - \vartheta|]$ is significantly reduced when using input fragmentation. This is because fragmentation reduces the per-step effective energy of input samples, inducing a decrease in the instantaneous input current. With a smaller input current, the membrane potential does not increase excessively, increasing the likelihood that the potential remains within the surrogate gradient corridor, as reflected by the reduced deviation. This, in turn, mitigates the gradient vanishing problem, as shown by the reduced $Pr(|\nabla_W \mathcal{L}| < \sigma)$ under fragmentation. This table also presents the classification accuracy of SNN models with and without fragmentation under SAFs. Models using fragmentation classify data samples more accurately than those without fragmentation across three SNN models. This improvement arises from the reduced likelihood of gradient vanishing, enabling the models with fragmentation to learn data samples effectively.

## 5. Proposed Mechanism

### 5.1. Overview

Our fragmentation mechanism converts each input sample into a temporal fragment sequence, injecting a single fragment per time step. By distributing the per-step effective input energy across multiple fragments, the mechanism reduces the per-step information load, thereby preventing abrupt potential drift that would otherwise induce corridor escape. The goal of our mechanism is to learn an optimal fragmentation strategy in which the per-step information load does not exceed the usable learning capacity of a faulty SNN, while minimizing performance degradation from excessive fragmentation. To this end, we train fragmentation parameters only at the input level, jointly with SNN training, without modifying or accessing internal circuits. We achieve our goal through three key features.

**1. Training division lines:** The input sample is partitioned

into $T$ fragments using $T-1$ learnable lines, each parameterized by three variables defining the fragmentation boundary.

**2. Selecting the number of fragments:** To find the best number of fragments, our mechanism learns the number by selecting among the candidate set using a differentiable Gumbel-Softmax selector, enabling joint optimization with the division line parameters and SNN weights, while yielding a fixed fragment count at inference.

**3. Regularization during learning:** We include regularization terms that encourage balanced per-fragment energy. These constraints prevent degenerate fragment configurations, such as allocating the most informative regions of the data samples to a few fragments.

### 5.2. Learnable fragmentation with division lines

Three parameters are used to identify each division line, and these parameters are shared across all input samples and updated via backpropagation during SNN training. Specifically, $T - 1$ lines are defined by $\ell_k(x, y) = 0$, $k = 1, 2, \cdots, T - 1$, where $\ell_k(x, y) = a_k x + b_k y + c_k$. Let $\mathbf{I} \in \mathbb{R}^{H \times W}$ denote an input sample, and $(x, y)$ refers to the coordinate of a pixel center[3]. $a_k$ and $b_k$ specify which direction the division line faces, whereas $c_k$ specifies where the line is placed in the input plane. For stable training, we adopt a bounded reparameterization (Duda & Hart, 1972):

$$(a_k, b_k) = \frac{(h_k, v_k)}{\sqrt{h_k^2 + v_k^2 + \epsilon}}, \quad c_k = d \tanh(r_k) \quad (3)$$

where $(h_k, v_k, r_k)$ are learnable parameters for the $k^{th}$ division line. The line offset $c_k$ is bounded within a range determined by $d$. $\epsilon > 0$ is a constant for numerical stability.

We compute a partition of each input sample using masks, which are expressed by Equation (4) and Equation (5):

$$m_t(x, y) = soft_t(x, y) \prod_{i < t} \left(1 - soft_i(x, y)\right), \quad t = 1, \cdots, T$$
$$(4)$$

$$soft_t(x, y) = \begin{cases} \text{sigmoid}(-\ell_t(x, y)), & t < T \\ 1, & t = T \end{cases} \quad (5)$$

where $m_t(x, y)$ denotes the mask for a pixel $(x, y)$ to generate $t^{th}$ fragment. The fragment is made by $\mathbf{I}_t = \mathbf{M_t} \odot \mathbf{I}$, where $\mathbf{M_t}$ is the mask matrix and $\odot$ is the Hadamard product. During training, binary masks[4] are used in the forward pass, while the soft masks are exploited in the backward pass using a straight-through estimator, allowing gradients to flow through discrete fragment assignments.

---

[3] When using 1D sequential data, we convert the sequence into a 2D structure, following the procedure in Appendix B.1.1.

[4] To generate the binary mask, we replace $soft_t(x, y)$ with $Heaviside(soft_t(x, y) - 0.5)$.

## 5.3. Selecting the number of fragments

In our mechanism, we select the best number of fragments from a candidate set $\mathcal{T} = \{T_1, \ldots, T_N\}$. A simple approach is to train separate SNN models configured with different $T$ values and select the one maximizing accuracy. However, this approach significantly increases training cost from repeated forward and backward passes, thereby increasing training time and energy.

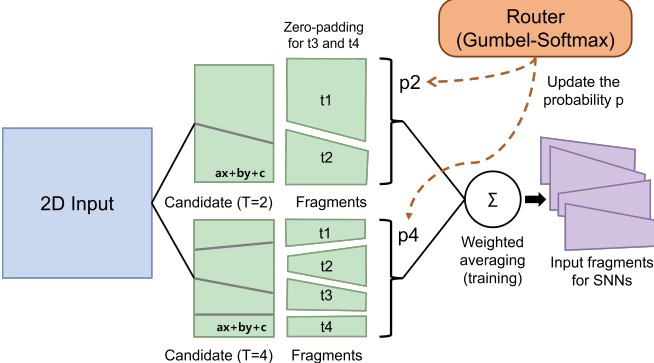

*Figure 2.* Procedure for creating the mixed fragments to train the number of fragments using a single SNN model. Each fragment is created by a binary mask in Equation (5)

For efficient training, we design a new method that trains the number of fragments jointly with division lines and SNN weights using a single SNN model. The key idea ensuring both efficiency and accuracy is to generate mixed fragments, obtained by forming a combination of fragments corresponding to different $T$ using weight $p^{(T)}, T \in \mathcal{T}$, and to train a single SNN using the mixed fragments, as depicted in Figure 2. This mixing allows the model to explore multiple fragmentation granularities during training without committing to a single discrete choice. To this end, we learn the weights $p^{(T)}$ by training logits $\nu^{(T)}$ via a differentiable Gumbel-Softmax relaxation, described as follows.

$$p^{(T)} = \frac{\exp\left(\nu^{(T)} + n^{(T)}\right)}{\sum_{T' \in \mathcal{T}} \exp\left(\nu^{(T')} + n^{(T')}\right)}, \quad n^{(T)} \sim \text{Gumbel}(0, 1) \tag{6}$$

where $p^{(T)}$ serves as a soft selection weight for the candidate $T$. Through this approach, we derive a simplex vector $\mathbf{p} = (p^{(T)})_{T \in \mathcal{T}}$, ensuring that the gradients propagate to $\nu^{(T)}$. The Gumbel-Softmax relaxation provides a differentiable approximation to categorical sampling with Gumbel noise.

A practical challenge is that different $T$ implies different temporal lengths. To avoid running SNNs separately for each candidate $T$, we convert every candidate sequence to a fixed length $T_{\max} = \max(\mathcal{T})$ using a simple temporal alignment function $\mathcal{A}_T$, (e.g., zero-padding $T_{\max} - T$ fragments). We then mix candidates at the fragment level after temporal alignment, $\widetilde{\mathbf{F}}^{(T_{max})}$.

$$\widetilde{\mathbf{F}}^{(T_{\max})} = \sum_{T \in \mathcal{T}} p^{(T)} \mathcal{A}_T\left(\mathbf{F}^{(T)}\right) \tag{7}$$

where $\mathbf{F}^{(T)} = (\mathbf{I}_1^{(T)}, \cdots, \mathbf{I}_T^{(T)})$ is a temporal input sequence when using $T$ as the number of fragments. Both the selector logits $\nu^{(T)}$ and the corresponding learnable division line parameters $\{h_k, v_k, r_k\}, k \in \{1, \ldots, T-1\}$ associated with each $T \in \mathcal{T}$ are updated, while all operations remain confined to input construction. After training SNNs and $p^{(T)}$, we select $T_{select}$ as follows: $T_{select} = \underset{T \in \mathcal{T}}{\arg\max}\, p^{(T)}$. Then, at inference, the selected $T$ is used to generate fragments to avoid additional overhead.

## 5.4. Regularization and entropy-based decoding

We use RMSE as the main loss for SNNs, $\mathcal{L}_{main}$[5]. Without constraints, the fragmentation can collapse to degenerate solutions, such as allocating most informative regions to a few steps. To avoid this issue, we encourage each fragment to have comparable data energy. For each $T \in \mathcal{T}$, we define the data energy of fragment at time step $t$, $\varsigma_t^{(T)} = \sum_{x,y} |I(x,y)| \cdot m_t^{(T)}(x,y)$ where $I(x,y)$ is the input value at $(x,y)$. We define the normalized distribution $q_t^{(T)} = \varsigma_t^{(T)} / \sum_j \varsigma_j^{(T)}$ and penalize the deviation from uniformity

$$\mathcal{L}_{\text{bal}}^{(T)} = \frac{1}{T} \sum_{t=1}^{T} \left(q_t^{(T)} - \frac{1}{T}\right)^2 \tag{8}$$

For training across $T$, we use the selector-weighted regularizer $\mathcal{L}_{\text{bal}} = \sum_{T \in \mathcal{T}} p^{(T)} \mathcal{L}_{\text{bal}}^{(T)}$ so that regularization strength aligns with the relevance of each candidate. The balance loss equalizes per-fragment energy so that no single fragment's input current pushes the neuron's membrane potential out of the surrogate-gradient corridor (i.e., the effective nonzero-gradient range near threshold).

The total training loss is given by Equation (9)

$$\mathcal{L} = \mathcal{L}_{\text{main}} + \lambda_{\text{bal}} \mathcal{L}_{\text{bal}} \tag{9}$$

where $\lambda_{\text{bal}}$ balances regularization loss against $\mathcal{L}_{\text{main}}$[6].

We adopt an entropy-based output decoding technique to aggregate SNN outputs across all time steps, which is suited to fragmented inputs yielding outputs of different confidence, using Equation (10) (Qiu et al., 2025)

$$\hat{\mathbf{o}} = \sum_{t=1}^{T} e_t \mathbf{o_t}, \qquad e_t = \frac{\exp\left(-\gamma S(\text{softmax}(\mathbf{o_t}))\right)}{\sum_{s=1}^{T} \exp\left(-\gamma S(\text{softmax}(\mathbf{o_s}))\right)} \tag{10}$$

where $\hat{\mathbf{o}}$ is the final output vector after entropy-weighted aggregation, and $\mathbf{o_t}$ is an output vector of the SNN at time $t$. $S(\text{softmax}(\mathbf{o_t}))$ denotes the Shannon entropy of the softmax of the output, which serves as a proxy for estimation

---

[5]RMSE can be replaced with other terms (e.g., cross-entropy).
[6]In our experiment, we set $\lambda_{\text{bal}}$ to 0.01.

confidence, so that time steps producing more confident (lower entropy) outputs are emphasized. $\gamma$ controls the strength of this emphasis, and $e_t$ is the resulting weight derived from output entropy. We use Shannon entropy because lower entropy indicates a more peaked predicted class distribution, which means a more decisive and confident prediction, whereas higher entropy suggests ambiguous evidence (Qiu et al., 2025; Gal & Ghahramani, 2016).

### 5.5. Putting all components together

Here, we explain the overall training procedures of our mechanism using pseudo-code.

---

**Algorithm 1** Training procedure of our mechanism

---

**Require:** mini-batch $(\mathbf{I}, target)$, candidate set $\mathcal{T}$, SNN model $net$, line parameters $(\mathbf{H}^{(\mathbf{T})}, \mathbf{V}^{(\mathbf{T})}, \mathbf{R}^{(\mathbf{T})})$, and $\nu^{(T)}$ for training the number of fragments for $T \in \mathcal{T}$

1: **for** $i = 1$ **to** $n_{epoch}$ **do**
2:     **for each** $T \in \mathcal{T}$ **do**
3:         $\{M_t^{(T)}\}_{t=1}^T \leftarrow \text{BuildMasks}(H^{(T)}, V^{(T)}, R^{(T)})$
4:         $\mathbf{I_t^{(T)}} \leftarrow \mathbf{I} \odot \mathbf{M_t^{(T)}} \quad \forall t \in \{1, \ldots, T\}$
5:         $p^{(T)} \leftarrow \text{GumbelSoftmax}(\nu^{(T)})$
6:     **end for**
7:     $\hat{I}_t^{(T)} \leftarrow \mathbf{1}_{\{t \leq T\}} I_{\min(t,T)}^{(T)}, \quad \forall t \in \{1, \cdots, T_{\max}\}$
        $\{\hat{\mathbf{I}}_{\mathbf{t}}^{(\mathbf{T})}$ is the result of the alignment function.$\}$
8:     $\widetilde{\mathbf{I}}_{\mathbf{t}} \leftarrow \sum_{T \in \mathcal{T}} p^{(T)} \hat{I}_t^{(T)} \quad \forall t \in \{1, \ldots, T_{\max}\}$
9:     $\mathbf{o_t} \leftarrow net\left(\widetilde{\mathbf{I}}_{\mathbf{t}}\right) \quad \forall t \in \{1, \ldots, T_{\max}\}$
10:    $\hat{\mathbf{o}} \leftarrow \text{Decode}(\mathbf{o}_1, \cdots, \mathbf{o}_{T_{\max}})$
11:    $\mathcal{L} \leftarrow \mathcal{L}_{\text{main}} + \lambda_{\text{bal}} \mathcal{L}_{\text{bal}}$
12:    Update $(\mathbf{W}, \mathbf{H}^{(\mathbf{T})}, \mathbf{V}^{(\mathbf{T})}, \mathbf{R}^{(\mathbf{T})}, \nu^{(T)})$ for $T \in \mathcal{T}$
13: **end for**

---

Algorithm 1 describes the overall training procedure of our mechanism. In lines 2-6, our mechanism converts input samples into $T$ fragments using masks based on the division lines $(\mathbf{H}^{(\mathbf{T})}, \mathbf{V}^{(\mathbf{T})}, \mathbf{R}^{(\mathbf{T})})$. In line 5, the selection probability $p_T$ is calculated using the Gumbel-Softmax method. In lines 7 and 8, we generate a mixed fragment sequence by temporal alignment and taking a weighted average of fragments over $T$. In line 9, the SNN produces per-step outputs $\mathbf{o_t}$, and decodes $\hat{\mathbf{o}}$ through entropy-based decoding. In line 10, we add the balance loss that promotes balanced fragment usage and minimizes the sum of the main and balance losses. In line 11, the optimizer updates synaptic weights $\mathbf{W}$, $(\mathbf{H}^{(\mathbf{T})}, \mathbf{V}^{(\mathbf{T})}, \mathbf{R}^{(\mathbf{T})})$, and $\nu^{(T)}$ through backpropagation during training.

## 6. Experiments

### 6.1. Experimental settings

To demonstrate the effectiveness of our mechanism across diverse settings, we conduct experiments with vanilla MLP (LIF neurons), VGG-7/11/15 (LIF neurons), and ResNet-

18/34 (IF neurons) (Hu et al., 2023). We implement SNN models with SpikingJelly, a widely used SNN framework (Fang et al., 2023). We use various datasets: MNIST, FMNIST, UCI-HAR, and AudioMNIST for MLP; CIFAR-10/100 for VGG-7/11/15 and ResNet-18/34; and Tiny-ImageNet for ResNet-34 (LeCun et al., 1998; Xiao et al., 2017; Krizhevsky, 2009; Reyes-Ortiz et al., 2013; Becker et al., 2024; Deng et al., 2015). We inject SAFs during training and measure classification accuracy at inference. The SA0:SA1 ratio is set to $1.75 : 9.04$ (Chen et al., 2017), and the synaptic weight range is constrained to $[-1, 1]$ (Le Gallo et al., 2023; Lammie et al., 2022). In addition to SAFs, we consider other fault models such as RWFs and CEFs. Faults are injected into each synapse according to a Bernoulli distribution with the probability corresponding to the fault ratio, resulting in uniformly distributed faults across synapses.

We compare our mechanism with six benchmarks (ECOC (Liu et al., 2019), SoftSNN (Putra et al., 2022), Routing (Yang et al., 2022), Astrocyte (Han et al., 2023), FalVolt (Siddique & Hoque, 2023), and LIFA (Yunusoglu et al., 2025)) and one baseline (SNN models without any fault-mitigation mechanism). For all benchmarks and the baseline, 8 time steps are used; RMSE is adopted as the loss function except for ECOC (ECOC uses cross-entropy (Liu et al., 2019)), and Adam is employed as the optimizer (Fang et al., 2023). Models are trained for 50 epochs with a batch size of 100. The learning rate is 0.001 for MLP/VGG models and 0.01 for ResNet models. The number of fragments (time steps) is chosen from $\mathcal{T} = \{2, 4, 8\}$, which are widely used time step settings in SNNs (Li et al., 2023). We use the Poisson encoder to convert input samples into spike trains. For statistical reliability, all experiments are repeated 10 times with different random seeds, and results are presented with a 95% confidence interval.

Due to page limitations, results on additional datasets (UCI-HAR, AudioMNIST, and Tiny-ImageNet), experiments with different time steps for benchmarks (2 and 4 steps), other fault types (RWFs and CEFs), ablation studies, and per-fragment analysis are provided in Appendices B.1, B.2, and B.3. We also demonstrate that our mechanism improves the fault tolerance of hardware-implemented SNNs using a real FPGA device in Appendix B.7.

### 6.2. Comparison with representative benchmarks

Figure 3 compares the classification accuracy of the baseline, the benchmarks, and the proposed mechanism under SAFs.

#### 6.2.1. MLP MODEL

On MNIST and FMNIST, our mechanism consistently outperforms the baseline and benchmarks. This improvement is consistent with our motivation study: input fragmentation prevents the per-step membrane potential from increasing

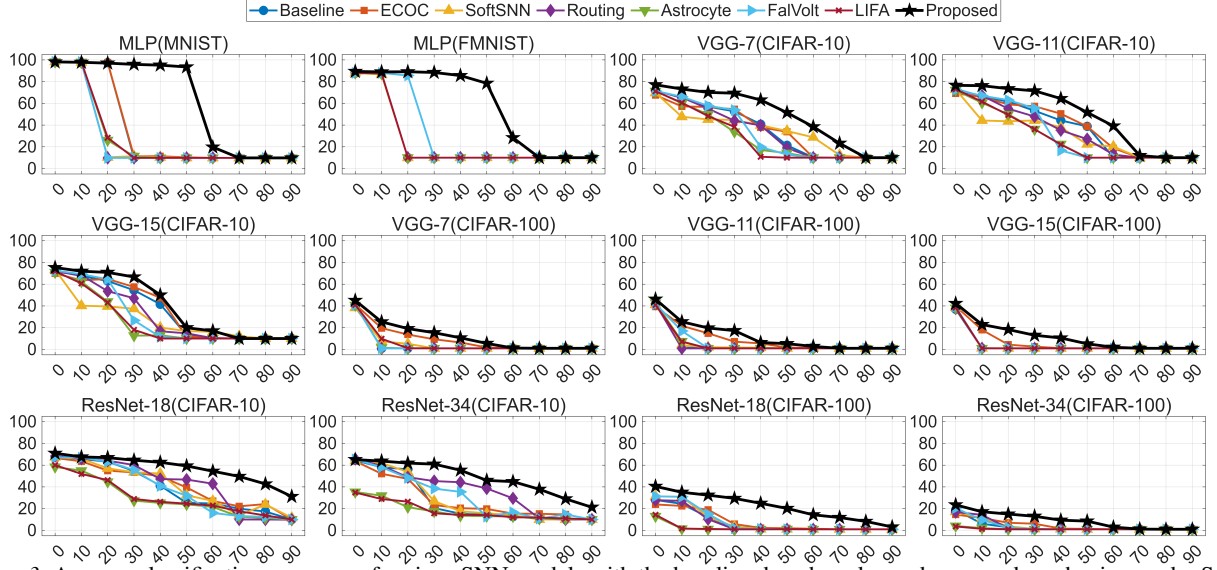

*Figure 3.* Average classification accuracy of various SNN models with the baseline, benchmarks, and proposed mechanism under SAFs. The x-axis is the fault ratio (%) and the y-axis is the accuracy (%).

excessively, thus reducing the likelihood that neuron states remain outside the surrogate gradient corridor and stabilizing learning under synaptic faults.

We observe higher accuracy on MNIST than on FMNIST, which can be attributed to the higher structural complexity of FMNIST samples, requiring finer temporal decomposition to fully mitigate fault-induced capacity degradation. However, the tuned number of fragments for both MNIST and FMNIST is the same according to our measurement. When the same number of fragments is used for both datasets, residual performance gaps remain, indicating that dataset complexity interacts with the degree of fragmentation.

### 6.2.2. VGG MODELS

For CIFAR-10 and CIFAR-100, our mechanism achieves higher accuracy than all benchmarks and the baseline across a wide range of fault ratios. In particular, our method maintains meaningful performance up to fault ratios (e.g., approximately 50–60% for CIFAR-10) higher than the benchmarks. This robustness is consistent with the effect of fragmentation in regulating input load, which mitigates membrane potential saturation even as fault severity increases. We also observe the accuracy decline as model depth increases. This is because gradient vanishing is more likely to occur in deeper networks (Guo et al., 2024).

### 6.2.3. RESNET MODELS

Compared to VGG models, ResNet models exhibit stronger inherent robustness to synaptic faults, with both benchmarks and our method maintaining reasonable accuracy up to fault ratios of 80-90% on CIFAR-10. This robustness is consistent with the improved gradient calculation through residual

blocks. On CIFAR-100, ResNet models also outperform VGG models, reflecting their higher representational capacity. Notably, our mechanism further extends the fault-tolerant regime: ResNet-18 remains effective up to approximately 60-70% fault ratios only with our method, while ResNet-34 remains effective up to roughly 30–40%. These results indicate that our mechanism enhances fault tolerance, even with complex datasets and large models.

We observe that the astrocyte-based approaches (Astrocyte and LIFA) provide limited benefits for deep ResNet models. This is because these methods mimic only biological mechanisms of neuronal activity in brains, which improves robustness in shallow and highly bio-plausible models using a biological unsupervised learning rule (Han et al., 2023; Yunusoglu et al., 2025). In contrast, our mechanism consistently improves fault tolerance across SNN models by directly addressing the bottleneck problem in faulty SNNs.

### 6.3. Comparison with lightweight approaches

We compare the classification accuracy and inference time of the proposed mechanism with lightweight approaches for SNNs' fault tolerance: input suppression, fault hopping, and threshold tuning (Saha et al., 2023; Spyrou et al., 2021; Saha et al., 2024). Table 4 presents the accuracy and summation of inference time over 100 iterations (assuming the actual scenarios) for the SNN models using the lightweight approaches and the proposed mechanism under 30% SAFs in software-based and FPGA-implemented environments[7]. Our mechanism exhibits more effective fault mitigation than other lightweight approaches, while consuming less or com-

---

[7]We explain the settings of our FPGA-implemented environment in Appendix B.7.

*Table 4.* The SNN model's classification accuracy and summation of inference time over 100 repetitions in a 95% confidence interval with other lightweight approaches and our mechanism under the 30% fault ratio of SAFs.

| DATASETS (MODELS) | INPUT SUPPRESSION | FAULT HOPPING | THRESHOLD TUNING | PROPOSED |
|---|---|---|---|---|
| ACCURACY (%) IN SOFTWARE-BASED SNN MODELS | | | | |
| MLP (MNIST) | 94.57 ± 1.29 | 93.11 ± 1.15 | 94.14 ± 1.67 | **95.89 ± 1.45** |
| MLP (FMNIST) | 87.6 ± 1.18 | 85.95 ± 1.24 | 85.13 ± 1.41 | **88.43 ± 2.05** |
| VGG-7 (CIFAR-10) | 62.77 ± 5.16 | 60.97 ± 4.95 | 58.84 ± 5.08 | **69.36 ± 5.2** |
| ACCURACY (%) IN FPGA-IMPLEMENTED SNN MODELS | | | | |
| MLP (MNIST) | 90.18 ± 2.47 | 89.92 ± 2.59 | 90.64 ± 2.44 | **92.96 ± 2.74** |
| MLP (FMNIST) | 80.61 ± 3.73 | 81.11 ± 3.52 | 79.72 ± 3.58 | **84.01 ± 3.65** |
| VGG-7 (CIFAR-10) | 61.09 ± 3.8 | 56.85 ± 3.9 | 53.58 ± 3.72 | **64.71 ± 5.44** |
| INFERENCE TIME (sec) IN SOFTWARE-BASED SNN MODELS | | | | |
| MLP (MNIST) | 1134.56 ± 9.87 | 1017.72 ± 9.16 | 854.49 ± 8.86 | **812.84 ± 8.52** |
| MLP (FMNIST) | 1151.09 ± 14.92 | 1025.16 ± 9.67 | 858.34 ± 9.04 | **848.17 ± 8.35** |
| VGG-7 (CIFAR-10) | 2001.34 ± 26.95 | 1306.74 ± 15.43 | 1170.69 ± 18.34 | **1187.24 ± 13.94** |
| INFERENCE TIME (sec) IN FPGA-IMPLEMENTED SNN MODELS | | | | |
| MLP (MNIST) | 321.04 ± 4.34 | 249.4 ± 3.72 | 204.56 ± 2.49 | **204.16 ± 3.89** |
| MLP (FMNIST) | 326.57 ± 5.13 | 258.37 ± 5.21 | 203.12 ± 2.68 | **211.63 ± 4.16** |
| VGG-7 (CIFAR-10) | 496.78 ± 9.58 | 407.49 ± 10.43 | 314.29 ± 6.86 | **307.44 ± 5.85** |

parable inference time.

We analyze the pros and cons of the proposed mechanism in comparison with the lightweight approaches mentioned in Subsection 3.2 (Saha et al., 2023; Spyrou et al., 2021; Saha et al., 2024). Our mechanism exhibits stronger fault mitigation capability than the lightweight approaches, incurring less overhead than the other approaches. Existing lightweight mechanisms for hardware SNN fault mitigation rely on continuously monitoring internal neuron currents or spike statistics and mitigating the adversarial effects of hardware defects (Spyrou et al., 2021; Saha et al., 2023). Such designs are effective for neuromorphic devices that frequently perform inferences due to their simple operations. However, their computational overhead significantly increases in large SNN models because scanning all weights or neuron states and aggregating their statistics grows exponentially as the model's size increases. Conversely, our mechanism avoids direct inspection of all elements within SNNs or any modification of internal circuits in hardware. Thus, the external controller only needs to exchange a small amount of metadata with the neuromorphic device, rather than full weight or neuron maps (Khan et al., 2024).

With respect to test-based schemes, their computational cost is amortized by executing them only intermittently (Spyrou et al., 2021; Saha et al., 2024). However, such approaches cannot respond promptly to permanent faults that arise during deployment. On the other hand, our mechanism operates continuously during both training and inference, incurs modest per-execution overhead, and can dynamically adapt the fragmentation strategy to the current fault state in ag-

ile deployment environments [8]. The proposed mechanism provides the following complementary points in the design space. First, it scales better than forward-pass approaches with scanning to large hardware-implemented SNNs. The scalability is particularly relevant for modern neuromorphic devices (Yin et al., 2024). Second, although our mechanism incurs higher continuous overhead than periodic self-test approaches, it achieves substantially stronger mitigation of permanent faults.

## 7. Conclusion

In this paper, we propose a practical fault-mitigation mechanism for hardware-implemented SNNs that operates only through input data control without direct access to internal SNN circuits. The proposed mechanism divides each input sample into multiple fragments using a learnable fragmentation strategy, in which both the division boundaries and the number of fragments are optimized during training. Experimental results across multiple SNN models, datasets, and fault settings demonstrate that our mechanism consistently outperforms benchmarks, including those in real hardware environments. By improving robustness to hardware faults without accessing internal circuits, our mechanism provides a hardware-compatible pathway for the reliable deployment of SNNs in practical neuromorphic systems.

---

[8]Input-side adaptation with the fragmentation does not require any access to internal components, allowing the proposed mechanism to be simply deployable. If targeted devices are SpiNNaker 2, Intel Loihi 2, and BrainScale S-2, which support on-chip learning, fine-tuning may be possible in principle. However, our mechanism does not rely on such capabilities (Rostami et al., 2022).

## Acknowledgements

This work was partially sponsored by the National Research Foundation of Korea (NRF) (grant no. RS-2026-25469726) and by Institute of Information & communications Technology Planning & Evaluation (IITP) grant funded by the Korea government (MSIT) (grant no. RS-2026-25523890).

## Impact Statement

This paper presents work whose goal is to advance the fault tolerance of SNNs in neuromorphic devices. There are many potential societal consequences of our work, none of which we feel must be specifically highlighted here.

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

## Code availability

The source code for our work is available on the first author's GitHub: [https://github.com/LeeHyunJongSNN/](https://github.com/LeeHyunJongSNN/) [Learnable_Flow_Control_for_Fault_Tolerant_SNNs](https://github.com/LeeHyunJongSNN/Learnable_Flow_Control_for_Fault_Tolerant_SNNs).

## Appendix index

# A. Discussion

We discuss the limitations of our work and the justification of our assumptions about fault occurrence while training neuromorphic devices.

## A.1. Limitations

Although we proposed a practical mechanism to enhance the fault tolerance of SNNs in neuromorphic devices without access to their internal circuits, it has several limitations. First, we implement SNN models on the FPGA with software-based offline training and do not use online training for FPGA-implemented SNNs, though we conduct simulations on trainable FPGA-implemented SNNs with Vivado emulator. Second, we lack real-world scenarios for neuromorphic devices in our FPGA experiments. Third, the input sequence of fragments and the decoding method do not consider intrinsic SNN dynamics, such as spiking patterns and the temporal memory of LIF neurons. We will address these limitations of our mechanism in ongoing work by adopting a learnable fragmentation and decoding strategy based on the characteristics of LIF neurons.

## A.2. On-chip learning based on surrogate gradients

We target the scenario in which gradient-based learning is executed on-chip during training. Recent studies have demonstrated that neuromorphic devices such as SpiNNaker 2, Intel Loihi 2, and BrainScale S-2 support full or partial on-chip learning with surrogate gradients (Rostami et al., 2022; Lagorce et al., 2015; Stewart et al., 2020; Cramer et al., 2022; Yin et al., 2024; Payvand et al., 2020; Renner et al., 2024). As gradient-based on-chip learning becomes increasingly prevalent, ensuring robustness to hardware faults that may arise during training is critical for maintaining the reliability of neuromorphic systems (Eslami et al., 2024; Martemucci et al., 2025; Rostami et al., 2022; Lagorce et al., 2015; Stewart et al., 2020; Cramer et al., 2022; Yin et al., 2024). Accordingly, investigating fault tolerance in neuromorphic devices that employ surrogate-gradient learning is an important and timely problem. Our problem definition is aligned with the direction in which current neuromorphic hardware is evolving.

One might argue that SNNs can instead be trained offline and subsequently deployed onto neuromorphic hardware. However, this workflow introduces two intrinsic sources of performance degradation. First, deploying offline-trained models onto neuromorphic devices is inherently error-prone: programming noise, device mismatch, and temporal drift can distort imported weights and progressively degrade accuracy (Li et al., 2018; Wang et al., 2019). Second, neuromorphic systems are often designed for edge-computing scenarios, where personalization and non-stationary operating conditions require continual on-device adaptation (Martemucci et al., 2025; Zhang et al., 2024a). Such frequent weight updates increase exposure to write disturbances and endurance-related degradation, making hardware faults more likely over time (Tung et al., 2023).

In this context, fault-mitigation strategies that are compatible with on-chip learning are essential. By operating exclusively at the input level and avoiding any modification of internal synaptic weights or hardware circuits, the proposed mechanism is naturally suited to improving the reliability of on-chip learning under realistic hardware constraints, thereby supporting robust deployment across diverse real-world applications.

## B. Additional experimental results on classification accuracy

We present additional experimental results demonstrating the effectiveness of our mechanism under diverse environments and scenarios. We further include experimental results obtained from a real FPGA-based implementation. Furthermore, we analyze the performance of our mechanism by changing the settings of our mechanism.

### B.1. Additional datasets beyond MNIST, FMNIST, CIFAR-10, and CIFAR-100

We use UCI-HAR, AudioMNIST, and Tiny-ImageNet to evaluate the fault mitigation ability of our mechanism on a sequential and large-scale dataset.

### B.1.1. SEQUENTIAL DATASET

● **Input data conversion:** For spatial fragmentation to a 1D sequence data samples, we reshape each 1D sequence into a fixed 2D representation of size $\mathcal{H} \times \mathcal{W}$. $(\mathcal{H}, \mathcal{W})$ is chosen by satisfying three conditions: 1) $\mathcal{H}\mathcal{W} \geq \mathcal{N}$ and 2) $arg\ min(\mathcal{H}\mathcal{W} - \mathcal{N})$, and 3) $|\mathcal{H} - \mathcal{W}| < \mathcal{K}$. Here, $\mathcal{N}$ is the dimension of the 1D data sequence, and $\mathcal{K}$ is the configurable parameter that determines the shape of the 2D rectangle. The second condition indicates that the number of unused pixels, which are zero-padded, should be minimized (Yin et al., 2018). We use a deterministic row-major mapping from the 1D sequence to the 2D grid. Therefore, flattening the grid recovers the original 1D order, while enabling spatial fragmentation.

To demonstrate that our mechanism works well with the models using sequential datasets, we conduct experiments with UCI-HAR and AudioMNIST. UCI-HAR comprises six types of human activities collected by smartphone inertial sensors, and AudioMNIST contains verbal sounds of digits (Reyes-Ortiz et al., 2013; Becker et al., 2024).

*Table 5.* The MLP model's classification accuracy in a 95% confidence interval using UCI-HAR, and AudioMNIST under SAFs.

| FAULT RATIO(%) | BASELINE | ECOC | SOFTSNN | ROUTING | ASTROCYTE | FALVOLT | LIFA | PROPOSED |
|---|---|---|---|---|---|---|---|---|
| | | | ACCURACY (%) WITH MLP (UCI-HAR) | | | | | |
| 0 | $67.73 \pm 4.64$ | $69.86 \pm 4.43$ | $66.41 \pm 4.81$ | $66.08 \pm 4.71$ | $65.41 \pm 4.48$ | $68.47 \pm 4.29$ | $68.61 \pm 4.19$ | $\mathbf{71.86 \pm 3.75}$ |
| 10 | $36.39 \pm 4.12$ | $38.02 \pm 4.37$ | $34.49 \pm 3.93$ | $35.89 \pm 4.02$ | $32.57 \pm 4.22$ | $37.94 \pm 3.82$ | $30.19 \pm 4.84$ | $\mathbf{64.62 \pm 4.71}$ |
| 20 | $27.01 \pm 5.54$ | $26.27 \pm 4.65$ | $29.72 \pm 4.85$ | $26.91 \pm 4.32$ | $24.49 \pm 3.98$ | $26.34 \pm 3.95$ | $25.02 \pm 3.87$ | $\mathbf{59.86 \pm 4.89}$ |
| 30 | $20.39 \pm 3.29$ | $17.1 \pm 0$ | $21.9 \pm 2.26$ | $21.16 \pm 3.02$ | $17.1 \pm 0$ | $20.85 \pm 3.19$ | $17.1 \pm 0$ | $\mathbf{52.83 \pm 4.07}$ |
| 40 | $17.1 \pm 0$ | $17.1 \pm 0$ | $17.1 \pm 0$ | $17.1 \pm 0$ | $17.1 \pm 0$ | $16.93 \pm 0$ | $17.1 \pm 0$ | $\mathbf{52.07 \pm 4.16}$ |
| 50 | $17.1 \pm 0$ | $17.1 \pm 0$ | $17.1 \pm 0$ | $17.1 \pm 0$ | $17.1 \pm 0$ | $16.93 \pm 0$ | $17.1 \pm 0$ | $\mathbf{48.55 \pm 3.85}$ |
| 60 | $17.1 \pm 0$ | $17.1 \pm 0$ | $17.1 \pm 0$ | $17.1 \pm 0$ | $17.1 \pm 0$ | $16.93 \pm 0$ | $17.1 \pm 0$ | $\mathbf{33.46 \pm 4.24}$ |
| 70 | $17.1 \pm 0$ | $17.1 \pm 0$ | $17.1 \pm 0$ | $17.1 \pm 0$ | $17.1 \pm 0$ | $16.93 \pm 0$ | $17.1 \pm 0$ | $17.1 \pm 0$ |
| 80 | $17.1 \pm 0$ | $17.1 \pm 0$ | $17.1 \pm 0$ | $17.1 \pm 0$ | $17.1 \pm 0$ | $16.93 \pm 0$ | $17.1 \pm 0$ | $17.1 \pm 0$ |
| 90 | $17.1 \pm 0$ | $17.1 \pm 0$ | $17.1 \pm 0$ | $17.1 \pm 0$ | $17.1 \pm 0$ | $16.93 \pm 0$ | $17.1 \pm 0$ | $17.1 \pm 0$ |
| | | | ACCURACY (%) WITH MLP (AUDIOMNIST) | | | | | |
| 0 | $96.91 \pm 1.14$ | $97.08 \pm 0.97$ | $96.83 \pm 1.01$ | $96.69 \pm 0.91$ | $96.85 \pm 1.13$ | $96.79 \pm 0.99$ | $96.75 \pm 1.27$ | $\mathbf{97.49 \pm 1.02}$ |
| 10 | $95.52 \pm 1.07$ | $95.76 \pm 1.51$ | $95.61 \pm 1.42$ | $95.4 \pm 1.76$ | $95.19 \pm 1.56$ | $95.62 \pm 1.63$ | $95.14 \pm 1.83$ | $\mathbf{95.57 \pm 1.48}$ |
| 20 | $94.61 \pm 2.64$ | $95.01 \pm 2.13$ | $94.93 \pm 2.96$ | $94.68 \pm 3.05$ | $90.37 \pm 3.71$ | $94.93 \pm 2.84$ | $91.96 \pm 4.23$ | $\mathbf{94.92 \pm 2.25}$ |
| 30 | $93.46 \pm 3.47$ | $94.22 \pm 3.51$ | $90.14 \pm 4.05$ | $93.78 \pm 3.73$ | $59.81 \pm 6.04$ | $93.77 \pm 3.51$ | $62.67 \pm 6.15$ | $\mathbf{94.07 \pm 3.68}$ |
| 40 | $90.62 \pm 3.86$ | $92.68 \pm 3.94$ | $67.58 \pm 7.18$ | $91.47 \pm 3.69$ | $52.24 \pm 6.67$ | $91.49 \pm 3.81$ | $59.64 \pm 7.34$ | $93.7 \pm 3.73$ |
| 50 | $87.02 \pm 4.24$ | $88.93 \pm 4.36$ | $56.83 \pm 7.43$ | $88.79 \pm 4.86$ | $31.67 \pm 4.34$ | $88.62 \pm 4.49$ | $37.29 \pm 5.29$ | $\mathbf{92.83 \pm 4.14}$ |
| 60 | $56.43 \pm 5.69$ | $68.71 \pm 5.36$ | $41.94 \pm 6.49$ | $69.16 \pm 5.71$ | $11.73 \pm 1.73$ | $67.48 \pm 5.61$ | $19.09 \pm 3.62$ | $\mathbf{81.99 \pm 5.87}$ |
| 70 | $20.29 \pm 4.31$ | $31.71 \pm 5.29$ | $12.12 \pm 2.12$ | $28.29 \pm 3.42$ | $10 \pm 0$ | $32.36 \pm 4.93$ | $10 \pm 0$ | $\mathbf{46.47 \pm 6.12}$ |
| 80 | $14.13 \pm 2.82$ | $22.48 \pm 4.86$ | $10 \pm 0$ | $16.34 \pm 3.03$ | $10 \pm 0$ | $18.76 \pm 4.19$ | $10 \pm 0$ | $\mathbf{32.49 \pm 5.7}$ |
| 90 | $12.04 \pm 1.53$ | $15.86 \pm 3.97$ | $10 \pm 0$ | $13.18 \pm 1.49$ | $10 \pm 0$ | $13.71 \pm 2.6$ | $10 \pm 0$ | $\mathbf{21.68 \pm 4.84}$ |

Table 5 presents classification accuracy for MLP-based SNNs on UCI-HAR and AudioMNIST under SAFs. Across both datasets, models integrated with our mechanism achieve higher accuracy than the baseline and benchmarks and remain robust over a wider range of fault ratios (up to approximately 60% for UCI-HAR and 90% for AudioMNIST). These results indicate that input fragmentation remains effective for sequential data, extending the applicability of our approach to various domains, such as audio data and sensing data.

### B.1.2. LARGE IMAGE DATASET WITH COMPLEX MODELS

We utilize the vanilla ResNet-34, SEW-ResNet-34, and SpikFormer models to classify samples from Tiny-ImageNet and ImageNet-1K. ImageNet is an image dataset containing $224 \times 224$ pixel images across 1000 classes, and Tiny-ImageNet is a subset of the ImageNet dataset comprising $64 \times 64$ pixel images across 200 classes (Hu et al., 2023; Zhou et al., 2023). We compare the classification accuracy of models equipped with our mechanism against baseline and benchmark models on Tiny-ImageNet under SAF conditions. Unlike experiments using MNIST, CIFAR-10, and CIFAR-100, we use 200 training epochs to train the models with complex datasets.

*Table 6.* Various large models' classification accuracy in a 95% confidence interval using Tiny-ImageNet and ImageNet-1K under SAFs.

| FAULT RATIO(%) | BASELINE | ECOC | SOFTSNN | ROUTING | ASTROCYTE | FALVOLT | LIFA | PROPOSED |
|---|---|---|---|---|---|---|---|---|
| ACCURACY (%) WITH VANILLA RESNET-34 (TINY-IMAGENET) | | | | | | | | |
| 0 | $50.17 \pm 1.16$ | $\mathbf{50.96 \pm 1.02}$ | $49.14 \pm 1.24$ | $50.25 \pm 1.19$ | $50.14 \pm 1.23$ | $50.22 \pm 1.15$ | $50.19 \pm 1.26$ | $50.24 \pm 1.27$ |
| 10 | $42.35 \pm 3.91$ | $43.27 \pm 3.78$ | $42.06 \pm 4.16$ | $43.68 \pm 4.04$ | $38.52 \pm 5.26$ | $44.27 \pm 4.28$ | $40.02 \pm 5.11$ | $\mathbf{45.89 \pm 3.96}$ |
| 20 | $19.77 \pm 5.16$ | $23.65 \pm 5.45$ | $20.35 \pm 5.23$ | $24.52 \pm 5.38$ | $18.14 \pm 6.12$ | $25.36 \pm 5.64$ | $18.89 \pm 6.34$ | $\mathbf{27.41 \pm 5.59}$ |
| 30 | $5.58 \pm 2.43$ | $8.16 \pm 2.61$ | $6.74 \pm 2.48$ | $9.94 \pm 2.73$ | $6.02 \pm 2.39$ | $10.15 \pm 2.97$ | $6.38 \pm 2.36$ | $\mathbf{16.4 \pm 3.82}$ |
| 40 | $0.5 \pm 0$ | $0.5 \pm 0$ | $0.5 \pm 0$ | $0.5 \pm 0$ | $0.5 \pm 0$ | $0.5 \pm 0$ | $0.5 \pm 0$ | $\mathbf{7.79 \pm 2.65}$ |
| 50 | $0.5 \pm 0$ | $0.5 \pm 0$ | $0.5 \pm 0$ | $0.5 \pm 0$ | $0.5 \pm 0$ | $0.5 \pm 0$ | $0.5 \pm 0$ | $\mathbf{3.32 \pm 1.04}$ |
| 60 | $0.5 \pm 0$ | $0.5 \pm 0$ | $0.5 \pm 0$ | $0.5 \pm 0$ | $0.5 \pm 0$ | $0.5 \pm 0$ | $0.5 \pm 0$ | $0.5 \pm 0$ |
| 70 | $0.5 \pm 0$ | $0.5 \pm 0$ | $0.5 \pm 0$ | $0.5 \pm 0$ | $0.5 \pm 0$ | $0.5 \pm 0$ | $0.5 \pm 0$ | $0.5 \pm 0$ |
| 80 | $0.5 \pm 0$ | $0.5 \pm 0$ | $0.5 \pm 0$ | $0.5 \pm 0$ | $0.5 \pm 0$ | $0.5 \pm 0$ | $0.5 \pm 0$ | $0.5 \pm 0$ |
| 90 | $0.5 \pm 0$ | $0.5 \pm 0$ | $0.5 \pm 0$ | $0.5 \pm 0$ | $0.5 \pm 0$ | $0.5 \pm 0$ | $0.5 \pm 0$ | $0.5 \pm 0$ |
| ACCURACY (%) WITH SEW-RESNET-34 (IMAGENET-1K) | | | | | | | | |
| 0 | $67.15 \pm 1.02$ | $67.92 \pm 1.11$ | $66.8 \pm 1.23$ | $67.26 \pm 0.98$ | $66.28 \pm 1.36$ | $67.65 \pm 1.21$ | $67.01 \pm 1.18$ | $\mathbf{68.64 \pm 1.09}$ |
| 10 | $62.69 \pm 3.42$ | $63.88 \pm 3.36$ | $61.02 \pm 3.72$ | $65.29 \pm 3.5$ | $60.48 \pm 4.47$ | $64.15 \pm 4.06$ | $61.26 \pm 4.21$ | $\mathbf{65.62 \pm 3.88}$ |
| 20 | $51.72 \pm 5.81$ | $53.64 \pm 5.78$ | $51.13 \pm 5.94$ | $53.97 \pm 5.84$ | $48.35 \pm 7.19$ | $52.02 \pm 6.21$ | $48.99 \pm 7.34$ | $\mathbf{55.41 \pm 6.28}$ |
| 30 | $39.48 \pm 7.75$ | $41.25 \pm 8.01$ | $38.29 \pm 8.58$ | $42.85 \pm 8.36$ | $36.4 \pm 8.74$ | $41.93 \pm 7.92$ | $37.64 \pm 8.84$ | $\mathbf{44.34 \pm 8.26}$ |
| 40 | $21.37 \pm 6.19$ | $22.68 \pm 6.23$ | $20.22 \pm 6.11$ | $24.91 \pm 6.49$ | $12.51 \pm 4.96$ | $22.73 \pm 6.56$ | $14.87 \pm 5.67$ | $\mathbf{28.75 \pm 6.61}$ |
| 50 | $7.75 \pm 2.59$ | $12.51 \pm 3.48$ | $5.94 \pm 2.61$ | $14.56 \pm 3.93$ | $2.87 \pm 1.17$ | $10.52 \pm 3.28$ | $4.35 \pm 2.14$ | $\mathbf{20.29 \pm 4.28}$ |
| 60 | $0.1 \pm 0$ | $0.1 \pm 0$ | $0.1 \pm 0$ | $4.13 \pm 2.29$ | $0.1 \pm 0$ | $0.1 \pm 0$ | $0.1 \pm 0$ | $\mathbf{9.96 \pm 3.03}$ |
| 70 | $0.1 \pm 0$ | $0.1 \pm 0$ | $0.1 \pm 0$ | $0.1 \pm 0$ | $0.1 \pm 0$ | $0.1 \pm 0$ | $0.1 \pm 0$ | $\mathbf{4.58 \pm 1.96}$ |
| 80 | $0.1 \pm 0$ | $0.1 \pm 0$ | $0.1 \pm 0$ | $0.1 \pm 0$ | $0.1 \pm 0$ | $0.1 \pm 0$ | $0.1 \pm 0$ | $0.1 \pm 0$ |
| 90 | $0.1 \pm 0$ | $0.1 \pm 0$ | $0.1 \pm 0$ | $0.1 \pm 0$ | $0.1 \pm 0$ | $0.1 \pm 0$ | $0.1 \pm 0$ | $0.1 \pm 0$ |
| ACCURACY (%) WITH SPIKFORMER (IMAGENET-1K) | | | | | | | | |
| 0 | $71.92 \pm 0.46$ | $72.47 \pm 0.49$ | $72.01 \pm 0.4$ | $71.96 \pm 0.48$ | $71.33 \pm 0.57$ | $72.29 \pm 0.53$ | $71.37 \pm 0.49$ | $\mathbf{72.94 \pm 0.52}$ |
| 10 | $65.42 \pm 0.98$ | $68.11 \pm 0.86$ | $65.38 \pm 1.14$ | $66.29 \pm 1.08$ | $63.44 \pm 1.46$ | $66.81 \pm 0.94$ | $63.83 \pm 1.4$ | $\mathbf{70.15 \pm 0.93}$ |
| 20 | $60.19 \pm 1.75$ | $62.74 \pm 1.69$ | $60.84 \pm 2.02$ | $64.83 \pm 1.91$ | $57.01 \pm 3.36$ | $64.46 \pm 1.73$ | $58.19 \pm 3.23$ | $\mathbf{67.56 \pm 1.88}$ |
| 30 | $51.34 \pm 4.48$ | $53.48 \pm 4.52$ | $51.15 \pm 4.67$ | $57.29 \pm 4.54$ | $48.98 \pm 5.19$ | $56.61 \pm 4.47$ | $50.17 \pm 5.08$ | $\mathbf{58.38 \pm 4.61}$ |
| 40 | $35.61 \pm 6.72$ | $37.08 \pm 6.96$ | $29.48 \pm 7.58$ | $39.02 \pm 7.2$ | $33.12 \pm 7.71$ | $38.13 \pm 7.21$ | $33.96 \pm 7.53$ | $\mathbf{40.24 \pm 7.04}$ |
| 50 | $20.45 \pm 8.14$ | $21.33 \pm 8.97$ | $8.59 \pm 4.93$ | $22.91 \pm 7.86$ | $11.29 \pm 6.05$ | $22.57 \pm 7.93$ | $12.75 \pm 6.32$ | $\mathbf{25.55 \pm 8.49}$ |
| 60 | $11.6 \pm 5.21$ | $11.54 \pm 4.91$ | $0.1 \pm 0$ | $12.35 \pm 5.27$ | $1.48 \pm 0.92$ | $13.05 \pm 5.43$ | $2.36 \pm 1.15$ | $\mathbf{17.21 \pm 5.36}$ |
| 70 | $0.1 \pm 0$ | $0.1 \pm 0$ | $0.1 \pm 0$ | $0.1 \pm 0$ | $0.1 \pm 0$ | $0.1 \pm 0$ | $0.1 \pm 0$ | $\mathbf{11.42 \pm 3.42}$ |
| 80 | $0.1 \pm 0$ | $0.1 \pm 0$ | $0.1 \pm 0$ | $0.1 \pm 0$ | $0.1 \pm 0$ | $0.1 \pm 0$ | $0.1 \pm 0$ | $\mathbf{4.64 \pm 1.57}$ |
| 90 | $0.1 \pm 0$ | $0.1 \pm 0$ | $0.1 \pm 0$ | $0.1 \pm 0$ | $0.1 \pm 0$ | $0.1 \pm 0$ | $0.1 \pm 0$ | $0.1 \pm 0$ |

Table 6 presents the classification accuracy of the ResNet-34, SEW-ResNet-34, and SpikFormer models with baseline methods, benchmark methods, and the proposed mechanism on Tiny-ImageNet and ImageNet-1K. Similar to Table 5, the model employing our mechanism achieves the highest classification accuracy under a range of fault ratios when we use complex models and datasets.

## B.2. Changing the number of time steps for the baseline and benchmarks

We vary the number of time steps used by the benchmarks and baseline to two and four to demonstrate that our mechanism consistently outperforms these methods across different time steps. Note that our mechanism does not adhere to a fixed number of time steps, as it learns the effective number of time steps during training. The experimental results prove that the SNN models with our mechanism exhibit better fault robustness than the models with the benchmarks.

### B.2.1. 2 TIME STEPS

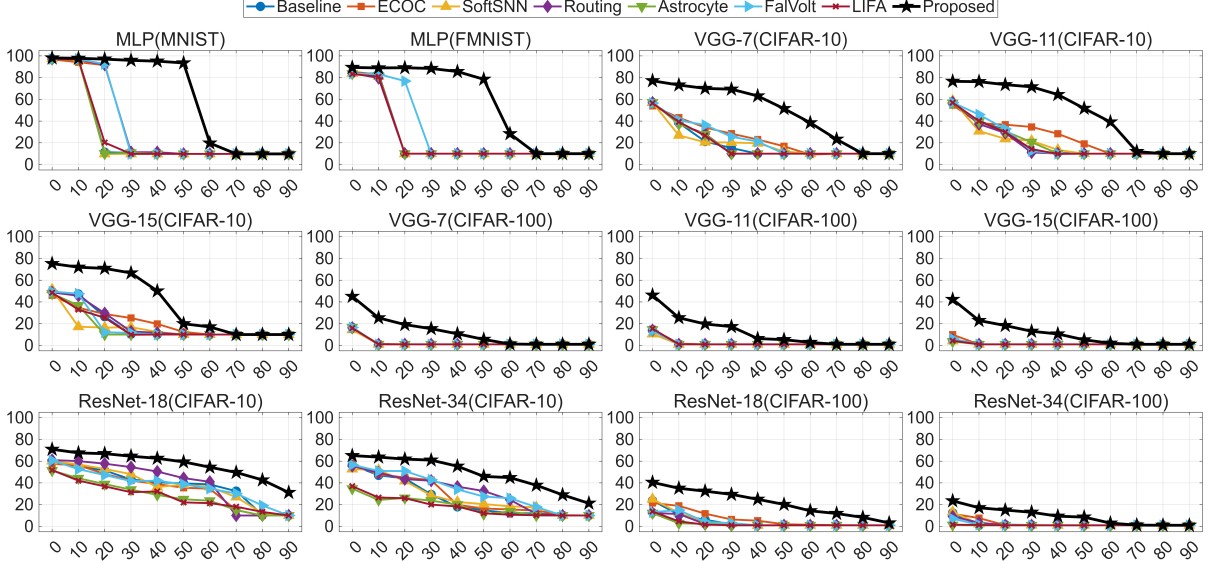

*Figure 4.* Average classification accuracy of various SNN models with the baseline, benchmarks, and proposed mechanism under SAFs when the number of time steps for benchmarks and baseline is two. The x-axis is the fault ratio (%) and the y-axis is the accuracy (%).

Figure 4 illustrates the classification accuracy of SNN models using the baseline, benchmark methods, and the proposed mechanism under SAFs when the number of time steps is set to two. Similar to Figure 3, models employing our mechanism achieve the highest classification accuracy.

### B.2.2. 4 TIME STEPS

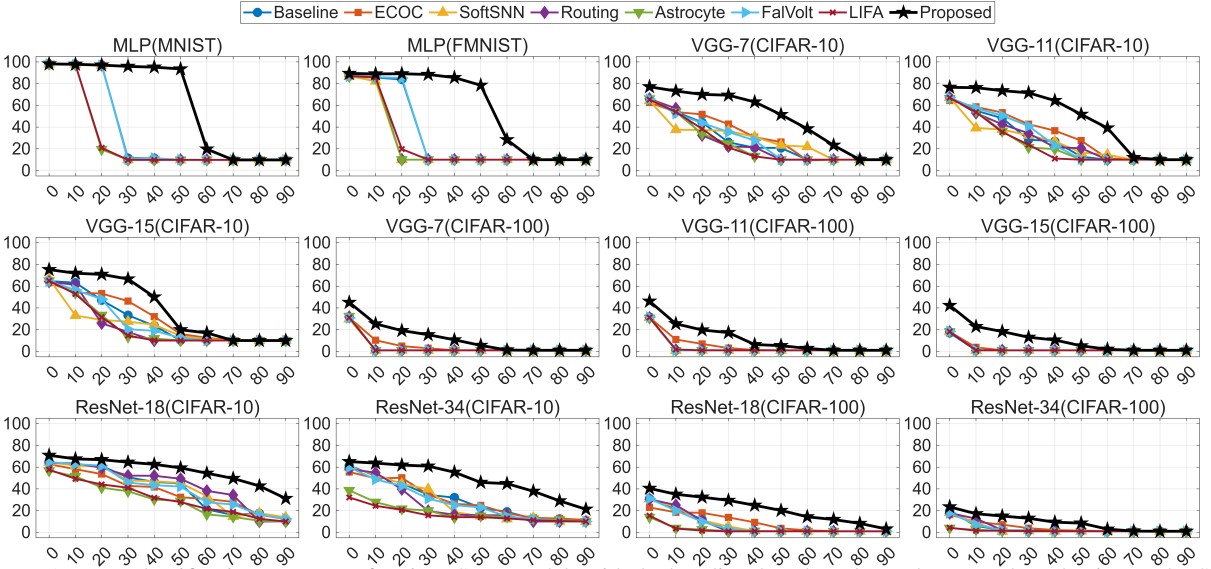

*Figure 5.* Average classification accuracy of various SNN models with the baseline, benchmarks, and proposed mechanism under SAFs when the number of time steps for benchmarks and baseline is four. The x-axis is the fault ratio (%) and the y-axis is the accuracy (%).

Figure 5 depicts the classification accuracy of SNN models with the baseline, the benchmarks, and the proposed mechanism

under SAFs when the number of time steps for the benchmarks and baseline is configured to four. Similar to the experimental results for two and eight time steps, models employing our mechanism achieve the highest classification accuracy.

## B.3. Under different types of synaptic faults

We inject RWFs and CEFs into the synapses of the SNN models and measure the fault mitigation ability of the benchmarks and the proposed mechanism.

### B.3.1. RWFs

We use a Gaussian distribution to model RWFs, setting the standard deviation of the distribution to 0.5 (Garaffa et al., 2021; Spyrou et al., 2021; Vatajelu et al., 2019).

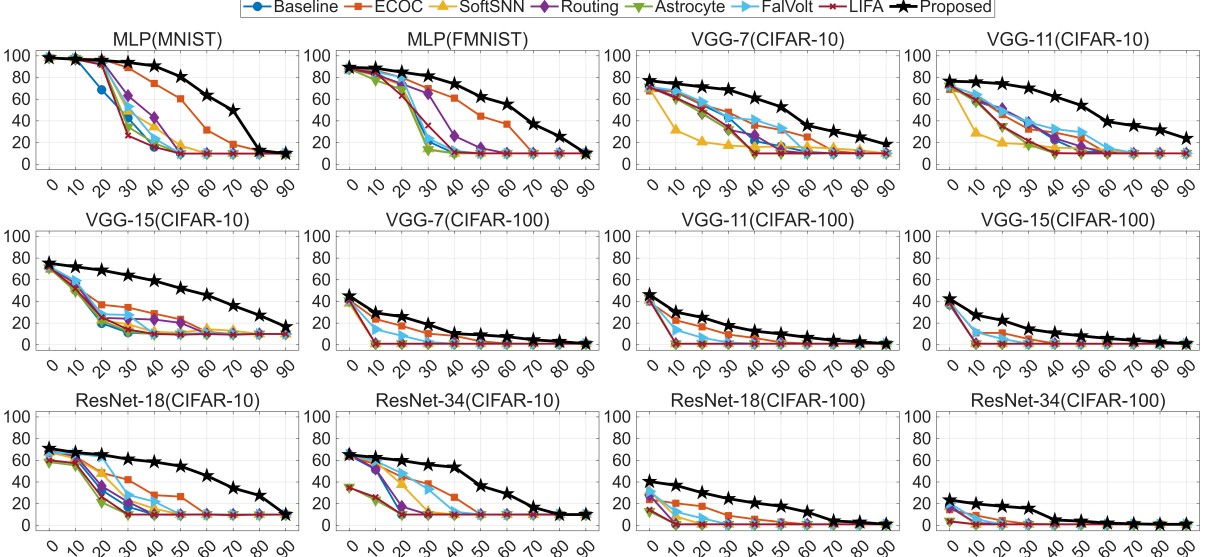

*Figure 6.* Average classification accuracy of various SNN models with the baseline, benchmarks, and proposed mechanism under RWFs. The x-axis is the fault ratio (%) and the y-axis is the accuracy (%).

Figure 6 depicts the classification accuracy of SNN models employing baseline methods, benchmark methods, and the proposed mechanism under RWFs. Across MNIST, FMNIST, CIFAR-10, and CIFAR-100, models with our mechanism consistently achieve the highest classification accuracy. Interestingly, ECOC also demonstrates strong fault-mitigation performance under RWFs. This observation is aligned with the use of error-correcting codes in ECOC, which are known to provide robustness against random Gaussian perturbations (Liu et al., 2019).

### B.3.2. CEFs

CEFs randomly alter synaptic connectivity patterns, changing the synaptic connection topology of SNNs and damaging the learned information in them (Vatajelu et al., 2019). We inject CEFs into synapses with a 0-90% fault ratio.

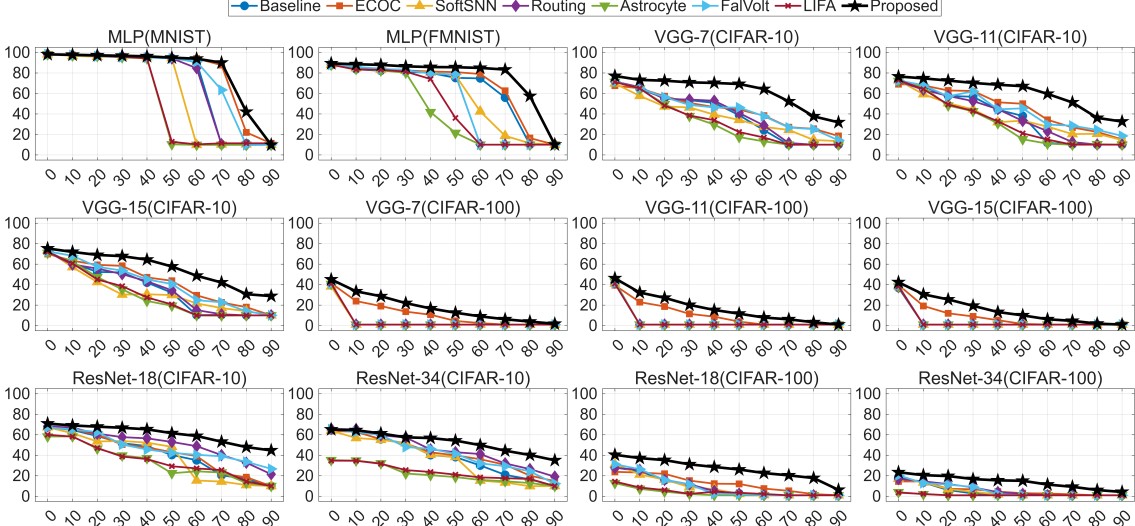

*Figure 7.* Average classification accuracy of various SNN models with the baseline, benchmarks, and proposed mechanism under CEFs. The x-axis is the fault ratio (%) and the y-axis is the accuracy (%).

Figure 7 illustrates the classification accuracy of SNN models employing the baseline method, benchmark methods, and the proposed mechanism under CEFs. Across fault ratios, datasets, and model architectures, our mechanism consistently outperforms the baseline and benchmark methods. Notably, the observed classification accuracy under CEFs is higher than that under SAFs and RWFs within the same experimental setting. Moreover, meaningful accuracy is maintained over a wide range of fault ratios (approximately up to 80% for MNIST/FMNIST and up to 90% for CIFAR-10). This empirical trend is consistent with the fault characteristics of CEFs, in which the weights of faulty synapses are fixed to values drawn from a uniform distribution. Such faults limit the rapid growth of membrane potentials and thereby reduce the likelihood that the membrane potential lies beyond the surrogate gradient corridor.

## B.4. Ablation study on the combination of our mechanism

We conduct an ablation study by varying the components of the proposed mechanism under the following settings: 1) Baseline, 2) LD, 3) LD+LN, 4) LD+LN+BL, and 5) LD+LN+BL+ED (Proposed). Here, Learnable Division (LD) denotes learnable division, which trains the division boundaries of input fragments (Section 5.2), while Learnable Number (LN) denotes the learnable number of fragments, which determines the number of fragments adaptively during training (Section 5.3). Balance Loss (BL) indicates loss regularization (Section 5.4), and Entropy Decoding (ED) refers to entropy-based decoding (Section 5.4). In baseline settings, neither input fragmentation nor loss regularization is adopted.

### B.4.1. MLP MODEL (MNIST AND FMNIST)

*Table 7.* The MLP model's classification accuracy in a 95% confidence interval with different settings of our mechanism under SAFs.

| FAULT RATIO(%) | BASELINE | LD | LD + LN | LD + LN + BL | PROPOSED |
|---|---|---|---|---|---|
| ACCURACY (%) WITH MLP (MNIST) | | | | | |
| 0 | **98.19 ± 0.76** | 97.49 ± 0.52 | 97.59 ± 0.61 | 97.76 ± 0.59 | 98.15 ± 0.84 |
| 10 | 97.58 ± 1.21 | 96.34 ± 1.26 | 96.52 ± 0.89 | 96.82 ± 1.12 | **97.79 ± 0.91** |
| 20 | **97.47 ± 1.41** | 96.42 ± 2.15 | 96.78 ± 2.19 | 96.16 ± 1.86 | 97.03 ± 1.37 |
| 30 | 9.8 ± 0 | 93.91 ± 3.88 | 94.96 ± 2.67 | 94.76 ± 2.92 | **95.89 ± 1.45** |
| 40 | 9.8 ± 0 | 87.18 ± 4.69 | 91.27 ± 3.31 | 93.24 ± 3.59 | **95.15 ± 1.66** |
| 50 | 9.8 ± 0 | 21.86 ± 2.91 | 34.31 ± 4.04 | 56.81 ± 5.15 | **93.52 ± 2.18** |
| 60 | 9.8 ± 0 | 9.8 ± 0 | 9.8 ± 0 | 9.8 ± 0 | **10.75 ± 0.95** |
| 70 | 9.8 ± 0 | 9.8 ± 0 | 9.8 ± 0 | 9.8 ± 0 | 9.8 ± 0 |
| 80 | 9.8 ± 0 | 9.8 ± 0 | 9.8 ± 0 | 9.8 ± 0 | 9.8 ± 0 |
| 90 | 9.8 ± 0 | 9.8 ± 0 | 9.8 ± 0 | 9.8 ± 0 | 9.8 ± 0 |
| ACCURACY (%) WITH MLP (FMNIST) | | | | | |
| 0 | 87.93 ± 1.28 | 87.82 ± 1.19 | 87.98 ± 1.19 | 88.35 ± 1.4 | **89.38 ± 1.13** |
| 10 | 87.81 ± 1.82 | 86.93 ± 1.81 | 87.12 ± 1.5 | 87.02 ± 1.84 | **88.95 ± 1.39** |
| 20 | 10 ± 0 | 85.84 ± 2.24 | 86.27 ± 2.31 | 87.34 ± 2.19 | **89.1 ± 1.52** |
| 30 | 10 ± 0 | 84.71 ± 3.26 | 84.96 ± 2.79 | 85.97 ± 2.72 | **88.43 ± 2.05** |
| 40 | 10 ± 0 | 83.68 ± 5.37 | 84.25 ± 5.01 | 85.27 ± 4.93 | **85.48 ± 4.83** |
| 50 | 10 ± 0 | 44.25 ± 6.63 | 58.74 ± 6.08 | 69.16 ± 6.87 | **78.48 ± 5.79** |
| 60 | 10 ± 0 | 10 ± 0 | 10.9 ± 0.9 | 12.32 ± 2.05 | **13.34 ± 2.11** |
| 70 | 10 ± 0 | 10 ± 0 | 10 ± 0 | 10 ± 0 | 10 ± 0 |
| 80 | 10 ± 0 | 10 ± 0 | 10 ± 0 | 10 ± 0 | 10 ± 0 |
| 90 | 10 ± 0 | 10 ± 0 | 10 ± 0 | 10 ± 0 | 10 ± 0 |

Table 7 presents the classification accuracy of the different ablation settings when an MLP-based SNN is evaluated on the MNIST and Fashion-MNIST datasets. Overall, we observe that both classification accuracy and the fault-robust regime (i.e., the range of fault ratios where meaningful accuracies are shown) improve as additional components of the proposed mechanism are incorporated. Interestingly, for the MNIST dataset, the baseline setting achieves the highest accuracy when the fault ratio is low (below 20%). This behavior can be attributed to the relatively simple structure of MNIST inputs, for which the effective processing capacity of the SNN is rarely constrained at low fault ratios. As a result, input fragmentation provides limited benefit in this regime, allowing the baseline setting to attain the highest accuracy.

### B.4.2. VGG-7 AND RESNET-18 MODELS (CIFAR-10 AND CIFAR-100)

*Table 8.* The VGG-7 and ResNet-18 models' classification accuracy in a 95% confidence interval with different settings of our mechanism under SAFs.

| FAULT RATIO(%) | BASELINE | LD | LD + LN | LD + LN + BL | PROPOSED |
|---|---|---|---|---|---|
| | | ACCURACY (%) WITH VGG-7 (CIFAR-10) | | | |
| 0 | $71.74 \pm 1.28$ | $74.92 \pm 1.42$ | $74.63 \pm 1.16$ | $75.72 \pm 1.35$ | $\mathbf{77.09 \pm 1.27}$ |
| 10 | $65.05 \pm 3.72$ | $71.06 \pm 2.35$ | $70.91 \pm 1.81$ | $71.87 \pm 1.61$ | $\mathbf{73.19 \pm 1.69}$ |
| 20 | $56.97 \pm 5.11$ | $66.87 \pm 4.23$ | $68.98 \pm 3.83$ | $69.19 \pm 3.74$ | $\mathbf{70.08 \pm 4.52}$ |
| 30 | $52.65 \pm 4.92$ | $63.45 \pm 5.48$ | $66.15 \pm 5.47$ | $67.45 \pm 5.62$ | $\mathbf{69.36 \pm 5.2}$ |
| 40 | $40.86 \pm 4.48$ | $57.14 \pm 5.81$ | $61.86 \pm 5.85$ | $62.49 \pm 5.88$ | $\mathbf{63.14 \pm 5.96}$ |
| 50 | $21.43 \pm 3.84$ | $42.27 \pm 5.65$ | $48.73 \pm 6.39$ | $49.34 \pm 6.71$ | $\mathbf{51.46 \pm 6.74}$ |
| 60 | $10.22 \pm 0.22$ | $31.76 \pm 4.97$ | $35.97 \pm 5.72$ | $36.69 \pm 5.41$ | $\mathbf{38.48 \pm 5.13}$ |
| 70 | $10 \pm 0$ | $17.92 \pm 3.83$ | $21.35 \pm 4.02$ | $22.15 \pm 3.57$ | $\mathbf{23.25 \pm 4.35}$ |
| 80 | $10 \pm 0$ | $10 \pm 0$ | $10 \pm 0$ | $10 \pm 0$ | $10 \pm 0$ |
| 90 | $10 \pm 0$ | $10 \pm 0$ | $10 \pm 0$ | $10 \pm 0$ | $10 \pm 0$ |
| | | ACCURACY (%) WITH RESNET-18 (CIFAR-100) | | | |
| 0 | $27.98 \pm 4.67$ | $37.92 \pm 4.55$ | $37.75 \pm 4.48$ | $38.92 \pm 4.82$ | $\mathbf{40.48 \pm 3.71}$ |
| 10 | $24.43 \pm 4.13$ | $33.03 \pm 4.49$ | $32.86 \pm 4.46$ | $34.54 \pm 4.56$ | $\mathbf{35.07 \pm 4.18}$ |
| 20 | $12.04 \pm 2.86$ | $28.48 \pm 5.43$ | $29.14 \pm 5.62$ | $32.01 \pm 5.45$ | $\mathbf{32.39 \pm 5.55}$ |
| 30 | $2.98 \pm 1.09$ | $26.38 \pm 5.29$ | $27.53 \pm 5.13$ | $28.93 \pm 5.31$ | $\mathbf{29.45 \pm 5.6}$ |
| 40 | $1.26 \pm 0.26$ | $21.64 \pm 4.57$ | $22.17 \pm 4.39$ | $22.86 \pm 4.72$ | $\mathbf{24.9 \pm 5.91}$ |
| 50 | $1.12 \pm 0.12$ | $17.93 \pm 4.48$ | $18.13 \pm 4.37$ | $19.94 \pm 4.19$ | $\mathbf{20.2 \pm 4.23}$ |
| 60 | $1.02 \pm 0.02$ | $11.55 \pm 3.92$ | $12.91 \pm 3.88$ | $13.15 \pm 3.82$ | $\mathbf{14.31 \pm 4.26}$ |
| 70 | $1 \pm 0$ | $7.13 \pm 2.01$ | $8.42 \pm 2.25$ | $8.72 \pm 2.74$ | $\mathbf{11.82 \pm 3.47}$ |
| 80 | $1 \pm 0$ | $4.24 \pm 1.52$ | $5.06 \pm 1.98$ | $5.46 \pm 1.69$ | $\mathbf{8.24 \pm 2.92}$ |
| 90 | $1 \pm 0$ | $1 \pm 0$ | $1.12 \pm 0.12$ | $2.45 \pm 0.85$ | $\mathbf{3.17 \pm 1.58}$ |

Table 8 presents the result of our ablation study using VGG-7 and ResNet-18 models on the CIFAR-10 and CIFAR-100 datasets. Consistent with the trends observed in Table 7, both classification accuracy and the fault-robust regime improve as additional components of the proposed mechanism are incorporated. Notably, in contrast to Table 7, input fragmentation remains effective even at low fault ratios. This is attributed to the higher complexity of CIFAR-10 and CIFAR-100 samples, which necessitates dividing the input load even in the absence of a fault to better align with the usable learning capacity of the SNN.

## B.5. In-depth study

To show the changes in the fault mitigation ability of our mechanism according to the changes in the settings of the spiking neurons, encoding, and fragmentation, we apply different settings to the neurons, neural encoders, and fragmentation mechanism.

### B.5.1. NEURON SETTINGS

We change the neuron settings of various SNN models with our mechanism and compare their classification accuracy. The default neuron settings for our mechanism are Arc-tangent for the surrogate gradient function, 1.0 for the threshold, and hard reset to initialize the membrane potential. We apply a different surrogate gradient function, threshold value, and reset configuration to the neurons.

*Table 9.* Classification accuracy under different surrogate gradient, threshold, and reset settings across fault ratios for MLP, VGG-7, and ResNet-18.

| ACCURACY (%) WITH SURROGATE GRADIENTS | | | | | | | | |
|---|---|---|---|---|---|---|---|---|
| FAULT RATIO(%) | MLP (MNIST) | | MLP (FMNIST) | | VGG-7 (CIFAR-10) | | RESNET-18 (CIFAR-100) | |
| | ARC-TANGENT | SIGMOID | ARC-TANGENT | SIGMOID | ARC-TANGENT | SIGMOID | ARC-TANGENT | SIGMOID |
| 0 | 98.15 ± 0.84 | 98.04 ± 0.87 | 89.38 ± 1.13 | 89.11 ± 1.24 | 77.09 ± 1.27 | 76.84 ± 1.46 | 40.48 ± 3.71 | 40.03 ± 3.89 |
| 10 | 97.79 ± 0.91 | 97.62 ± 1.06 | 88.95 ± 1.39 | 88.61 ± 1.58 | 73.19 ± 1.69 | 72.75 ± 2.15 | 35.07 ± 4.18 | 34.34 ± 4.47 |
| 20 | 97.03 ± 1.37 | 96.89 ± 1.39 | 89.1 ± 1.52 | 88.83 ± 1.66 | 70.08 ± 4.52 | 69.86 ± 4.59 | 32.39 ± 5.55 | 31.81 ± 5.73 |
| 30 | 95.89 ± 1.45 | 95.51 ± 1.47 | 88.43 ± 2.05 | 87.98 ± 2.19 | 69.36 ± 5.2 | 69.01 ± 5.23 | 29.45 ± 5.6 | 28.78 ± 5.77 |
| 40 | 95.15 ± 1.66 | 94.63 ± 1.75 | 85.48 ± 4.83 | 84.71 ± 5.06 | 63.14 ± 5.96 | 62.38 ± 6.24 | 24.9 ± 5.91 | 23.76 ± 6.17 |
| 50 | 93.52 ± 2.18 | 93.12 ± 2.34 | 78.48 ± 5.79 | 77.42 ± 6.08 | 51.46 ± 6.74 | 50.29 ± 7.3 | 20.2 ± 4.23 | 18.68 ± 4.64 |
| 60 | 10.75 ± 0.95 | 9.8 ± 0 | 13.34 ± 2.11 | 12.97 ± 2.21 | 38.48 ± 5.13 | 37.34 ± 5.21 | 14.31 ± 4.26 | 13.22 ± 4.41 |
| 70 | 9.8 ± 0 | 9.8 ± 0 | 10 ± 0 | 10 ± 0 | 23.25 ± 4.35 | 21.98 ± 4.08 | 11.82 ± 3.47 | 10.84 ± 3.42 |
| 80 | 9.8 ± 0 | 9.8 ± 0 | 10 ± 0 | 10 ± 0 | 10 ± 0 | 10 ± 0 | 8.24 ± 2.92 | 7.61 ± 3.00 |
| 90 | 9.8 ± 0 | 9.8 ± 0 | 10 ± 0 | 10 ± 0 | 10 ± 0 | 10 ± 0 | 3.17 ± 1.38 | 2.89 ± 1.33 |

| ACCURACY (%) WITH THRESHOLD VALUES | | | | | | | | |
|---|---|---|---|---|---|---|---|---|
| FAULT RATIO(%) | MLP (MNIST) | | MLP (FMNIST) | | VGG-7 (CIFAR-10) | | RESNET-18 (CIFAR-100) | |
| | 1.0 | 2.0 | 1.0 | 2.0 | 1.0 | 2.0 | 1.0 | 2.0 |
| 0 | 98.15 ± 0.84 | 98.11 ± 0.82 | 89.38 ± 1.13 | 89.27 ± 1.20 | 77.09 ± 1.27 | 76.58 ± 1.5 | 40.48 ± 3.71 | 40.12 ± 3.86 |
| 10 | 97.79 ± 0.91 | 97.62 ± 0.95 | 88.95 ± 1.39 | 88.79 ± 1.50 | 73.19 ± 1.69 | 71.96 ± 1.96 | 35.07 ± 4.18 | 34.46 ± 4.36 |
| 20 | 97.03 ± 1.37 | 96.59 ± 1.44 | 89.1 ± 1.52 | 88.98 ± 1.64 | 70.08 ± 4.52 | 69.71 ± 4.56 | 32.39 ± 5.55 | 32.04 ± 5.64 |
| 30 | 95.89 ± 1.45 | 95.91 ± 1.46 | 88.43 ± 2.05 | 88.54 ± 2.18 | 69.36 ± 5.2 | 68.84 ± 5.35 | 29.45 ± 5.6 | 29.21 ± 5.81 |
| 40 | 95.15 ± 1.66 | 95.38 ± 1.54 | 85.48 ± 4.83 | 85.91 ± 4.98 | 63.14 ± 5.96 | 64.37 ± 5.79 | 24.9 ± 5.91 | 25.58 ± 6.04 |
| 50 | 93.52 ± 2.18 | 94.14 ± 2.05 | 78.48 ± 5.79 | 79.41 ± 6.09 | 51.46 ± 6.74 | 54.29 ± 6.62 | 20.2 ± 4.23 | 21.67 ± 4.48 |
| 60 | 10.75 ± 0.95 | 12.73 ± 1.24 | 13.34 ± 2.11 | 15.11 ± 3.92 | 38.48 ± 5.13 | 41.47 ± 5.28 | 14.31 ± 4.26 | 16.08 ± 4.57 |
| 70 | 9.8 ± 0 | 10.6 ± 0.8 | 10 ± 0 | 10 ± 0 | 23.25 ± 4.35 | 26.38 ± 4.59 | 11.82 ± 3.47 | 13.37 ± 3.79 |
| 80 | 9.8 ± 0 | 9.8 ± 0 | 10 ± 0 | 10 ± 0 | 10 ± 0 | 11.41 ± 1.41 | 8.24 ± 2.92 | 9.31 ± 3.46 |
| 90 | 9.8 ± 0 | 9.8 ± 0 | 10 ± 0 | 10 ± 0 | 10 ± 0 | 10 ± 0 | 3.17 ± 1.38 | 4.73 ± 2.68 |

| ACCURACY (%) WITH RESET SETTINGS | | | | | | | | |
|---|---|---|---|---|---|---|---|---|
| FAULT RATIO(%) | MLP (MNIST) | | MLP (FMNIST) | | VGG-7 (CIFAR-10) | | RESNET-18 (CIFAR-100) | |
| | HARD | SOFT | HARD | SOFT | HARD | SOFT | HARD | SOFT |
| 0 | 98.15 ± 0.84 | 97.62 ± 1.01 | 89.38 ± 1.13 | 88.59 ± 1.44 | 77.09 ± 1.27 | 75.68 ± 1.92 | 40.48 ± 3.71 | 39.30 ± 4.04 |
| 10 | 97.79 ± 0.91 | 96.95 ± 1.39 | 88.95 ± 1.39 | 88.01 ± 1.83 | 73.19 ± 1.69 | 72.02 ± 2.03 | 35.07 ± 4.18 | 33.98 ± 4.47 |
| 20 | 97.03 ± 1.37 | 96.18 ± 1.52 | 89.1 ± 1.52 | 88.13 ± 1.70 | 70.08 ± 4.52 | 68.83 ± 4.77 | 32.39 ± 5.55 | 31.37 ± 5.78 |
| 30 | 95.89 ± 1.45 | 94.73 ± 1.76 | 88.43 ± 2.05 | 87.24 ± 2.39 | 69.36 ± 5.2 | 68.11 ± 5.61 | 29.45 ± 5.6 | 28.50 ± 5.78 |
| 40 | 95.15 ± 1.66 | 93.91 ± 2.03 | 85.48 ± 4.83 | 84.23 ± 5.16 | 63.14 ± 5.96 | 61.87 ± 6.18 | 24.9 ± 5.91 | 24.02 ± 6.05 |
| 50 | 93.52 ± 2.18 | 92.69 ± 2.49 | 78.48 ± 5.79 | 77.69 ± 6.16 | 51.46 ± 6.74 | 50.75 ± 7.24 | 20.2 ± 4.23 | 19.41 ± 4.41 |
| 60 | 10.75 ± 0.95 | 9.8 ± 0 | 13.34 ± 2.11 | 11.54 ± 1.54 | 38.48 ± 5.13 | 36.93 ± 5.46 | 14.31 ± 4.26 | 13.21 ± 4.47 |
| 70 | 9.8 ± 0 | 9.8 ± 0 | 10 ± 0 | 10 ± 0 | 23.25 ± 4.35 | 21.67 ± 3.97 | 11.82 ± 3.47 | 10.99 ± 3.59 |
| 80 | 9.8 ± 0 | 9.8 ± 0 | 10 ± 0 | 10 ± 0 | 10 ± 0 | 10 ± 0 | 8.24 ± 2.92 | 7.67 ± 2.95 |
| 90 | 9.8 ± 0 | 9.8 ± 0 | 10 ± 0 | 10 ± 0 | 10 ± 0 | 10 ± 0 | 3.17 ± 1.38 | 2.48 ± 1.12 |

The experimental results in Table 9 demonstrate that the fault mitigation ability of our mechanism does not change substantially according to the different settings in the neurons. This point indicates that our mechanism can be adopted to various SNN models with different neuron settings without degradation of fault mitigation ability.

### B.5.2. ENCODING SETTINGS

We change the encoding scheme of various SNN models with our mechanism and compare their classification accuracy. The default encoding scheme for our mechanism is Poisson encoding, and we apply Time-to-First Spike (TTFS) encoding to SNN models for comparison.

*Table 10.* Classification accuracy under different encoding schemes across fault ratios for MLP, VGG-7, and ResNet-18.

| | ACCURACY (%) WITH DIFFERENT ENCODING SCHEMES | | | | | | | |
| --- | --- | --- | --- | --- | --- | --- | --- | --- |
| FAULT RATIO(%) | MLP (MNIST) | | MLP (FMNIST) | | VGG-7 (CIFAR-10) | | RESNET-18 (CIFAR-100) | |
| | POISSON | TTFS | POISSON | TTFS | POISSON | TTFS | POISSON | TTFS |
| 0 | $98.15 \pm 0.84$ | $98.02 \pm 0.93$ | $89.38 \pm 1.13$ | $88.97 \pm 1.28$ | $77.09 \pm 1.27$ | $76.54 \pm 1.51$ | $40.48 \pm 3.71$ | $39.27 \pm 4.02$ |
| 10 | $97.79 \pm 0.91$ | $97.46 \pm 1.15$ | $88.95 \pm 1.39$ | $88.42 \pm 1.59$ | $73.19 \pm 1.69$ | $72.82 \pm 1.78$ | $35.07 \pm 4.18$ | $33.49 \pm 4.49$ |
| 20 | $97.03 \pm 1.37$ | $96.81 \pm 1.31$ | $89.1 \pm 1.52$ | $88.36 \pm 1.78$ | $70.08 \pm 4.52$ | $68.96 \pm 4.68$ | $32.39 \pm 5.55$ | $30.67 \pm 5.96$ |
| 30 | $95.89 \pm 1.45$ | $94.96 \pm 1.58$ | $88.43 \pm 2.05$ | $87.57 \pm 2.29$ | $69.36 \pm 5.2$ | $67.14 \pm 5.7$ | $29.45 \pm 5.6$ | $27.54 \pm 6.03$ |
| 40 | $95.15 \pm 1.66$ | $94.38 \pm 1.89$ | $85.48 \pm 4.83$ | $84.21 \pm 5.16$ | $63.14 \pm 5.96$ | $61.56 \pm 6.36$ | $24.9 \pm 5.91$ | $22.87 \pm 6.31$ |
| 50 | $93.52 \pm 2.18$ | $92.19 \pm 2.37$ | $78.48 \pm 5.79$ | $76.91 \pm 6.11$ | $51.46 \pm 6.74$ | $49.89 \pm 7.59$ | $20.2 \pm 4.23$ | $18.37 \pm 4.56$ |
| 60 | $10.75 \pm 0.95$ | $9.8 \pm 0$ | $13.34 \pm 2.11$ | $11.08 \pm 1.08$ | $38.48 \pm 5.13$ | $37.11 \pm 5.37$ | $14.31 \pm 4.26$ | $12.77 \pm 4.63$ |
| 70 | $9.8 \pm 0$ | $9.8 \pm 0$ | $10 \pm 0$ | $10 \pm 0$ | $23.25 \pm 4.35$ | $21.93 \pm 4.49$ | $11.82 \pm 3.47$ | $10.49 \pm 3.81$ |
| 80 | $9.8 \pm 0$ | $9.8 \pm 0$ | $10 \pm 0$ | $10 \pm 0$ | $10 \pm 0$ | $10 \pm 0$ | $8.24 \pm 2.92$ | $7.17 \pm 3.21$ |
| 90 | $9.8 \pm 0$ | $9.8 \pm 0$ | $10 \pm 0$ | $10 \pm 0$ | $10 \pm 0$ | $10 \pm 0$ | $3.17 \pm 1.38$ | $2.61 \pm 1.47$ |

Table 10 shows that our mechanism improves the fault tolerance of various SNN models under different encoding schemes.

### B.5.3. FRAGMENTATION SETTINGS

We change the fragmentation settings of various SNN models with our mechanism and compare their classification accuracy. The default fragmentation settings for our mechanism are linear line fragmentation and candidate set $\{2, 4, 8\}$. We apply nonlinear (quadratic function) fragmentation lines and a candidate set $\{2, 4, 8, 16\}$.

*Table 11.* Classification accuracy under different fragmentation lines and candidate sets across fault ratios for MLP, VGG-7, and ResNet-18.

| | ACCURACY (%) WITH DIFFERENT FRAGMENTATION LINE SETTING | | | | | | | |
| --- | --- | --- | --- | --- | --- | --- | --- | --- |
| FAULT RATIO(%) | MLP (MNIST) | | MLP (FMNIST) | | VGG-7 (CIFAR-10) | | RESNET-18 (CIFAR-100) | |
| | LINEAR | NONLINEAR | LINEAR | NONLINEAR | LINEAR | NONLINEAR | LINEAR | NONLINEAR |
| 0 | $98.15 \pm 0.84$ | $98.23 \pm 0.8$ | $89.38 \pm 1.13$ | $89.66 \pm 1.08$ | $77.09 \pm 1.27$ | $77.68 \pm 1.16$ | $40.48 \pm 3.71$ | $40.89 \pm 3.59$ |
| 10 | $97.79 \pm 0.91$ | $97.83 \pm 0.92$ | $88.95 \pm 1.39$ | $89.11 \pm 1.37$ | $73.19 \pm 1.69$ | $73.53 \pm 1.62$ | $35.07 \pm 4.18$ | $35.34 \pm 4.12$ |
| 20 | $97.03 \pm 1.37$ | $97.17 \pm 1.34$ | $89.1 \pm 1.52$ | $89.27 \pm 1.49$ | $70.08 \pm 4.52$ | $70.29 \pm 4.49$ | $32.39 \pm 5.55$ | $32.58 \pm 5.47$ |
| 30 | $95.89 \pm 1.45$ | $96.09 \pm 1.51$ | $88.43 \pm 2.05$ | $88.68 \pm 2.11$ | $69.36 \pm 5.2$ | $69.71 \pm 5.24$ | $29.45 \pm 5.6$ | $29.73 \pm 5.66$ |
| 40 | $95.15 \pm 1.66$ | $95.34 \pm 1.62$ | $85.48 \pm 4.83$ | $85.70 \pm 4.79$ | $63.14 \pm 5.96$ | $63.42 \pm 5.92$ | $24.9 \pm 5.91$ | $25.13 \pm 5.94$ |
| 50 | $93.52 \pm 2.18$ | $92.68 \pm 2.31$ | $78.48 \pm 5.79$ | $78.26 \pm 5.82$ | $51.46 \pm 6.74$ | $51.88 \pm 6.69$ | $20.2 \pm 4.23$ | $20.25 \pm 4.39$ |
| 60 | $10.75 \pm 0.95$ | $9.8 \pm 0$ | $13.34 \pm 2.11$ | $12.29 \pm 1.76$ | $38.48 \pm 5.13$ | $37.31 \pm 5.37$ | $14.31 \pm 4.26$ | $13.64 \pm 4.51$ |
| 70 | $9.8 \pm 0$ | $9.8 \pm 0$ | $10 \pm 0$ | $10 \pm 0$ | $23.25 \pm 4.35$ | $22.12 \pm 4.41$ | $11.82 \pm 3.47$ | $10.91 \pm 3.73$ |
| 80 | $9.8 \pm 0$ | $9.8 \pm 0$ | $10 \pm 0$ | $10 \pm 0$ | $10 \pm 0$ | $10 \pm 0$ | $8.24 \pm 2.92$ | $7.32 \pm 3.16$ |
| 90 | $9.8 \pm 0$ | $9.8 \pm 0$ | $10 \pm 0$ | $10 \pm 0$ | $10 \pm 0$ | $10 \pm 0$ | $3.17 \pm 1.38$ | $2.83 \pm 1.83$ |

| | ACCURACY (%) WITH DIFFERENT FRAGMENTATION CANDIDATE SETS | | | | | | | |
| --- | --- | --- | --- | --- | --- | --- | --- | --- |
| FAULT RATIO(%) | MLP (MNIST) | | MLP (FMNIST) | | VGG-7 (CIFAR-10) | | RESNET-18 (CIFAR-100) | |
| | $\{2, 4, 8\}$ | $\{2, 4, 8, 16\}$ | $\{2, 4, 8\}$ | $\{2, 4, 8, 16\}$ | $\{2, 4, 8\}$ | $\{2, 4, 8, 16\}$ | $\{2, 4, 8\}$ | $\{2, 4, 8, 16\}$ |
| 0 | $98.15 \pm 0.84$ | $97.42 \pm 0.99$ | $89.38 \pm 1.13$ | $88.71 \pm 1.28$ | $77.09 \pm 1.27$ | $77.32 \pm 1.21$ | $40.48 \pm 3.71$ | $40.96 \pm 3.58$ |
| 10 | $97.79 \pm 0.91$ | $97.06 \pm 1.04$ | $88.95 \pm 1.39$ | $88.29 \pm 1.56$ | $73.19 \pm 1.69$ | $73.95 \pm 1.47$ | $35.07 \pm 4.18$ | $35.88 \pm 4.12$ |
| 20 | $97.03 \pm 1.37$ | $96.89 \pm 1.57$ | $89.1 \pm 1.52$ | $88.88 \pm 1.77$ | $70.08 \pm 4.52$ | $71.12 \pm 3.99$ | $32.39 \pm 5.55$ | $33.58 \pm 5.21$ |
| 30 | $95.89 \pm 1.45$ | $95.13 \pm 1.64$ | $88.43 \pm 2.05$ | $87.74 \pm 2.33$ | $69.36 \pm 5.2$ | $69.8 \pm 5.18$ | $29.45 \pm 5.6$ | $30.88 \pm 5.33$ |
| 40 | $95.15 \pm 1.66$ | $94.24 \pm 1.89$ | $85.48 \pm 4.83$ | $84.59 \pm 5.09$ | $63.14 \pm 5.96$ | $64.28 \pm 5.7$ | $24.9 \pm 5.91$ | $26.12 \pm 5.65$ |
| 50 | $93.52 \pm 2.18$ | $92.68 \pm 2.51$ | $78.48 \pm 5.79$ | $77.63 \pm 6.18$ | $51.46 \pm 6.74$ | $53.52 \pm 6.57$ | $20.2 \pm 4.23$ | $21.48 \pm 4.18$ |
| 60 | $10.75 \pm 0.95$ | $9.8 \pm 0$ | $13.34 \pm 2.11$ | $11.92 \pm 1.86$ | $38.48 \pm 5.13$ | $40.76 \pm 5.32$ | $14.31 \pm 4.26$ | $15.02 \pm 4.37$ |
| 70 | $9.8 \pm 0$ | $9.8 \pm 0$ | $10 \pm 0$ | $10 \pm 0$ | $23.25 \pm 4.35$ | $25.91 \pm 4.73$ | $11.82 \pm 3.47$ | $12.54 \pm 3.61$ |
| 80 | $9.8 \pm 0$ | $9.8 \pm 0$ | $10 \pm 0$ | $10 \pm 0$ | $10 \pm 0$ | $13.16 \pm 2.94$ | $8.24 \pm 2.92$ | $8.81 \pm 3.01$ |
| 90 | $9.8 \pm 0$ | $9.8 \pm 0$ | $10 \pm 0$ | $10 \pm 0$ | $10 \pm 0$ | $10 \pm 0$ | $3.17 \pm 1.38$ | $4.69 \pm 2.05$ |

As presented in Table 11, nonlinear fragmentation usually achieves better fault mitigation ability than linear fragmentation. However, the nonlinear fragmentation requires 250 epochs to reach the top accuracy, while the linear fragmentation requires only 50 epochs for the top accuracy. With more fragments, our mechanism exhibits better fault mitigation ability when we use complex datasets. This is because fragments from complex data samples have more energy than fragments from simple data samples, leading our mechanism to make them as small as possible to reduce the energy per fragment.

## B.6. Intermediate evidence of our mechanism's fault mitigation ability

To demonstrate that the proposed mechanism prevents the membrane potential from escaping the surrogate corridor, thereby helping SNNs avoid gradient vanishing, we show the average energy, potential deviation ($\mathbb{E}[|u_t - \vartheta|]$), probability of gradient vanishing ($Pr(|\nabla_w \mathcal{L}|) < \sigma,\ \sigma = 0.0001$), and L1 norm of gradients.

### B.6.1. AVERAGE ENERGY OF EACH FRAGMENT CREATED BY OUR MECHANISM

We measure the average energy (L2 norm) of each input fragment produced by our fragmentation mechanism to assess the effect of the fragmentation strategy and the balancing loss ($L_{bal}$) on the distribution of the energies (Gonçalves et al., 1997).

*Table 12.* Average of each fragment's energy created by our mechanism during inference in the SAF-injected MLP model with 50% fault ratio.

|  | FRAG 1 | FRAG 2 |
|---|---|---|
| MNIST | $19.62 \pm 0.97$ | $20.08 \pm 1.13$ |
| FMNIST | $19.01 \pm 1.25$ | $20.39 \pm 1.62$ |

Table 12 shows the average fragment energy for MLP models under SAFs on MNIST and FMNIST. In MLP models, the proposed mechanism converges to using two fragments. This outcome is consistent with the relatively low input energy of MNIST and FMNIST samples, for which a small number of fragments is sufficient to distribute energy and mitigate surrogate-gradient corridor escape. As intended, the energy across fragments is well balanced.

*Table 13.* Average of each fragment's energy created by our mechanism during inference in SAF-injected VGG-7 and ResNet-18 models with 50% fault ratio.

|  | FRAG 1 | FRAG 2 | FRAG 3 | FRAG 4 | FRAG 5 | FRAG 6 | FRAG 7 | FRAG 8 |
|---|---|---|---|---|---|---|---|---|
| CIFAR-10 (VGG-7) | $31.09 \pm 4.65$ | $32.18 \pm 4.72$ | $30.57 \pm 5.12$ | $29.59 \pm 4.96$ | $31.26 \pm 5.21$ | $28.99 \pm 5.63$ | $27.87 \pm 4.85$ | $27.3 \pm 5.14$ |
| CIFAR-100 (RESNET-18) | $33.26 \pm 4.8$ | $30.2 \pm 5.13$ | $31.89 \pm 4.93$ | $28.04 \pm 4.75$ | $27.99 \pm 6.09$ | $26.23 \pm 5.32$ | $29.16 \pm 4.34$ | $28.77 \pm 4.81$ |

Table 13 presents the average fragment energy for VGG-7 and ResNet-18 models under SAFs on CIFAR-10 and CIFAR-100. In contrast to MNIST and FMNIST, the proposed mechanism converges to using eight fragments. This behavior is consistent with the higher input complexity and larger input energy of CIFAR-10 and CIFAR-100 samples, for which reducing the energy per fragment requires a larger number of fragments. Similar to the MLP case, the proposed mechanism effectively equalizes the energy across fragments.

These results using MLP/VGG/ResNet indicate that the proposed mechanism yields well-balanced fragment energies at inference time. This balance is consistent with the intended role of the fragmentation strategy, as preventing any single fragment from carrying disproportionately large energy limits sharp increases in membrane potential when fragments are processed by the SNN.

### B.6.2. POTENTIAL DEVIATION, GRADIENT VANISHING PROBABILITY, AND L1 NORM OF GRADIENTS

We present evidence that our mechanism suppresses excessive increases in the membrane potential, alleviates gradient vanishing, and enhances fault tolerance.

*Table 14.* The absolute deviation of the membrane potential from the threshold, the probability that the absolute value of the gradient is below 0.0001 across all neurons, and the L1 norm of the gradients' absolute value across all neurons with a 50% fault ratio of SAF in MLP, VGG-7, and ResNet-18 using MNIST, CIFAR-10, and CIFAR-100.

|  | BASELINE (NO FAULTS) | BASELINE | ECOC | SOFTSNN | ROUTING | ASTROCYTE | FALVOLT | LIFA | PROPOSED |
|---|---|---|---|---|---|---|---|---|---|
| | | | | MLP (MNIST) | | | | | |
| DEVIATION | $0.25 \pm 0.05$ | $80.18 \pm 6.48$ | $68.23 \pm 7.15$ | $79.64 \pm 6.86$ | $58.81 \pm 6.23$ | $72.34 \pm 6.88$ | $64.54 \pm 5.51$ | $71.05 \pm 6.72$ | $15.36 \pm 3.65$ |
| PROBABILITY | $9.2 \times 10^{-3}$ | $9.9 \times 10^{-1}$ | $9.6 \times 10^{-1}$ | $9.8 \times 10^{-1}$ | $9.7 \times 10^{-1}$ | $9.7 \times 10^{-1}$ | $9.6 \times 10^{-1}$ | $9.7 \times 10^{-1}$ | $1.9 \times 10^{-1}$ |
| L1 NORM | $5.89 \times 10^{-3}$ | $5.21 \times 10^{-8}$ | $1.86 \times 10^{-7}$ | $5.43 \times 10^{-8}$ | $8.47 \times 10^{-7}$ | $8.48 \times 10^{-8}$ | $7.61 \times 10^{-7}$ | $9.07 \times 10^{-8}$ | $1.75 \times 10^{-3}$ |
| | | | | VGG-7 (CIFAR-10) | | | | | |
| DEVIATION | $0.81 \pm 0.23$ | $30.94 \pm 5.27$ | $8.44 \pm 2.61$ | $9.78 \pm 3.01$ | $24.57 \pm 5.15$ | $29.01 \pm 4.73$ | $26.93 \pm 5.31$ | $29.42 \pm 4.38$ | $6.01 \pm 1.98$ |
| PROBABILITY | $1.9 \times 10^{-2}$ | $8.7 \times 10^{-1}$ | $3.6 \times 10^{-1}$ | $3.4 \times 10^{-1}$ | $8.5 \times 10^{-1}$ | $8.7 \times 10^{-1}$ | $8.6 \times 10^{-1}$ | $8.7 \times 10^{-1}$ | $2.2 \times 10^{-1}$ |
| L1 NORM | $1.08 \times 10^{-3}$ | $6.39 \times 10^{-7}$ | $4.84 \times 10^{-4}$ | $3.19 \times 10^{-4}$ | $7.72 \times 10^{-7}$ | $6.45 \times 10^{-7}$ | $7.28 \times 10^{-7}$ | $6.51 \times 10^{-7}$ | $8.57 \times 10^{-4}$ |
| | | | | RESNET-18 (CIFAR-100) | | | | | |
| DEVIATION | $0.39 \pm 0.11$ | $2.41 \pm 0.38$ | $2.13 \pm 0.41$ | $2.36 \pm 0.37$ | $2.28 \pm 0.38$ | $2.39 \pm 0.39$ | $2.31 \pm 0.34$ | $2.38 \pm 0.36$ | $0.96 \pm 0.52$ |
| PROBABILITY | $1.1 \times 10^{-2}$ | $9.6 \times 10^{-1}$ | $9.2 \times 10^{-1}$ | $9.6 \times 10^{-1}$ | $9.6 \times 10^{-1}$ | $9.6 \times 10^{-1}$ | $9.3 \times 10^{-1}$ | $9.6 \times 10^{-1}$ | $3.1 \times 10^{-1}$ |
| L1 NORM | $9.61 \times 10^{-4}$ | $8.86 \times 10^{-9}$ | $7.65 \times 10^{-9}$ | $8.71 \times 10^{-9}$ | $8.64 \times 10^{-9}$ | $8.83 \times 10^{-9}$ | $8.49 \times 10^{-9}$ | $8.72 \times 10^{-9}$ | $1.45 \times 10^{-4}$ |

As shown in Table 14, our mechanism successfully keeps the potential within the surrogate corridor, preventing gradient vanishing.

## B.7. FPGA implementations

We deploy SNN models with our mechanism and benchmarks on the real FPGA device (inference only) and Vivado-based emulator (training and inference) to evaluate the fault mitigation of our mechanism in FPGA environments.

### B.7.1. EVALUATIONS WITH REAL FPGA HARDWARE

We implement the MLP-based SNN model on a real FPGA device (AMD Virtex UltraScale+ HBM VU47P) deployed on an Amazon F2 instance using SpikerPlus, a library that converts Python-based SNN descriptions into VHSIC Hardware Description Language (VHDL) (Carpegna et al., 2024). We choose an FPGA platform for SNN implementation because FPGAs are widely used for developing hardware-based SNN systems (Karamimanesh et al., 2025). The FPGA circuit is synthesized using Xilinx Vivado and the Amazon FPGA Image (AFI) workflow, which is commonly adopted for managing FPGAs on Amazon F2 instances. The SNN models are implemented on the FPGA, while the proposed mechanism is executed on an additional control processor (e.g., a computer that commands FPGA devices with PCIe sockets) connected to the FPGA. We set the bit-widths of the membrane potential and synaptic weights to 8 and 6 bits, respectively, and use 32-bit floating-point precision for the input data samples. This configuration is selected, referring to real hardware settings reported in previous work (Inc., 2025). All other settings follow those described in Subsection 6.1.

*Table 15.* The classification accuracy of MLP-based SNN on FPGA device in a 95% confidence interval using MNIST, FMNIST, UCI-HAR, and AudioMNIST under SAFs.

| FAULT RATIO(%) | BASELINE | ECOC | SOFTSNN | ROUTING | ASTROCYTE | FALVOLT | LIFA | PROPOSED |
|---|---|---|---|---|---|---|---|---|
| ACCURACY (%) WITH HARDWARE-IMPLEMENTED MLP (MNIST) | | | | | | | | |
| 0 | $94.82 \pm 0.52$ | $94.27 \pm 0.56$ | $94.47 \pm 0.49$ | $94.21 \pm 0.71$ | $94.15 \pm 0.53$ | $94.79 \pm 0.83$ | $94.55 \pm 0.93$ | $\mathbf{95.02 \pm 0.49}$ |
| 10 | $93.36 \pm 1.34$ | $93.44 \pm 2.07$ | $93.39 \pm 1.64$ | $93.47 \pm 1.71$ | $93.4 \pm 1.52$ | $93.96 \pm 1.68$ | $93.58 \pm 1.87$ | $\mathbf{94.86 \pm 0.72}$ |
| 20 | $90.49 \pm 2.67$ | $90.62 \pm 2.81$ | $11.35 \pm 0$ | $9.8 \pm 0$ | $22.31 \pm 3.15$ | $9.8 \pm 0$ | $23.42 \pm 4.48$ | $\mathbf{93.57 \pm 1.65}$ |
| 30 | $9.8 \pm 0$ | $11.35 \pm 0$ | $9.8 \pm 0$ | $9.8 \pm 0$ | $9.8 \pm 0$ | $9.8 \pm 0$ | $9.8 \pm 0$ | $\mathbf{92.96 \pm 2.74}$ |
| 40 | $9.8 \pm 0$ | $11.35 \pm 0$ | $9.8 \pm 0$ | $9.8 \pm 0$ | $9.8 \pm 0$ | $9.8 \pm 0$ | $9.8 \pm 0$ | $\mathbf{92.02 \pm 4.18}$ |
| 50 | $9.8 \pm 0$ | $10.32 \pm 0$ | $9.8 \pm 0$ | $9.8 \pm 0$ | $9.8 \pm 0$ | $9.8 \pm 0$ | $9.8 \pm 0$ | $\mathbf{89.34 \pm 4.51}$ |
| 60 | $9.8 \pm 0$ | $9.8 \pm 0$ | $9.8 \pm 0$ | $9.8 \pm 0$ | $9.8 \pm 0$ | $9.8 \pm 0$ | $9.8 \pm 0$ | $\mathbf{11.7 \pm 1.63}$ |
| 70 | $9.8 \pm 0$ | $9.8 \pm 0$ | $9.8 \pm 0$ | $9.8 \pm 0$ | $9.8 \pm 0$ | $9.8 \pm 0$ | $9.8 \pm 0$ | $9.8 \pm 0$ |
| 80 | $9.8 \pm 0$ | $9.8 \pm 0$ | $9.8 \pm 0$ | $9.8 \pm 0$ | $9.8 \pm 0$ | $9.8 \pm 0$ | $9.8 \pm 0$ | $9.8 \pm 0$ |
| 90 | $9.8 \pm 0$ | $9.8 \pm 0$ | $9.8 \pm 0$ | $9.8 \pm 0$ | $9.8 \pm 0$ | $9.8 \pm 0$ | $9.8 \pm 0$ | $9.8 \pm 0$ |
| ACCURACY (%) WITH HARDWARE-IMPLEMENTED MLP (FMNIST) | | | | | | | | |
| 0 | $85.16 \pm 1.47$ | $84.64 \pm 1.89$ | $84.72 \pm 2.29$ | $84.73 \pm 1.72$ | $84.23 \pm 1.96$ | $84.38 \pm 1.77$ | $84.63 \pm 1.69$ | $\mathbf{87.52 \pm 1.3}$ |
| 10 | $84.75 \pm 2.16$ | $84.82 \pm 2.29$ | $83.91 \pm 2.52$ | $84.09 \pm 2.48$ | $83.78 \pm 2.38$ | $84.21 \pm 2.48$ | $83.94 \pm 2.27$ | $\mathbf{85.24 \pm 2.46}$ |
| 20 | $10 \pm 0$ | $10 \pm 0$ | $10 \pm 0$ | $10 \pm 0$ | $10 \pm 0$ | $81.23 \pm 3.48$ | $10 \pm 0$ | $\mathbf{84.79 \pm 3.11}$ |
| 30 | $10 \pm 0$ | $10 \pm 0$ | $10 \pm 0$ | $10 \pm 0$ | $10 \pm 0$ | $10 \pm 0$ | $10 \pm 0$ | $\mathbf{84.01 \pm 3.7}$ |
| 40 | $10 \pm 0$ | $10 \pm 0$ | $10 \pm 0$ | $10 \pm 0$ | $10 \pm 0$ | $10 \pm 0$ | $10 \pm 0$ | $\mathbf{82.47 \pm 4.29}$ |
| 50 | $10 \pm 0$ | $10 \pm 0$ | $10 \pm 0$ | $10 \pm 0$ | $10 \pm 0$ | $10 \pm 0$ | $10 \pm 0$ | $\mathbf{36.17 \pm 5.55}$ |
| 60 | $10 \pm 0$ | $10 \pm 0$ | $10 \pm 0$ | $10 \pm 0$ | $10 \pm 0$ | $10 \pm 0$ | $10 \pm 0$ | $\mathbf{23.91 \pm 4.86}$ |
| 70 | $10 \pm 0$ | $10 \pm 0$ | $10 \pm 0$ | $10 \pm 0$ | $10 \pm 0$ | $10 \pm 0$ | $10 \pm 0$ | $10 \pm 0$ |
| 80 | $10 \pm 0$ | $10 \pm 0$ | $10 \pm 0$ | $10 \pm 0$ | $10 \pm 0$ | $10 \pm 0$ | $10 \pm 0$ | $10 \pm 0$ |
| 90 | $10 \pm 0$ | $10 \pm 0$ | $10 \pm 0$ | $10 \pm 0$ | $10 \pm 0$ | $10 \pm 0$ | $10 \pm 0$ | $10 \pm 0$ |
| ACCURACY (%) WITH HARDWARE-IMPLEMENTED MLP (UCI-HAR) | | | | | | | | |
| 0 | $64.24 \pm 4.33$ | $66.38 \pm 4.15$ | $65.56 \pm 4.38$ | $64.21 \pm 4.24$ | $64.33 \pm 4.19$ | $64.66 \pm 4.32$ | $64.05 \pm 4.38$ | $\mathbf{67.39 \pm 4.07}$ |
| 10 | $30.81 \pm 4.29$ | $34.74 \pm 4.45$ | $31.08 \pm 4.61$ | $30.15 \pm 4.36$ | $24.19 \pm 4.34$ | $32.85 \pm 4.61$ | $25.26 \pm 4.54$ | $\mathbf{61.96 \pm 4.69}$ |
| 20 | $24.13 \pm 4.82$ | $25.49 \pm 4.77$ | $25.37 \pm 4.57$ | $24.62 \pm 4.92$ | $22.62 \pm 4.87$ | $23.91 \pm 5.18$ | $22.91 \pm 3.99$ | $\mathbf{54.73 \pm 5.11}$ |
| 30 | $17.1 \pm 0$ | $17.1 \pm 0$ | $17.1 \pm 0$ | $17.1 \pm 0$ | $17.1 \pm 0$ | $16.93 \pm 0$ | $17.1 \pm 0$ | $\mathbf{49.61 \pm 4.83}$ |
| 40 | $17.1 \pm 0$ | $17.1 \pm 0$ | $17.1 \pm 0$ | $17.1 \pm 0$ | $17.1 \pm 0$ | $16.93 \pm 0$ | $17.1 \pm 0$ | $\mathbf{48.15 \pm 4.56}$ |
| 50 | $17.1 \pm 0$ | $17.1 \pm 0$ | $17.1 \pm 0$ | $17.1 \pm 0$ | $17.1 \pm 0$ | $16.93 \pm 0$ | $17.1 \pm 0$ | $\mathbf{43.86 \pm 4.42}$ |
| 60 | $17.1 \pm 0$ | $17.1 \pm 0$ | $17.1 \pm 0$ | $17.1 \pm 0$ | $17.1 \pm 0$ | $16.93 \pm 0$ | $17.1 \pm 0$ | $\mathbf{29.14 \pm 4.18}$ |
| 70 | $17.1 \pm 0$ | $17.1 \pm 0$ | $17.1 \pm 0$ | $17.1 \pm 0$ | $17.1 \pm 0$ | $16.93 \pm 0$ | $17.1 \pm 0$ | $17.1 \pm 0$ |
| 80 | $17.1 \pm 0$ | $17.1 \pm 0$ | $17.1 \pm 0$ | $17.1 \pm 0$ | $17.1 \pm 0$ | $16.93 \pm 0$ | $17.1 \pm 0$ | $17.1 \pm 0$ |
| 90 | $17.1 \pm 0$ | $17.1 \pm 0$ | $17.1 \pm 0$ | $17.1 \pm 0$ | $17.1 \pm 0$ | $16.93 \pm 0$ | $17.1 \pm 0$ | $17.1 \pm 0$ |
| ACCURACY (%) WITH HARDWARE-IMPLEMENTED MLP (AUDIOMNIST) | | | | | | | | |
| 0 | $93.86 \pm 1.14$ | $94.12 \pm 1.29$ | $93.72 \pm 1.34$ | $93.89 \pm 1.01$ | $93.79 \pm 1.24$ | $93.48 \pm 1.35$ | $96.75 \pm 1.27$ | $\mathbf{94.49 \pm 1.25}$ |
| 10 | $92.51 \pm 1.07$ | $92.69 \pm 1.35$ | $92.97 \pm 1.62$ | $92.67 \pm 1.49$ | $91.69 \pm 1.57$ | $92.68 \pm 1.59$ | $92.09 \pm 1.76$ | $\mathbf{93.07 \pm 1.67}$ |
| 20 | $90.26 \pm 2.64$ | $91.35 \pm 2.42$ | $91.19 \pm 2.57$ | $90.58 \pm 2.93$ | $87.48 \pm 3.86$ | $90.84 \pm 2.97$ | $88.87 \pm 4.15$ | $\mathbf{91.98 \pm 2.71}$ |
| 30 | $89.72 \pm 3.47$ | $90.78 \pm 3.67$ | $88.26 \pm 4.14$ | $89.96 \pm 3.89$ | $54.19 \pm 6.12$ | $89.89 \pm 3.68$ | $57.93 \pm 6.09$ | $\mathbf{91.24 \pm 3.74}$ |
| 40 | $84.83 \pm 3.86$ | $88.54 \pm 4.04$ | $63.45 \pm 7.02$ | $87.27 \pm 4.26$ | $48.23 \pm 5.98$ | $86.28 \pm 4.12$ | $54.19 \pm 7.28$ | $\mathbf{89.12 \pm 3.96}$ |
| 50 | $83.39 \pm 4.24$ | $85.13 \pm 4.71$ | $51.98 \pm 7.68$ | $84.25 \pm 4.59$ | $26.92 \pm 4.73$ | $83.51 \pm 4.53$ | $33.38 \pm 5.41$ | $\mathbf{88.54 \pm 4.28}$ |
| 60 | $52.05 \pm 5.69$ | $64.62 \pm 5.23$ | $37.82 \pm 6.15$ | $65.63 \pm 5.87$ | $10 \pm 0$ | $62.97 \pm 5.42$ | $14.37 \pm 3.93$ | $\mathbf{77.25 \pm 5.79}$ |
| 70 | $16.29 \pm 4.31$ | $27.44 \pm 5.39$ | $10 \pm 0$ | $23.76 \pm 4.39$ | $10 \pm 0$ | $30.29 \pm 4.84$ | $10 \pm 0$ | $\mathbf{42.08 \pm 6.33}$ |
| 80 | $10 \pm 0$ | $17.96 \pm 4.75$ | $10 \pm 0$ | $13.02 \pm 2.88$ | $10 \pm 0$ | $14.02 \pm 4.32$ | $10 \pm 0$ | $\mathbf{25.06 \pm 5.19}$ |
| 90 | $10 \pm 0$ | $11.31 \pm 1.31$ | $10 \pm 0$ | $10 \pm 0$ | $10 \pm 0$ | $10 \pm 0$ | $10 \pm 0$ | $\mathbf{15.28 \pm 3.94}$ |

Table 15 presents the classification accuracy of the MLP model with baseline methods, benchmark methods, and the proposed mechanism under SAFs on MNIST, FMNIST, UCI-HAR, and AudioMNIST, with all models implemented on FPGA-based hardware. Across all datasets, the model employing our mechanism consistently achieves higher classification accuracy than the baseline and benchmark methods under the same hardware settings.

We also observe that the overall classification accuracy on the FPGA is lower than that obtained using the SpikingJelly software framework. This degradation is consistent with the reduced numerical precision of FPGA implementations, where membrane potentials and synaptic weights are represented using limited bit widths. Such quantization effects introduce distortions in trained synaptic weights and neuronal state variables, leading to a performance gap relative to high-precision software execution.

We implement VGG-7, 11, and 15 models using the SyncNN framework and evaluate the classification accuracy of the benchmark methods and the proposed mechanism under SAFs (Panchapakesan et al., 2021). The bit-width of synaptic weights is set to eight bits. All other SyncNN-related configurations follow those described in the original SyncNN paper (Panchapakesan et al., 2021). For the FPGA-based VGG models, we adopt the same SNN model settings as those described in Subsection 6.1.

*Table 16.* The classification accuracy of VGG-7, 11, and 15 SNN models on a real FPGA device in a 95% confidence interval using CIFAR-10 under SAFs.

| Fault ratio(%) | Baseline | ECOC | SoftSNN | Routing | Astrocyte | FalVolt | LIFA | Proposed |
|---|---|---|---|---|---|---|---|---|
| | Accuracy (%) with Hardware-implemented VGG-7 | | | | | | | |
| 0 | $67.61 \pm 4.29$ | $63.45 \pm 4.56$ | $65.64 \pm 4.74$ | $67.29 \pm 4.8$ | $66.31 \pm 4.32$ | $67.03 \pm 4.17$ | $66.17 \pm 4.63$ | $\mathbf{72.68 \pm 4.38}$ |
| 10 | $60.36 \pm 4.83$ | $52.83 \pm 4.78$ | $42.78 \pm 4.69$ | $60.61 \pm 4.51$ | $55.63 \pm 4.47$ | $61.31 \pm 4.95$ | $56.57 \pm 5.12$ | $\mathbf{68.79 \pm 4.98}$ |
| 20 | $52.24 \pm 4.97$ | $51.92 \pm 4.94$ | $40.26 \pm 5.14$ | $50.21 \pm 3.39$ | $45.43 \pm 4.26$ | $53.63 \pm 4.07$ | $43.4 \pm 3.98$ | $\mathbf{65.65 \pm 5.45}$ |
| 30 | $48.13 \pm 5.19$ | $49.88 \pm 5.07$ | $39.97 \pm 5.29$ | $39.09 \pm 4.92$ | $29.47 \pm 4.15$ | $49.52 \pm 4.02$ | $33.65 \pm 0$ | $\mathbf{64.71 \pm 5.44}$ |
| 40 | $36.08 \pm 4.25$ | $33.17 \pm 4.29$ | $35.39 \pm 4.08$ | $34.89 \pm 4.31$ | $12.74 \pm 1.98$ | $14.82 \pm 3.81$ | $10 \pm 0$ | $\mathbf{59.04 \pm 5.07}$ |
| 50 | $16.73 \pm 3.46$ | $28.92 \pm 3.46$ | $29.91 \pm 3.88$ | $14.65 \pm 3.74$ | $10 \pm 0$ | $10 \pm 0$ | $10 \pm 0$ | $\mathbf{46.98 \pm 4.79}$ |
| 60 | $10 \pm 0$ | $10 \pm 0$ | $24.01 \pm 3.81$ | $10 \pm 0$ | $10 \pm 0$ | $10 \pm 0$ | $10 \pm 0$ | $\mathbf{34.3 \pm 4.42}$ |
| 70 | $10 \pm 0$ | $10 \pm 0$ | $10 \pm 0$ | $10 \pm 0$ | $10 \pm 0$ | $10 \pm 0$ | $10 \pm 0$ | $\mathbf{18.87 \pm 2.66}$ |
| 80 | $10 \pm 0$ | $10 \pm 0$ | $10 \pm 0$ | $10 \pm 0$ | $10 \pm 0$ | $10 \pm 0$ | $10 \pm 0$ | $10 \pm 0$ |
| 90 | $10 \pm 0$ | $10 \pm 0$ | $10 \pm 0$ | $10 \pm 0$ | $10 \pm 0$ | $10 \pm 0$ | $10 \pm 0$ | $10 \pm 0$ |
| | Accuracy (%) with Hardware-implemented VGG-11 | | | | | | | |
| 0 | $68.77 \pm 4.76$ | $64.27 \pm 4.31$ | $68.17 \pm 4.25$ | $67.53 \pm 4.24$ | $67.26 \pm 4.62$ | $68.05 \pm 4.96$ | $68.75 \pm 4.89$ | $\mathbf{71.8 \pm 5.03}$ |
| 10 | $62.05 \pm 4.25$ | $61.81 \pm 3.86$ | $39.46 \pm 3.99$ | $62.3 \pm 4.17$ | $55.56 \pm 4.23$ | $63.03 \pm 4.45$ | $57.33 \pm 4.28$ | $\mathbf{70.22 \pm 5.24}$ |
| 20 | $56.6 \pm 4.52$ | $54.14 \pm 3.45$ | $39.22 \pm 3.57$ | $50.12 \pm 3.91$ | $45.94 \pm 3.93$ | $58.57 \pm 4.22$ | $45.03 \pm 3.96$ | $\mathbf{68.51 \pm 4.76}$ |
| 30 | $48.46 \pm 4.71$ | $53.22 \pm 3.8$ | $39.84 \pm 4.27$ | $43.28 \pm 3.83$ | $31.77 \pm 3.62$ | $50.83 \pm 4.37$ | $32.09 \pm 3.9$ | $\mathbf{65.07 \pm 5.99}$ |
| 40 | $40.14 \pm 5.63$ | $46.34 \pm 3.77$ | $33.02 \pm 4.64$ | $30.54 \pm 4.02$ | $18.12 \pm 3.18$ | $11.88 \pm 1.88$ | $17.18 \pm 3.07$ | $\mathbf{59.92 \pm 6.17}$ |
| 50 | $33.8 \pm 4.34$ | $34.5 \pm 3.45$ | $17.55 \pm 4.83$ | $22.2 \pm 3.81$ | $10 \pm 0$ | $10 \pm 0$ | $10 \pm 0$ | $\mathbf{47.65 \pm 4.32}$ |
| 60 | $10 \pm 0$ | $13.99 \pm 2.79$ | $15.59 \pm 3.38$ | $10 \pm 0$ | $10 \pm 0$ | $10 \pm 0$ | $10 \pm 0$ | $\mathbf{34.28 \pm 4.69}$ |
| 70 | $10 \pm 0$ | $10 \pm 0$ | $10 \pm 0$ | $10 \pm 0$ | $10 \pm 0$ | $10 \pm 0$ | $10 \pm 0$ | $\mathbf{10.68 \pm 0.68}$ |
| 80 | $10 \pm 0$ | $10 \pm 0$ | $10 \pm 0$ | $10 \pm 0$ | $10 \pm 0$ | $10 \pm 0$ | $10 \pm 0$ | $10 \pm 0$ |
| 90 | $10 \pm 0$ | $10 \pm 0$ | $10 \pm 0$ | $10 \pm 0$ | $10 \pm 0$ | $10 \pm 0$ | $10 \pm 0$ | $10 \pm 0$ |
| | Accuracy (%) with Hardware-implemented VGG-15 | | | | | | | |
| 0 | $68.11 \pm 4.87$ | $65.66 \pm 4.64$ | $68.62 \pm 4.3$ | $66.99 \pm 4.85$ | $66.42 \pm 4.1$ | $67.67 \pm 4.74$ | $67.74 \pm 4.94$ | $\mathbf{70.52 \pm 4.17}$ |
| 10 | $62.7 \pm 4.59$ | $58.94 \pm 4.41$ | $35.61 \pm 4.28$ | $63.56 \pm 4.7$ | $57.62 \pm 5.15$ | $65.05 \pm 4.53$ | $56.17 \pm 4.86$ | $\mathbf{67.43 \pm 4.86}$ |
| 20 | $58.61 \pm 4.55$ | $60.57 \pm 4.39$ | $35.2 \pm 4.51$ | $49.39 \pm 4.59$ | $39.26 \pm 5.28$ | $59.73 \pm 4.49$ | $38.58 \pm 4.72$ | $\mathbf{65.88 \pm 5.22}$ |
| 30 | $50.24 \pm 4.21$ | $52.77 \pm 4.99$ | $33.37 \pm 3.97$ | $42.84 \pm 4.15$ | $10 \pm 0$ | $22.38 \pm 4.31$ | $13.07 \pm 2.47$ | $\mathbf{61.9 \pm 5.07}$ |
| 40 | $37.15 \pm 4.36$ | $43.23 \pm 4.31$ | $15.6 \pm 3.28$ | $12.89 \pm 2.89$ | $10 \pm 0$ | $10 \pm 0$ | $10 \pm 0$ | $\mathbf{45.58 \pm 5.11}$ |
| 50 | $10 \pm 0$ | $10 \pm 0$ | $12.07 \pm 2.07$ | $10 \pm 0$ | $10 \pm 0$ | $10 \pm 0$ | $10 \pm 0$ | $\mathbf{15.4 \pm 2.54}$ |
| 60 | $10 \pm 0$ | $10 \pm 0$ | $11.98 \pm 1.98$ | $10 \pm 0$ | $10 \pm 0$ | $10 \pm 0$ | $10 \pm 0$ | $\mathbf{12.13 \pm 2.13}$ |
| 70 | $10 \pm 0$ | $10 \pm 0$ | $10 \pm 0$ | $10 \pm 0$ | $10 \pm 0$ | $10 \pm 0$ | $10 \pm 0$ | $10 \pm 0$ |
| 80 | $10 \pm 0$ | $10 \pm 0$ | $10 \pm 0$ | $10 \pm 0$ | $10 \pm 0$ | $10 \pm 0$ | $10 \pm 0$ | $10 \pm 0$ |
| 90 | $10 \pm 0$ | $10 \pm 0$ | $10 \pm 0$ | $10 \pm 0$ | $10 \pm 0$ | $10 \pm 0$ | $10 \pm 0$ | $10 \pm 0$ |

Table 16 reports the classification accuracy of the VGG-7/11/15 models, along with the baseline, benchmarks, and proposed mechanism under SAFs, using CIFAR-10. In the cases with deep convolution SNNs, our mechanism successfully enhances the fault tolerance of hardware-implemented SNNs based on the FPGA device.

### B.7.2. Learnable environment with Vivado-based FPGA emulation

We implement SNN models with our mechanism on an FPGA emulator based on Vivado. This emulator supports online training, and we demonstrate that our mechanism successfully improves the fault tolerance in online-trainable hardware environments.

*Table 17.* The classification accuracy of MLP and VGG-7 SNN models on Vivado emulator in a 95% confidence interval using MNIST, FMNIST, and CIFAR-10 under SAFs.

| Fault ratio(%) | Baseline | ECOC | SoftSNN | Routing | Astrocyte | FalVolt | LIFA | Proposed |
|---|---|---|---|---|---|---|---|---|
| | Accuracy (%) with Vivado-emulated MLP (MNIST) | | | | | | | |
| 0 | $95.18 \pm 4.11$ | $95.26 \pm 4.49$ | $95.06 \pm 4.51$ | $94.91 \pm 4.45$ | $94.79 \pm 4.21$ | $95.18 \pm 3.91$ | $94.98 \pm 4.49$ | $\mathbf{96.16 \pm 4.09}$ |
| 10 | $93.92 \pm 4.61$ | $94.03 \pm 4.46$ | $93.69 \pm 4.32$ | $93.86 \pm 4.29$ | $93.63 \pm 4.18$ | $94.43 \pm 4.66$ | $93.89 \pm 4.78$ | $\mathbf{95.03 \pm 4.67}$ |
| 20 | $91.76 \pm 4.56$ | $92.1 \pm 4.74$ | $11.35 \pm 4.84$ | $9.8 \pm 0$ | $25.52 \pm 4.07$ | $9.8 \pm 0$ | $27.35 \pm 3.61$ | $\mathbf{94.22 \pm 5.13}$ |
| 30 | $9.8 \pm 0$ | $11.35 \pm 0$ | $9.8 \pm 0$ | $9.8 \pm 0$ | $9.8 \pm 0$ | $9.8 \pm 0$ | $9.8 \pm 0$ | $\mathbf{93.64 \pm 5.02}$ |
| 40 | $9.8 \pm 0$ | $11.35 \pm 0$ | $9.8 \pm 0$ | $9.8 \pm 0$ | $9.8 \pm 0$ | $9.8 \pm 0$ | $9.8 \pm 0$ | $\mathbf{93.17 \pm 4.65}$ |
| 50 | $9.8 \pm 0$ | $10.32 \pm 0$ | $9.8 \pm 0$ | $9.8 \pm 0$ | $9.8 \pm 0$ | $9.8 \pm 0$ | $9.8 \pm 0$ | $\mathbf{90.51 \pm 4.52}$ |
| 60 | $9.8 \pm 0$ | $9.8 \pm 0$ | $9.8 \pm 0$ | $9.8 \pm 0$ | $9.8 \pm 0$ | $9.8 \pm 0$ | $9.8 \pm 0$ | $\mathbf{13.55 \pm 2.97}$ |
| 70 | $9.8 \pm 0$ | $9.8 \pm 0$ | $9.8 \pm 0$ | $9.8 \pm 0$ | $9.8 \pm 0$ | $9.8 \pm 0$ | $9.8 \pm 0$ | $9.8 \pm 0$ |
| 80 | $9.8 \pm 0$ | $9.8 \pm 0$ | $9.8 \pm 0$ | $9.8 \pm 0$ | $9.8 \pm 0$ | $9.8 \pm 0$ | $9.8 \pm 0$ | $9.8 \pm 0$ |
| 90 | $9.8 \pm 0$ | $9.8 \pm 0$ | $9.8 \pm 0$ | $9.8 \pm 0$ | $9.8 \pm 0$ | $9.8 \pm 0$ | $9.8 \pm 0$ | $9.8 \pm 0$ |
| | Accuracy (%) with Vivado-emulated MLP (FMNIST) | | | | | | | |
| 0 | $85.33 \pm 1.21$ | $84.96 \pm 1.63$ | $85.27 \pm 2.08$ | $85.38 \pm 1.51$ | $84.47 \pm 1.71$ | $84.59 \pm 1.49$ | $84.88 \pm 1.36$ | $\mathbf{87.86 \pm 1.07}$ |
| 10 | $84.98 \pm 1.93$ | $85.19 \pm 2.11$ | $84.24 \pm 2.26$ | $84.41 \pm 2.25$ | $84.02 \pm 2.12$ | $84.42 \pm 2.17$ | $84.21 \pm 2.08$ | $\mathbf{85.59 \pm 2.18}$ |
| 20 | $10 \pm 0$ | $10 \pm 0$ | $10 \pm 0$ | $10 \pm 0$ | $10 \pm 0$ | $81.37 \pm 3.16$ | $10 \pm 0$ | $\mathbf{85.11 \pm 2.83}$ |
| 30 | $10 \pm 0$ | $10 \pm 0$ | $10 \pm 0$ | $10 \pm 0$ | $10 \pm 0$ | $10 \pm 0$ | $10 \pm 0$ | $\mathbf{84.75 \pm 3.43}$ |
| 40 | $10 \pm 0$ | $10 \pm 0$ | $10 \pm 0$ | $10 \pm 0$ | $10 \pm 0$ | $10 \pm 0$ | $10 \pm 0$ | $\mathbf{82.81 \pm 5.01}$ |
| 50 | $10 \pm 0$ | $10 \pm 0$ | $10 \pm 0$ | $10 \pm 0$ | $10 \pm 0$ | $10 \pm 0$ | $10 \pm 0$ | $\mathbf{36.46 \pm 5.22}$ |
| 60 | $10 \pm 0$ | $10 \pm 0$ | $10 \pm 0$ | $10 \pm 0$ | $10 \pm 0$ | $10 \pm 0$ | $10 \pm 0$ | $\mathbf{24.23 \pm 4.58}$ |
| 70 | $10 \pm 0$ | $10 \pm 0$ | $10 \pm 0$ | $10 \pm 0$ | $10 \pm 0$ | $10 \pm 0$ | $10 \pm 0$ | $10 \pm 0$ |
| 80 | $10 \pm 0$ | $10 \pm 0$ | $10 \pm 0$ | $10 \pm 0$ | $10 \pm 0$ | $10 \pm 0$ | $10 \pm 0$ | $10 \pm 0$ |
| 90 | $10 \pm 0$ | $10 \pm 0$ | $10 \pm 0$ | $10 \pm 0$ | $10 \pm 0$ | $10 \pm 0$ | $10 \pm 0$ | $10 \pm 0$ |
| | Accuracy (%) with Vivado-emulated VGG-7 (CIFAR-10) | | | | | | | |
| 0 | $68.34 \pm 4.87$ | $65.67 \pm 4.64$ | $66.86 \pm 4.3$ | $68.05 \pm 4.85$ | $67.64 \pm 4.1$ | $68.31 \pm 4.74$ | $67.96 \pm 4.94$ | $\mathbf{73.12 \pm 4.17}$ |
| 10 | $60.9 \pm 4.59$ | $54.83 \pm 4.41$ | $43.61 \pm 4.28$ | $61.54 \pm 4.7$ | $56.82 \pm 5.15$ | $62.59 \pm 4.53$ | $57.71 \pm 4.86$ | $\mathbf{71.04 \pm 4.86}$ |
| 20 | $52.87 \pm 4.55$ | $52.56 \pm 4.39$ | $41.24 \pm 4.51$ | $51.13 \pm 4.59$ | $46.63 \pm 5.28$ | $55.8 \pm 4.49$ | $45.32 \pm 4.72$ | $\mathbf{69.44 \pm 4.91}$ |
| 30 | $49.26 \pm 4.21$ | $50.72 \pm 4.99$ | $40.48 \pm 3.97$ | $40.85 \pm 4.15$ | $30.93 \pm 0$ | $50.51 \pm 4.31$ | $34.5 \pm 2.47$ | $\mathbf{66.31 \pm 5.07}$ |
| 40 | $37.41 \pm 4.36$ | $35.01 \pm 4.31$ | $36.92 \pm 3.28$ | $36.39 \pm 2.89$ | $14.21 \pm 0$ | $16.37 \pm 0$ | $10 \pm 0$ | $\mathbf{60.86 \pm 5.11}$ |
| 50 | $17.69 \pm 0$ | $29.78 \pm 0$ | $31.21 \pm 2.07$ | $15.92 \pm 0$ | $10 \pm 0$ | $10 \pm 0$ | $10 \pm 0$ | $\mathbf{48.93 \pm 5.54}$ |
| 60 | $10 \pm 0$ | $10 \pm 0$ | $25.26 \pm 1.98$ | $10 \pm 0$ | $10 \pm 0$ | $10 \pm 0$ | $10 \pm 0$ | $\mathbf{35.65 \pm 5.13}$ |
| 70 | $10 \pm 0$ | $10 \pm 0$ | $10 \pm 0$ | $10 \pm 0$ | $10 \pm 0$ | $10 \pm 0$ | $10 \pm 0$ | $\mathbf{19.89 \pm 3.98}$ |
| 80 | $10 \pm 0$ | $10 \pm 0$ | $10 \pm 0$ | $10 \pm 0$ | $10 \pm 0$ | $10 \pm 0$ | $10 \pm 0$ | $10 \pm 0$ |
| 90 | $10 \pm 0$ | $10 \pm 0$ | $10 \pm 0$ | $10 \pm 0$ | $10 \pm 0$ | $10 \pm 0$ | $10 \pm 0$ | $10 \pm 0$ |

Table 17 shows the classification accuracy of MLP and VGG-7 SNNs with the benchmarks and proposed mechanism on the learnable FPGA emulator based on Vivado using MNIST, FMNIST, and CIFAR-10. As shown in the table, our mechanism enhances the fault tolerance of SNNs substantially with online trainable environments.

## C. Efficiency analysis based on time/space complexity and time/energy consumption

To demonstrate that the proposed mechanism improves the fault tolerance of SNN models without relying on complex algorithms, we analyze its computational and space complexity in comparison with benchmark methods. In addition, we measure the energy consumption and training time of the proposed mechanism on an FPGA device in comparison with the benchmark methods.

### C.1. Complexity analysis

We derive the time and space complexities of the proposed mechanism and the benchmarks.

#### C.1.1. NOTATIONS

Here, we explain the notations of our time and space complexity equations.

- $B$: batch size

- $CPU$: the cost of operating a CPU

- $L$: ECOC code length (bits)

- $T_0$: baseline number of SNN time steps (e.g., `num_steps`)

- $T_{\text{selected}}$: the number of SNN time steps selected by the proposed mechanism

- $Train(T)$: cost of running the base SNN for $T$ time steps with forward+backward (loop/BPTT)

- $Test(T)$: cost of running the base SNN for $T$ time steps (forward only)

- $P$: total number of Conv/Linear weight parameters (weights touched by weight-based methods)

- $P_{\text{fault}}$: total number of Conv/Linear weight parameters under faults

- $P_{\text{mask}}$: total number of Conv/Linear weight parameters masked

- $P_{\text{limit}}$: per-layer/channel thresholds and hyperparameters to set the limit of weights

- $P_0$: total number of initial weight parameters in SNNs

- $P_{\text{healthy}}$: total number of weight parameters of synapses without faults

- $A$: total number of activation elements processed by slot-activity tracking per timestep

- $N_{classes}$: number of classes

- $N_{neurons}$: number of neurons

- $N_{card}$: cardinality value

- $S_{\text{input}}$: size of input samples

- $C_x$: size of the channels for a convolution layer, $x$ the name of channels (e.g, input and output), and $\ell$ the depth of layers

- The proposed mechanism's candidates set $\mathcal{T}$ with $M = |\mathcal{T}|$, $S_T = \sum_{T \in \mathcal{T}} T$, $T_{\max} = \max(\mathcal{T})$, and $T_{\text{sel}}$ selected steps at inference

### C.1.2. TRAINING

We thoroughly analyze the time and space complexity of the benchmarks and the proposed mechanism during training. The following items present the time complexity of the benchmarks and the proposed mechanism.

1. **ECOC**

   (a) Time complexity: $\mathcal{O}(Train(T_0) + BN_{\text{class}}L)$ (note: last layer output dim becomes $L$)
   (b) space complexity: $\mathcal{O}(N_{\text{class}}L + BN_{\text{class}}L)$ (codebook + hamming distance storage)
   (c) Overhead evaluation result: Medium

2. **SoftSNN**

   (a) Time complexity: $\mathcal{O}(Train(T_0) + T_0P)$ (pre-forward hook touches all weights each timestep)
   (b) space complexity: $\mathcal{O}(P_{\text{limit}})$
   (c) Overhead evaluation result: High

3. **Routing**

   (a) Time complexity: $\mathcal{O}(Train(T_0) + T_0A)$+ one-time $\mathcal{O}(P)$ (routing update at fault epoch)
   (b) space complexity: $\mathcal{O}(P + \sum_{\ell} C_{\text{in},\ell})$ (fault masks + slot-activity exponential moving average
   (c) Overhead evaluation result: High

4. **Astrocyte**

   (a) Time complexity: $\mathcal{O}(Train(T_0) + P)$ (extra pass over weights in backward via grad hook)
   (b) space complexity: $\mathcal{O}(P)$ CPU (cached $P_0$, $P_{\textbf{healthy}}$, and $q$)
   (c) Overhead evaluation result: Very High

5. **FalVolt**

   (a) Time complexity: Worst-case: $\mathcal{O}(Train(T_0) + P_{\text{fault}})$, with $P_{\text{fault}} \leq P$
   (b) space complexity: $\mathcal{O}(N_{\text{neurons}}) + \mathcal{O}(P_{\text{mask}})$
   (c) Overhead evaluation result: Medium

6. **LIFA**

   (a) Time complexity: $\mathcal{O}(Train(T_0) + T_0A + P)$ (tracker + per-batch weight-hook repair)
   (b) space complexity: $\mathcal{O}(P)$ CPU $+\mathcal{O}(\sum C_{\text{in}} + \sum C_{\text{out}})$
   (c) Overhead evaluation result: Very High

7. **Proposed**

   (a) Time complexity : $\mathcal{O}(Train(T_{\max}) + BC_xS_{\text{input}}(S_T + N_{card}T_{\max}))$
       with $\mathcal{T} = \{2, 4, 8\}$: $\mathcal{O}(Train(8) + 38BC_xS_{\text{input}})$ $(38 = 2 + 4 + 8 + 8 * 3, (3 = Num(\mathcal{T})))$
   (b) space complexity: $\mathcal{O}(BC_xS_{\text{input}}S_T)$
       with $\mathcal{T} = \{2, 4, 8\}$: $\mathcal{O}(14BC_xS_{\text{input}})$ (store all candidate frags)
   (c) Overhead evaluation result: **High (lower than Astrocyte and LIFA, comparable to SoftSNN)**

### C.1.3. INFERENCE

We evaluate the time and space complexity of the benchmarks and the proposed mechanism during inference in detail. The following items present the space complexity of the benchmarks and the proposed mechanism

1. **ECOC**

   (a) Time complexity: $\mathcal{O}(Test(T_0) + BN_{\text{classes}}L)$ (decode = nearest codeword)
   (b) space complexity: $\mathcal{O}(N_{\text{classes}}L + BN_{\text{classes}}L)$

    (c) Overhead evaluation result: Medium

2. **SoftSNN**

    (a) Time complexity: $\mathcal{O}(Test(T_0) + T_0 P)$ (pre-forward hook still active)

    (b) space complexity: $\mathcal{O}(P_{\text{limit}})$

    (c) Overhead evaluation result: High

3. **Routing**

    (a) Time complexity: $\mathcal{O}(Test(T_0) + T_0 A)$ (tracker still runs unless removed)

    (b) space complexity: $\mathcal{O}(P + \sum_\ell C_{\text{in},\ell})$

    (c) Overhead evaluation result: High

4. **Astrocyte**

    (a) Time complexity: $\mathcal{O}(Test(T_0))$ (no backward, so grad hook not executed)

    (b) space complexity: $\mathcal{O}(P)$ CPU cached tensors

    (c) Overhead evaluation result: High

5. **FalVolt**

    (a) Time complexity: $\mathcal{O}(Test(T_0))$

    (b) space complexity: $\mathcal{O}(N_{\text{neurons}})$ (plus masks if stored)

    (c) Overhead evaluation result: Low

6. **LIFA**

    (a) Time complexity: $\mathcal{O}(Test(T_0) + T_0 A)$ (tracker overhead persists)

    (b) space complexity: $\mathcal{O}(P)$ CPU $+\mathcal{O}(\sum C_{\text{in}} + \sum C_{\text{out}})$

    (c) Overhead evaluation result: High

7. **Proposed**

    (a) Time complexity: $\mathcal{O}(Test(T_{\text{selected}}) + BC_x S_{\text{input}} T_{\text{selected}})$
        with $\mathcal{T} = \{2, 4, 8\}$, $T_{\text{selected}} = 8$: $\mathcal{O}(Test(8) + 8BC_x S_{\text{input}})$

    (b) space complexity: $\mathcal{O}(BC_x S_{\text{input}} T_{\text{sel}})$ (peak)
        with $\mathcal{T} = \{2, 4, 8\}$, $T_{\text{selected}} = 8$: $\mathcal{O}(8BC_x S_{\text{input}})$

    (c) Overhead evaluation result: **Medium (lower than Astrocyte, LIFA, and SoftSNN, comparable to ECOC)**

## C.2. Training time on the GPU

We measure the training time of the baseline, benchmarks, and the proposed mechanism using the MLP (MNIST), VGG-7 (CIFAR-10), ResNet-18 (CIFAR-100), and ResNet-34 (Tiny-ImageNet) models with 8 time steps. We train the models on a workstation with an Nvidia GeForce RTX 4080 GPU with Ubuntu 24.04.

*Table 18.* Various models' training time (sec) in a 95% confidence interval with the baseline, benchmarks, and proposed mechanism on a workstation under SAFs with a fault ratio of 50% (using 8 time steps for the baseline and benchmarks),

| | Baseline | ECOC | SoftSNN | Routing | Astrocyte | FalVolt | LIFA | Proposed |
|---|---|---|---|---|---|---|---|---|
| MLP (MNIST) | 770.46 ± 9.07 | 791.38 ± 2.85 | 846.06 ± 10.38 | 793.16 ± 12.4 | 1156.99 ± 17.74 | 805.58 ± 12.94 | 1175.44 ± 19.38 | 852.74 ± 19.65 |
| VGG-7 (CIFAR-10) | 1159.64 ± 14.46 | 1187.48 ± 15.52 | 1277.47 ± 15.19 | 1194.14 ± 16.09 | 1407.29 ± 21.08 | 1213.76 ± 17.08 | 1426.59 ± 22.6 | 1242.37 ± 24.47 |
| ResNet-18 (CIFAR-100) | 1520.11 ± 15.32 | 1542.45 ± 16.23 | 1638.04 ± 18.07 | 1587.27 ± 25.95 | 2887.85 ± 26.62 | 1639.94 ± 20.15 | 2899.36 ± 25.37 | 1598.84 ± 19.68 |
| ResNet-34 (Tiny-ImageNet) | 16538.41 ± 64.47 | 17207.4 ± 79.52 | 17991.54 ± 86.28 | 17403.32 ± 102.37 | 32021.14 ± 145.75 | 17271.56 ± 103.16 | 32616.66 ± 131.32 | 17847.34 ± 101.39 |

Table 18 reports the training time of SNNs with benchmarks and our mechanism. The SNN with our mechanism consumes significantly less training time than weight-scanning approaches (Astrocyte and LIFA), as we estimate time complexities in Subsection C.1 of the Appendix. We demonstrate that our mechanism definitely consumes less time than weight-scanning approaches, which introduce additional weight/state processing overhead during optimization in actual situations. This is because the overall overhead of our mechanism remains moderate across model and dataset scales, compared to weight-scanning approaches.

### C.3. Latency and energy consumption on the real FPGA device

We measure the energy consumption of the model with the baseline, benchmarks, and proposed mechanism on the FPGA device during inference. Table 20 exhibits the energy consumption of the FPGA-based MLP with MNIST/FMNIST and FPGA-based VGG-7/11/15 with CIFAR-10 using 8 time steps.

*Table 19.* The SNN models' time consumption (μ sec) to process a single sample in a 95% confidence interval with the baseline, benchmarks, and proposed mechanism on the real FPGA hardware under SAFs with a fault ratio of 50% (using 8 time steps for the baseline and benchmarks, 200 cycles in FPGA) during inference.

|                    | Baseline        | ECOC            | SoftSNN         | Routing         | Astrocyte         | FalVolt         | LIFA              | Proposed        |
|--------------------|-----------------|-----------------|-----------------|-----------------|-------------------|-----------------|-------------------|-----------------|
| MLP (MNIST)        | $297.59 \pm 2.27$ | $312.53 \pm 2.41$ | $354.08 \pm 2.96$ | $338.15 \pm 2.49$ | $406.78 \pm 3.18$   | $323.6 \pm 2.92$  | $413.65 \pm 3.09$   | $306.42 \pm 2.34$ |
| MLP (FMNIST)       | $305.81 \pm 2.38$ | $318.76 \pm 2.66$ | $368.93 \pm 3.17$ | $345.36 \pm 3.32$ | $419.11 \pm 3.87$   | $331.52 \pm 3.46$ | $430.28 \pm 4.15$   | $314.48 \pm 2.73$ |
| VGG-7 (CIFAR-10)   | $638.72 \pm 5.03$ | $656.09 \pm 5.27$ | $686.18 \pm 5.41$ | $665.59 \pm 5.58$ | $765.26 \pm 6.23$   | $653.24 \pm 5.45$ | $783.1 \pm 6.75$    | $649.81 \pm 5.32$ |
| VGG-11 (CIFAR-10)  | $817.38 \pm 6.86$ | $848.21 \pm 6.95$ | $883.78 \pm 7.14$ | $869.26 \pm 6.82$ | $957.53 \pm 7.62$   | $862.35 \pm 6.74$ | $969.58 \pm 7.24$   | $829.72 \pm 6.87$ |
| VGG-15 (CIFAR-10)  | $903.84 \pm 7.39$ | $916.62 \pm 7.68$ | $946.47 \pm 7.75$ | $932.8 \pm 7.51$  | $1048.39 \pm 9.01$  | $927.44 \pm 7.49$ | $1062.59 \pm 9.14$  | $918.37 \pm 7.64$ |

Table 19 shows the time latency of hardware-implemented SNNs employing benchmark methods and the proposed mechanism to process a single sample. SNNs with the proposed mechanism exhibit a latency comparable to that of ECOC-based SNNs. This result indicates that our measurement matches the time and space complexity analysis presented in Appendix C.1.

*Table 20.* The SNN models' energy consumption (μJ) to process a single sample in a 95% confidence interval with the baseline, benchmarks, and proposed mechanism on the real FPGA hardware under SAFs with a fault ratio of 50% (using 8 time steps for the baseline and benchmarks, 200 cycles in FPGA) during inference.

|                    | Baseline        | ECOC            | SoftSNN         | Routing         | Astrocyte         | FalVolt         | LIFA              | Proposed        |
|--------------------|-----------------|-----------------|-----------------|-----------------|-------------------|-----------------|-------------------|-----------------|
| MLP (MNIST)        | $171.62 \pm 2.12$ | $177.64 \pm 2.55$ | $182.39 \pm 2.37$ | $180.87 \pm 2.63$ | $189.35 \pm 2.59$   | $190.23 \pm 2.54$ | $193.27 \pm 2.46$   | $176.03 \pm 2.09$ |
| MLP (FMNIST)       | $175.35 \pm 2.77$ | $181.52 \pm 3.12$ | $189.17 \pm 2.86$ | $185.96 \pm 3.13$ | $192.29 \pm 3.54$   | $196.4 \pm 3.51$  | $199.13 \pm 2.72$   | $180.69 \pm 2.48$ |
| VGG-7 (CIFAR-10)   | $405.78 \pm 4.34$ | $433.74 \pm 4.46$ | $441.61 \pm 3.94$ | $456.34 \pm 4.91$ | $463.07 \pm 5.16$   | $419.75 \pm 5.92$ | $485.36 \pm 5.34$   | $432.24 \pm 4.99$ |
| VGG-11 (CIFAR-10)  | $592.32 \pm 4.69$ | $634.19 \pm 4.78$ | $658.24 \pm 4.73$ | $671.88 \pm 5.35$ | $683.47 \pm 5.41$   | $625.41 \pm 6.26$ | $697.48 \pm 5.83$   | $629.79 \pm 5.23$ |
| VGG-15 (CIFAR-10)  | $726.09 \pm 5.27$ | $748.81 \pm 5.52$ | $791.02 \pm 5.76$ | $814.37 \pm 5.8$  | $829.94 \pm 6.05$   | $737.48 \pm 6.58$ | $849.28 \pm 6.44$   | $742.41 \pm 5.82$ |

Table 20 reports the energy consumption of hardware-implemented SNNs employing benchmark methods and the proposed mechanism to process a single sample. SNNs equipped with the proposed mechanism exhibit energy consumption comparable to that of ECOC-based SNNs, consistent with the time and space complexity analysis presented in Appendix C.1, as observed for the latency.

Considering the latency and energy consumption, in practical hardware implementations, the proposed mechanism does not need to generate fragments for all $T \in \mathcal{T}$ at inference; it generates fragments only for the selected $T_{selected}$. Consequently, the overall energy consumption of the proposed mechanism is comparable to ECOC and remains lower than most of the other benchmark methods.

## D. Detailed mathematical explanation of the motivation study

In this section, we mathematically demonstrate that synaptic faults reduce usable learning capacities of SNN models.

### D.1. Neuron model and surrogate gradient

Consider a spiking neuron in $l^{th}$ layer with membrane potential[9] $u_t^l \in \mathbb{R}$, threshold $\vartheta \in \mathbb{R}$, and spike output $s_t^l \in \{0, 1\}$. The neuronal dynamic is expressed as follows.

$$u_t^l = \alpha u_{t-1}^l + i_t^l - \beta s_{t-1}^l$$
$$s_t^l = H(u_t^l - \vartheta) \tag{11}$$

where $u_t^l$ and $i_t^l$ represent the membrane potential and the input current for a neuron at layer $l$ and time $t$, respectively. $H(\cdot)$ is the Heaviside step function; and $\beta$ is the reset strength. $i_t^l$ is expressed as

$$i_t^l = \sum_{i=1}^{n^{l-1}} w_i^{l-1} s_{t,i}^{l-1} + d \tag{12}$$

where $s_{t,i}^{l-1}$ is the output spike of input neuron $i$ in the previous layer $(l-1)$, $w_i^{l-1}$ denotes the synaptic weight that connects input neuron $i$ in the previous layer $(l-1)$, $n^{l-1}$ is the number of neurons in layer $l-1$, and $d$ represents the bias term.

During training, we use the surrogate gradient function $g_t^l \approx \frac{\partial s_t^l}{\partial u_t^l}$, which is typically used in Spatio-Temporal Backpropagation (STBP) (Wu et al., 2018). Similar to (Lian et al., 2023; Wu et al., 2018), in our analysis, we use the surrogate gradient function, $g_t^l = g(u_t^l)$, which is advantageous in on-chip learning. $g(u) > 0$, if $|u - \vartheta| < \xi$, and $g(u) \approx 0$, otherwise[10]. Here, $\xi$ denotes the width of the surrogate gradient corridor.

### D.2. Fault model

We consider synaptic faults that perturb synaptic weight as follows

$$\hat{w}_i^l = w_i^l + \delta w_i^l \tag{13}$$

where $\hat{w}_i^l$ is the synaptic weight of SNNs that suffered from SAF and $\delta w_i^l$ is the perturbation in a synaptic weight caused by the faults. In this analysis, we consider Stuck-At-Fault (SAF), where SA1 is dominant over SA0, which is normally observed in the experimental campaign (Chen et al., 2017).

The synaptic weight is bounded such that $w_i^l, \hat{w}_i^l \in [-k, k]$ where $k > 0$ is the synaptic weight when SA1 occurs. Then, $\hat{w}_i^l$ can be described as follows.

$$\hat{w}_i^l = \begin{cases} k, & \text{if SA1} \\ -k, & \text{if SA0} \\ w_i^l, & \text{no fault} \end{cases} \tag{14}$$

The perturbation in synaptic weights can be calculated as follows.

$$\delta_{w_i^l} = \hat{w}_i^l - w_i^l = \begin{cases} k - w_i^l, & \text{if SA1} \\ -k - w_i^l, & \text{if SA0} \\ 0, & \text{no fault} \end{cases} \tag{15}$$

Using Equation (15), the membrane potential of a neuron in $l^{th}$ layer when SAF occurs, $\hat{u}_t^l$, can be described as follows:

---

[9]To simplify the notation, we omit the neuron index.

[10]The representative functions for surrogate gradient functions are arctangent and rectangular functions.

$$\hat{u}_t^l = u_t^l + pert_t^l$$

$$pert_t^l = \alpha^t b_0^l + \sum_{m=0}^{t-1} \alpha^m k_{t-m}^l \tag{16}$$

$$k_t^l = \sum_{i=1}^{n^{l-1}} \delta_{w_i^{l-1}} \cdot x_{t,i}^{l-1}$$

where $pert_t^l$ is the perturbation in membrane potential caused by synaptic faults. $x_{t,i}^{l-1}$ is the input spike generated from neuron $i$ at the previous layer and equal to $s_{t,i}^{l-1}$.

### D.3. Impact of faults on membrane potentials outside the surrogate gradient corridor

In this subsection, we will show that the membrane potential is more likely to lie outside the surrogate gradient corridor when SAF occurs in SNNs.

**Lemma D.1** (Increased zero-gradient probability under SAF). *Let $u_t^l$ denote the membrane potential of a neuron at time $t$ and layer $l$ in the fault-free case, and let $\vartheta$ be the firing threshold. Let $\hat{u}_t^l = u_t^l + pert_t^l$ also be the membrane potential under SAF, where $pert_t^l$ denotes the SAF-induced perturbation. Define $X := u_t^l - \vartheta$ and $F := pert_t^l$.*

*Then, under the approximations stated below,*

$$P\big(g(\hat{u}_t^l) = 0\big) \geq P\big(g(u_t^l) = 0\big).$$

The random variable $X$ has a continuous, symmetric, unimodal density $f_X$ with a unique maximum at 0, and $f_X$ is non-increasing for $x \geq 0$.

*Proof.* By the definition of the surrogate gradient,

$$g(u_t^l) \neq 0 \iff |u_t^l - \vartheta| < \xi \iff |X| < \xi,$$

and under SAF,

$$g(\hat{u}_t^l) \neq 0 \iff |\hat{u}_t^l - \vartheta| < \xi \iff |X + F| < \xi.$$

Define

$$\phi(k) := P(|X - k| < \xi) = \int_{k-\xi}^{k+\xi} f_X(x)\,dx.$$

Since $f_X$ is symmetric and unimodal with its maximum at 0, $\phi(c)$ is an even function that is decreasing in $|c|$.

By the law of total expectation,

$$P(|X + F| < \xi) = \mathbb{E}\big[P(|X + F| < \xi \mid F)\big] = \int_k P(|X + k| < \xi) f_F(k) dk$$

where $f_F()$ is pdf of $F$.

As $\phi(c)$ is even and decreasing in $|c|$

$$\int_k P(|X + k| < \xi) f_F(k) dk \leq \int_k \phi(0) f_F(k) dk = \phi(0) = P(|X| < \xi).$$

Therefore,

$$P\big(g(\hat{u}_t^l) \neq 0\big) \leq P\big(g(u_t^l) \neq 0\big).$$

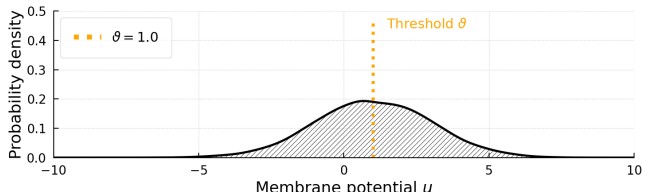

*Figure 8.* Probability distribution of membrane potential

Taking the complements yields

$$P\big(g(\hat{u}_t^l) = 0\big) \geq P\big(g(u_t^l) = 0\big),$$

which completes the proof.

$\square$

*Remark* D.2. (Justification of (A1) in Lemma D.1) Surrogate-gradient-based training provides a non-zero gradient primarily in a narrow neighborhood around the firing threshold, which induces an optimization pressure to keep membrane potentials visiting the threshold vicinity rather than saturating far away from it. This "threshold-centric" learning dynamics is common to widely-used SG frameworks such as STBP. In addition, recent works (Guo et al., 2022; 2023) explicitly identify unstable membrane-potential distributions as a key training bottleneck and provide methods typically adopted in SNN, which makes $X := u_t^l - \vartheta$ centered and unimodal around zero, consistent with a unimodal density peaked at zero.

To strengthen the justification of (A1), we measure membrane potentials during training an MLP using SpikingJelly. Our experimental results on Figure 8 demonstrate that the membrane potential follows a distribution, which is centered and unimodal around the spiking threshold. These results are equivalent to the fact that the potential shifted by the threshold, $X$, follows a unimodal density peaked at zero.

### D.4. Impact of faults on the gradient

The gradient at layer $l$, $\frac{\partial L}{\partial w^l}$ is defined as follows.

$$\begin{aligned} \frac{\partial L}{\partial w^l} &= \sum_{t=1}^{T} \frac{\partial L}{\partial u_t^l} \frac{\partial u_t^l}{\partial w^l} \\ &= \sum_{t=1}^{T} \frac{\partial L}{\partial u_t^l} x_t^l \end{aligned} \tag{17}$$

where $T$ is the number of time-steps and $x_t^l \in \{0, 1\}$ is the input to neuron in the $l^{th}$ layer.

Let $\delta_t^l := \frac{\partial L}{\partial u_t^l}$, which can be expressed by[11]

$$\begin{aligned} \delta_t^l &= \frac{\partial L}{\partial s_t^l} g_t^l + \delta_{t+1}^l \frac{\partial u_{t+1}^l}{\partial u_t^l} \\ &= \frac{\partial L}{\partial s_t^l} g_t^l + \delta_{t+1}^l \cdot \alpha \\ &= \underbrace{w^{l+1} \delta_t^{l+1} g_t^l}_{\text{spatial term}} + \underbrace{\delta_{t+1}^l \cdot \alpha}_{\text{temporal term}} \end{aligned} \tag{18}$$

---

[11]For analytical simplicity, we ignore the gradient through the reset path in the temporal Jacobian, i.e., $\frac{\partial u_{t+1}}{\partial u_t} \approx \alpha$. This is a common surrogate-gradient training practice to avoid error accumulation through instantaneous reset events and to improve training stability. For this reason, many recent works employ this simplification (Zhang et al., 2024b; Zenke & Vogels, 2021; Yamamoto et al., 2022; Fang et al., 2021).

Then, we unroll the recursive Equation (19), resulting in the general closed-form equation, which is expressed as follows.

$$\delta_t^l = \delta_{t,main}^l + \delta_{t,boundary}^l$$

$$\delta_{t,main}^l = \sum_{k_l=t}^{T} \sum_{k_{l+1}=k_l}^{T} \cdots \sum_{q=k_{L-1}}^{T} \alpha^{q-t}(\Pi_{j=l}^{L-1} g_{k_j}^j) \mathbb{W}_{(l+1),L} \delta_q^L \qquad (19)$$

$$\delta_{t,boundary}^l = \alpha^{T+1-t} \sum_{r=l}^{L-1} [\mathbb{W}_{(l+1),r} \mathbb{G}_{l,r-1} \delta_{T+1}^r]$$

where $\mathbb{W}_{a,b} := \Pi_{j=a}^{b} w_j$ and $\mathbb{G}_{a,b}(t) := \sum_{k_a=t}^{T} \sum_{k_{a+1}=k_a}^{T} \cdots \sum_{k_b=k_b-1}^{T} (\Pi_{j=a}^{b} g_{k_j}^j)$.

We note that $\Pi_{j=l}^{L-1} g_{k_j}^j$ becomes zero if at least one $g_{k_j}^j$ among $L-l+1$ terms go to zero. Lemma D.1 shows that the frequency of zero surrogate-gradient events increases under SAF, which means that the probability that a single $g_{k_j}^j$ goes to zero becomes larger under SAF. Thus, we conclude that $\delta_{t,main}^l$ is more strongly attenuated due to the increased frequency of zero surrogate-gradient events under SAF.

Moreover, the boundary term, $\delta_{t,boundary}^l$, is also strongly attenuated under SAF due to the following two interacting reasons. First, $\Pi_{j=a}^{b} g_{k_j}^j$ in $\mathbb{G}_{l,r-1}$ is more strongly attenuated due to the increased frequency of zero surrogate-gradient events under SAF. Second, as $0 < \alpha < 1$, $\alpha^{T-t+1}$ exponentially suppresses the boundary contribution. These two effects make $\delta_{t,boundary}^l$ decay more strongly under SAF, making it negligible.

As $\delta_{t,main}^l$ and $\delta_{t,boundary}^l$ are strongly attenuated under SAF, $\delta_t^l$ is significantly attenuated, rapidly decaying toward a very small value. From this reasoning, we conclude that SAF pushes $\frac{\partial L}{\partial w^l}$ toward zero, which shows that gradient vanishing is more likely to occur when SAF occurs.

It is obvious that $\Pi_{j=l}^{L-1} g_{k_j}^j$ more likely goes towards zero as $l$ decreases. Thus, the gradient can vanish with higher probability. This shows that the gradient is attenuated exponentially as $l$ decreases.

### D.5. Input fragmentation to mitigate the gradient vanishing problem

In this subsection, we analytically explain why input fragmentation can mitigate gradient vanishing by reducing the probability of surrogate-corridor escape.

When using input fragmentation, the input is distributed across multiple time steps, thereby reducing the effective per-time-step input intensity. To analytically isolate only this attenuation effect, we consider a simple fragmentation strategy, which is described as follows.

$$x_{t,i,\text{frag}}^{l-1} = x_{t,i}^{l-1} y_{t,i}, \qquad (20)$$

where $y_{t,i} \sim \text{Bernoulli}(1/N)$, $N :=$ Number of fragments. With input fragmentation, the instantaneous perturbation becomes

$$k_{t,\text{frag}}^l = \sum_{i=1}^{n^{l-1}} \delta_{w_i^{l-1}} x_{t,i,\text{frag}}^{l-1},$$

and the corresponding accumulated perturbation is

$$F_{\text{frag}} := pert_{t,\text{frag}}^l = \alpha^t b_{0,\text{frag}}^l + \sum_{k=0}^{t-1} \alpha^k k_{t-k,\text{frag}}^l.$$

**Lemma D.3.** *For any $\xi > 0$,*

$$P(|X + F_{\text{frag}}| < \xi) \geq P(|X + F| < \xi), \qquad (21)$$

*where $X := u_t^l - \vartheta$ is the fault-free baseline variable used in this inequality, under the following approximations: (1) $\delta_{w_i^{l-1}} \geq 0$ almost surely for all $i$[12]; (2) $b_{0,\text{frag}}^l = b_0^l$.*

*Proof.* As $x_{t,i}^{l-1} \in \{0,1\}$ and $y_{t,i} \in \{0,1\}$, the following inequality holds.

$$0 \leq x_{t,i,\text{frag}}^{l-1} = x_{t,i}^{l-1} y_{t,i} \leq x_{t,i}^{l-1}.$$

Thus, the following inequality can be derived.

$$0 \leq k_{t,\text{frag}}^l = \sum_{i=1}^{n^{l-1}} \delta_{w_i^{l-1}} x_{t,i,\text{frag}}^{l-1} \leq \sum_{i=1}^{n^{l-1}} \delta_{w_i^{l-1}} x_{t,i}^{l-1} = k_t^l$$

almost surely. Since $\alpha \geq 0$, the following inequality holds.

$$F_{\text{frag}} = pert_{t,\text{frag}}^l = \alpha^t b_{0,\text{frag}}^l + \sum_{k=0}^{t-1} \alpha^k k_{t-k,\text{frag}}^l \leq \alpha^t b_0^l + \sum_{k=0}^{t-1} \alpha^k k_{t-k}^l = F.$$

almost surely. From the proof of Lemma D.1, $\phi(c) := P(|X - c| < \xi)$ is an even function and is non-increasing in $|c|$. Since

$$0 \leq F_{\text{frag}} \leq F$$

almost surely. We obtain

$$\phi(F_{\text{frag}}) \geq \phi(F)$$

almost surely. Taking the expectation on both sides yields

$$\mathbb{E}[\phi(F_{\text{frag}})] \geq \mathbb{E}[\phi(F)].$$

Using the same conditioning argument as in Lemma D.1.

We have

$$P(|X + F_{\text{frag}}| < \xi) = \mathbb{E}[\phi(F_{\text{frag}})]$$

and

$$P(|X + F| < \xi) = \mathbb{E}[\phi(F)].$$

Hence,

$$P(|X + F_{\text{frag}}| < \xi) \geq P(|X + F| < \xi).$$

$\square$

This result shows that input fragmentation weakens the accumulated SAF-induced membrane perturbation and thus increases the probability that the perturbed membrane potential remains inside the surrogate-gradient corridor. Using similar reasoning to that in the previous subsection, the reduced frequency of zero-gradient events suggests that $\delta_t^l$ is less severely attenuated, implying that gradient vanishing is less likely under input fragmentation.

### D.6. Summary

Synaptic faults induce a shift in the membrane potential $pert_t^l$. This, in turn, increases the corridor-escape probability, $Pr(|\hat{u}_t^l - \vartheta| > \xi)$, which leads to the rise in the frequency of zero surrogate gradient $g_t^l \approx 0$, as shown in Lemma D.1. By applying this to the general closed-form equation for the gradient $\frac{\partial L}{\partial w^l}$, we can realize that the gradient attenuates more strongly with SAF, which shows that gradient vanishing occurs more frequently under SAF. Across layers, this attenuation multiplies Equation (19), making the gradient vanishing more serious as $l$ decreases. The input fragmentation approach can mitigate gradient-vanishing by decreasing the corridor-escape probability, as shown in Lemma D.3.

---

[12]We specialize the analysis to a nonnegative-perturbation regime to isolate the dominant drift effect under the SA1-dominant setting, which is normally observed in the experimental campaign (Chen et al., 2017).

