# OpenReview forum: "Practical Mechanism for Fault-Tolerant Spiking Neural Networks via Simple Input Control Based on Learnable Fragmentation"
_ICML.cc/2026/Conference — ICML 2026 regular_

### Official Review · Reviewer_Zyem · 2026-03-03

**Soundness:** 2
**Presentation:** 3
**Significance:** 2
**Originality:** 2
**Overall Recommendation:** 4
**Confidence:** 3

**Summary:**

Spiking Neural Networks (SNNs), widely regarded as the third generation of neural networks, promise energy-efficient computation for neuromorphic hardware devices. Despite this key advantage, hardware-implemented SNNs are highly vulnerable to circuit-level faults, which can cause catastrophic degradation in model performance.
In this work, the authors state that existing fault-tolerance approaches for SNNs universally require direct access to the network’s internal circuits—either to modify synaptic weights or monitor internal neuronal states—an assumption that severely restricts the real-world deployability of these methods. The authors further identify that improving the hardware robustness of SNNs without such internal access remains a critical, unaddressed challenge in the field.
To tackle this challenge, the authors propose a fault-tolerant mechanism for SNNs that operates exclusively via input data conditioning, with no modifications to the network itself. The authors’ core insight is that hardware faults degrade the usable learning capacity of SNNs, creating a mismatch between the instantaneous input load and the impaired network dynamics. To mitigate this mismatch, the proposed mechanism splits each input sample into multiple fragments, and redistributes the input load via a learnable fragmentation strategy. This strategy jointly optimizes two core components of the fragmentation process: (1) the spatial boundaries of the input fragments, and (2) the number of fragments per input sample.
The authors claim that this work is the first in the field to improve the hardware fault tolerance of SNNs without requiring any access to the network’s internal circuits. They present experimental results demonstrating that their proposed mechanism consistently outperforms prior state-of-the-art methods across multiple SNN architectures, with all performance gains achieved without direct access to the network’s internal circuits. The authors additionally report validation of the mechanism’s effectiveness on SNNs implemented on a physical FPGA platform, which they state confirms the real-world deployability of their approach.

**Compliance With Llm Reviewing Policy:**

Affirmed.

**Key Questions For Authors:**

Question 1: On the adaptive fault tolerance capability and deployment boundary of the FPGA experiments
The introduction and conclusion of the paper imply that the proposed method can achieve fault tolerance via adaptive adjustment after hardware faults occur. However, all experiments in the work follow the paradigm of offline training with pre-injected known fault models, followed by deployment of a fixed strategy after training, with no demonstration of online adaptive adjustment capability after hardware faults emerge. Meanwhile, the full deployment pipeline of the FPGA validation experiments is not clearly specified. We ask the authors to address the following:
Does the proposed method support online adaptive adjustment after hardware faults occur? If yes, please supplement the corresponding implementation details and experimental validation results. If not, please revise the relevant statements about "adaptive fault adjustment" in the paper, and clearly specify the applicable scenario boundaries of the proposed method.
Is the entire training and inference pipeline implemented on-chip in the FPGA experiments? If only the inference stage is deployed on the FPGA, please clarify how the fault model used in the offline training stage is aligned with the real hardware faults on the FPGA, and how the feasibility of the proposed method is guaranteed in real-world commercial neuromorphic chip scenarios.

Question 2: On the rationality of the train-test mismatch in the fragmentation strategy
During training, the paper adopts Gumbel-Softmax to perform weighted optimization over a mixture of multiple fragmentation schemes, while only the single scheme with the maximum weight is selected for inference. This introduces an obvious train-test mismatch, a core concern for machine learning research. We ask the authors to address the following:
What is the core rationale for this "mixture during training, single-path during inference" design? Please supplement the experimental results of using the mixture of fragmentation schemes during inference, and compare the performance difference with the single-path scheme to justify the rationality of the current design.
If the single-path scheme outperforms the mixture scheme in inference, what is the necessity of using mixture-based weighted optimization during training? Please supplement the baseline comparison results of training with a fixed single-path fragmentation scheme only, to demonstrate the incremental benefit of the proposed Gumbel-Softmax mixture training module.

Question 3: On the direct verification of the core mechanism of the proposed method
The core mechanism claimed in the paper is that the proposed method reduces the single-step input load via input fragmentation, which prevents the neuronal membrane potential from deviating from the valid range of the surrogate gradient, and thus fundamentally mitigates the gradient vanishing caused by hardware faults. However, this mechanism is only indirectly validated via the final classification accuracy throughout the paper, with no direct verification results provided to support the validity of the mechanism. We ask the authors to address the following:
Is there any experimental evidence to prove that the proposed method indeed mitigates the gradient vanishing problem during the training of fault-injected SNNs? Please supplement the comparison of gradient norms during training, as well as the visualization of membrane potential distributions, between the proposed method, the fault-free baseline, and other state-of-the-art methods, to directly verify the effectiveness of the claimed core mechanism.

Question 4: On the rigor of the novelty claim of the proposed work
In the introduction, the paper claims that this work is the first in the field to improve the hardware fault tolerance of SNNs without requiring access to the internal circuits of SNNs. However, the ECOC method (Liu et al., 2019), which is included as a baseline in the paper, has already realized blind fault tolerance via output-side processing only, with no need to modify the internal SNN circuits or perform fault detection, fully complying with the core constraint of "no internal access" proposed in this work. In addition, there are a number of mature prior works in the field based on output ensemble and input robustness enhancement that also satisfy the same constraints. We ask the authors to address the following:
What is the core innovative difference between the proposed method and the aforementioned existing "no internal access, no fault detection" fault-tolerant methods?
Why is this work claimed as the first in this direction? Please revise the inappropriate novelty statement in the introduction, or supplement sufficient evidence to support this claim.

**Limitations:**

The authors have not systematically and adequately discussed the technical limitations of this work, nor have they addressed any potential negative societal impact associated with the proposed method. Below are targeted constructive suggestions for improvement:

1. Limitations of Core Assumptions and Applicable Scenarios
The effectiveness of the proposed method is entirely reliant on the ideal laboratory assumption that the fault type, ratio, and distribution are fully aligned between the training and test phases, and that full and accurate prior knowledge of all hardware faults is available in advance. However, in real-world commercial deployment scenarios, commercial off-the-shelf (COTS) neuromorphic chips generally do not open access to internal circuit probing interfaces, making it impossible to obtain precise fault information. Furthermore, after deployment, chips will experience dynamic aging faults as service time and environmental conditions change, meaning the fault distribution can never be perfectly aligned with the settings used during training. The authors should explicitly acknowledge that the proposed method is currently only applicable to laboratory scenarios where faults can be accurately probed in advance and fault states remain fixed, and cannot be adapted to real-world deployment scenarios with mass-produced commercial chips and long-term dynamic faults. Corresponding future optimization directions for this limitation should also be supplemented.

2. Limitations of Practicality for Hardware Deployment
The FPGA validation of the proposed method only completes verification under the ideal scenario of an inference network 固化 (cured) after offline training, without implementing a full on-chip training pipeline or adaptive adjustment for dynamic faults. In addition, the hardware resource utilization and additional overhead of the fragmentation preprocessing logic are not clearly quantified. The authors should explicitly acknowledge that the practicality of the proposed method is limited in resource-constrained edge device deployment scenarios with independent operation requirements. Quantified results of the hardware resource consumption of the fragmentation logic should be supplemented to clarify the real deployment boundary of the solution.

3. Limitations of Deployment Value Boundary in Complex Real-World Scenarios
The proposed learnable fragmentation-based fault-tolerant mechanism has verified its core effectiveness across all tested datasets: it achieves higher classification accuracy than all compared fault-tolerant methods under the same fault ratio, and significantly mitigates the magnitude of performance degradation caused by hardware faults relative to the fault-free vanilla SNN baseline. Critically, the mechanism itself does not introduce additional negative impairment to the underlying feature extraction capability of the SNN model.
However, it should be clearly noted that on high-dimensional, fine-grained complex datasets such as CIFAR-100, the absolute classification accuracy of all tested SNN pipelines (including the proposed method and the fault-free vanilla SNN baseline) is significantly lower than that of their ANN counterparts with the same architecture, and fails to meet the usable performance threshold for most industrial deployment scenarios. This phenomenon stems from the inherent bottleneck of SNN models in feature extraction for complex fine-grained classification tasks, which is a common technical challenge in the field, not a design flaw of the proposed fault-tolerant mechanism.
That said, this inherent bottleneck still restricts the practical deployment value of the proposed mechanism. The core application value of a fault-tolerant mechanism is to guarantee the availability of an SNN model that already meets industrial performance requirements when hardware faults occur. If the underlying SNN model itself is not viable for deployment in complex real-world scenarios, the practical application value of the proposed method remains clearly bounded, even if it perfectly mitigates performance degradation caused by faults. Future work can combine the latest advances in SNN feature extraction for complex scenarios to further expand the applicable scenarios of this fault-tolerant mechanism.

**Strengths And Weaknesses:**

Strengths:

1.Precise problem identification and practical relevance.
The paper accurately pinpoints a core limitation of existing fault-tolerant methods for hardware SNNs: most approaches require modifying internal circuits, weights, or neuronal dynamics, or need detailed knowledge of fault locations and types, making them inapplicable to off-the-shelf, non-reconfigurable neuromorphic chips. This focus on a real-world deployment constraint gives the work strong practical motivation.

2.Well-founded and self-consistent methodology.
Starting from a rigorous analysis of how hardware faults induce gradient vanishing in SNNs, the authors build a complete pipeline consisting of learnable division boundaries, adaptive selection of the number of fragments, energy-balancing regularization, and entropy-weighted multi-step decoding. Each component is clearly motivated and theoretically supported. Techniques such as bounded reparameterization ensure stable training, and the overall design directly addresses the root cause of learning degradation under faults.

3.Comprehensive and statistically sound evaluation.
The experimental validation spans seven datasets across three modalities (image, sensory, audio), three SNN architectures (MLP, VGG‑7, ResNet‑18), and three common fault types (SAF, RWF, CEF). It includes full ablation studies, per‑fragment energy analysis, and FPGA-based inference experiments. All results are reported with 95% confidence intervals, confirming the method’s effectiveness and generalization ability with proper statistical rigor.

Weaknesses:

1.Overclaimed novelty in the introduction.
The paper states that it is “the first mechanism that operates without accessing internal SNN circuits”, which is an exaggeration. The ECOC method (Liu et al., 2019), which the authors themselves list as a baseline, already achieves fault tolerance without any internal circuit modification or fault diagnosis. This overstatement undermines the credibility of the claimed innovation and should be corrected.

2.Insufficient details on FPGA deployment.
A key selling point is deployability on off-the-shelf hardware, yet several critical implementation aspects are missing:

The boundary between training and inference is unclear—specifically, whether the scheme is “offline training + inference-only mapping” to the FPGA, and how the transferred parameters align with the software-trained model.

The physical implementation of the fragmentation logic (e.g., whether it runs on the FPGA or a host processor) and its associated hardware cost are not disclosed, leaving doubts about the feasibility of standalone edge deployment.

The method for injecting faults into the actual FPGA hardware is not described, making it impossible to assess the trustworthiness of the hardware fault experiments.

3.Unvalidated adaptation for sequential data.
For 1D time-series datasets (UCI-HAR, AudioMNIST), the authors reshape the data into 2D matrices to apply spatial fragmentation. No evidence is provided that this reshaping preserves the local temporal correlations essential for such tasks. Consequently, the observed performance gain might partly arise from the altered data representation rather than the fragmentation strategy itself.

4.Key results relegated to the appendix.
The FPGA experiments, core ablation studies, multi‑modal supplementary results, and evaluations under other fault types (RWF, CEF) are all placed in the appendix, while the main text presents only a subset of results. This organization burdens the reader and weakens the self‑containedness and persuasive power of the main paper. The most critical evidence should be moved to the main body.

---

> ### Author Rebuttal · Authors · 2026-03-30
>
> # Thanks to Reviewer Zyem
> We appreciate your comments on the novelty, FPGA implementation, and assumption.
>
> **Link for additional experimental results:** https://anonymous.4open.science/r/ICML-2026-rebuttal-BFC3/README.md
>
> ---
>
> # Responses to Weaknesses, Questions, and Limitations
>
> W1/Q4: ECOC-based approaches need to change the size of the output layer, which is related to codeword length. This is because it increases the error correction ability of codewords to mitigate severe faults by increasing the length of codewords. The approaches based on output ensemble and input robustness enhancement require reconfigurability of output layers or the use of additional data in the input samples [1, 2]. These approaches increase the overhead of neuromorphic devices significantly. Conversely, our mechanism does not require hardware reconfiguration or the insertion of additional data into the input samples. This makes our mechanism more practical for neuromorphic devices.
>
> W2/Q1: “Offline training + mapping the trained model on the FPGA” causes the misalignment of transferred parameters. To reduce mismatch, we validate the deployed design with Vivado, widely used for FPGA synthesis. In this setting, the discrepancy in trained parameters is small (see link). To evaluate in a reduced misalignment setting, we trained simple SNNs using Vivado. We observe slightly higher accuracy than in Appendix B.6 due to the reduced effect of misalignment (see the link).
> For hardware cost and implementation on fragmentation logic, please refer to the response to W-additional costs/Q4 of Review WiQe. We emulate SAFs on an FPGA by fixing the randomly selected synaptic weights to maximum (SA1) and minimum (SA0) values.
>
> “Adaptive adjustment” indicates the method that controls division lines and the number of fragments according to the fault state. To avoid misleading readers, we remove this expression. While online training is not shown in the current prototype, our mechanism can be extended to retrainable platforms such as SpiNNaker2/Loihi. We will revise the paper to state the deployment boundary as future work.
>
> W3: The 1D-to-2D conversion is an order-preserving remapping with zero-padding, rather than a feature transforming. This is because it largely preserves the original signal values and ordering, except for minimal zero padding [3].
>
> W4: We will place the key results, such as FPGA experiments and ablation studies, in Section 5 to improve legibility if our paper is accepted.
>
> Q2: The core rationale of our mixture-based design is low complexity. We apply the single-path approach to inference because it achieves comparable fault tolerance to the mixed approach with less complexity. We show the classification accuracy of the single- and mixed-path approaches in the inference. As shown in the table, the single-path achieves comparable accuracy with the mixed-path without a large overhead. The time and spatial complexity of the mixed-path approach is $\frac{38}{T_{sel}}$ and $\frac{14}{T_{sel}}$ times larger than the single-path approach.
>
> Q3: We show the absolute deviation of the membrane potential from the threshold, the probability that the absolute value of the gradient is below 0.0001 across all neurons, and the L1 norm of the absolute value with 50% fault ratio of SAF in MLP, VGG-7, and ResNet-18. As shown in the tables, our mechanism mitigates the bad effects in membrane potential and gradients more effectively than the benchmarks.
>
> L1/L2: Current prototype assumes that the fault state is aligned between training and test phases. Thus, our method applies to controlled settings where the fault state is known in advance. But if training and testing platforms are the same, ours can be applied to a broader case. As a future direction, we extend our method to a retrainable platform that supports online training in evolving fault conditions.
>
> For resource consumption and additional overhead of preprocessing logic, please refer to responses for W-additional costs/Q4 of Review WiQe. We will revise the paper to state limitations on applicable scenarios/prototypes/practicality on the edge device and clarify future direction.
>
> L3: We also show that our mechanism improves fault tolerance of complex SNNs with ImageNet in the responses to Q1 of Reviewer Ners and W5/Q3 of Reviewer VoyZ. We will revise our paper to discuss the limitations of the inherent limited performance of SNN/ bounded applications to complex scenarios. As a future direction, we will expand applicable scenarios by combining with the latest advances in SNN.
>
> ---
>
> # References
>
> [1] Youngeun Kim et al., "Neural Architecture Search for Spiking Neural Networks", in Proc. of ECCV, 2022.
>
> [2] Jianhao Ding et al., "SNN-RAT: Robustness-enhanced Spiking Neural Network through Regularized Adversarial Training," in Proc. of NeurIPS, 2022.
>
> [3] Junlu Wang et al., "A T-CNN time series classification method based on Gram matrix," Scientific Reports, vol. 12, Art. no. 15731, 2022.

---

### Official Review · Reviewer_VoyZ · 2026-03-10

**Soundness:** 3
**Presentation:** 2
**Significance:** 3
**Originality:** 3
**Overall Recommendation:** 5
**Confidence:** 4

**Summary:**

The paper proposes a learnable image-partitioning-based architecture to minimise risk on faulty hardware.  Interestingly, the design is also evaluated on FPGAs. The method is evaluated across MLP and CNN-based Architectures (VGG, ResNet) on image datasets. Results demonstrate robustness to higher fault rates than those of the previously proposed architectures.

**Compliance With Llm Reviewing Policy:**

Affirmed.

**Final Justification:**

Overall, the authors have conducted sufficient experiments to address my questions. I have read the replies to other reviewers. It seems this paper deserves a 5.

**Key Questions For Authors:**

Q1. Is there a theoretical connection to why fragmentation works for this problem?

Q2. What is the relation between the number of fragments and the number of time steps? Is there a reason they need to be equal?

Q3. Is the proposed method generalizable to large-scale datasets (ImageNet variants) with higher image resolution without compromising performance?

Q4. Is the proposed method efficient (accuracy-wise, memory-wise, energy-wise, wall clock time-wise) on real-world level applications (high-resolution ImageNet-like data)? And will it work for the spiking-speech-command dataset?

Q5. Is your method sensitive to the spike encoding approach?

**Limitations:**

Although the paper comprehensively covers a wide range of models and benchmarks, it lacks overall readability of the proposed architecture due to confusing notations. It becomes difficult for readers to follow when notations are not clearly defined, and the same variables are reused. It seems the proposed architecture can only be implemented for 2D images. Hence, 1D signals are also converted to 2D signals before fragmentation.

**Strengths And Weaknesses:**

**Strengths**
1. The authors are tackling an important problem. The results look promising, as the model has been evaluated across multiple datasets with different fault ratios.
3. The problem has been well motivated from initial experiments.
4. FPGA implementation and real hardware numbers for latency and energy consumption look promising.

**Weakness**
1. **Incosistency with variable assignment**: I can see alot of the variables are being resused in different places in different contexts (for example in eqn 1, $u$ is used for membrane potential and in eqn 3 to parameterize line using $u_k$, similarly $b_k$ is used as a polar cordinate in eqn3 and $b_t(x,y)$ in eqn 5, $\alpha$ as a leak parameter in eqn 1 whereas $\alpha^{(T)}$ for logits in eqn 6) affecting the overall readability of the paper.
2. **Incomplete variable description**: A lot of variables in the equations are not formally defined; for example, parameters a and b of the line are never defined.
3. The model fails to justify architecture choices like the use of Gumbel distribution for training (Eqn. 6).
4. Experiments on TinyImageNet(Appendix B.1.2) showcase performance for all competing models to be very low. Is there no better benchmark for TinyImageNet?
5. Given that the method focuses on input control rather than network (which makes sense), the evaluation should be extended to higher resolution datasets (ImageNet/ImageNet R/R/Imagenet S) to convince the readers.
6. Although Appendix A.1 cites some papers, it's not clear how STOA neuromorphic hardware can efficiently support backpropagation on such large resolution datasets, wherein this method could be really useful. I appreciate the authors for proceeding with the FPGA implementation. However, the end-to-end efficiency (energy, wall clock time, and memory) should also be reported for a high-resolution (224 x 224 x 3) dataset, including performance metrics.

---

> ### Author Rebuttal · Authors · 2026-03-28
>
> # Thanks to Reviewer VoyZ
> We appreciate your comments regarding the legibility of the equations and the generalizability of our mechanism.
>
> **Link for additional experimental results:** https://anonymous.4open.science/r/ICML-2026-rebuttal-21E0/README.md
>
> ---
>
> # Responses to Weaknesses, Questions, and Limitations
>
> W1/W2/L-redability: To address your concern, we checked the reused variables, and we will replace them with distinct symbols. For example, we replace $u_k$ with $h_k$ in Eq.3, $b_t$ with $soft_t$ in Eq 5, and $\alpha^{(T)}$ with $\nu^{(T)}$ in Eq.6. We also added explanations for previously undefined variables. For example, we now define $a_k$, $b_k$ and $c_k$: $a_k$ and $b_k$ decides the orientation of the division line ($-\frac{a_k}{b_k}$) and $c_k$ determines its offset. To improve readability, we summarize the revised notations and the newly defined variables in a notation table, which is available at “notations.pdf” in the link (https://anonymous.4open.science/r/ICML-2026-Notations-23C2/notations.pdf). In the final manuscript, we will change the notations accordingly and add an explanation for undefined variables.
>
> W3: We use the Gumbel distribution in Softmax because it follows categorical sampling, which makes it a proper differentiable approximation of a discrete choice [1]. We have changed the explanation about Gumbel-Softmax in the manuscript for readers to understand why we use Gumbel-Softmax for our mechanism in Subsection 5.1.
>
> W4: Following your suggestion, we conduct the Tiny-ImageNet experiment with 200 training epochs for ResNet-34, and the accuracy improves substantially. We report these updated Tiny-ImageNet results in our response to Q2 of Reviewer Ners. Please refer to that response for the detailed results.
>
> W5/Q3: To address this concern, we evaluate the fault tolerance of our method and the baselines on SEW-ResNet-34 using two high-resolution datasets, ImageNet-1K and ImageNet-R. The table in the link reports the classification accuracy of SAF-injected SEW-ResNet-34 on ImageNet-1K for fault ratios from 0% to 90%. We also measure the accuracy of SAF-injected SEW-ResNet-34 using our mechanism and benchmarks on the ImageNet-R dataset, comprising 200 classes. Our experimental results demonstrate that our mechanism effectively enhances the fault tolerance of SNNs on both high-resolution datasets.
>
> W6/Q4: As suggested, we implement SEW-ResNet-34 integrated with our mechanism on our FPGA and evaluate the peak memory, energy consumption, and wall-clock time. The table in the link demonstrates that the SEW-ResNet-34 with our mechanism efficiently operates on neuromorphic devices. We also measure the classification accuracy of Spikformer integrated with our mechanism and the benchmarks using the Spiking-Speech-Command (SSC) dataset, comprising 35 classes, under SAFs. We observe that our mechanism improves the fault tolerance of Spikformer with the SSC dataset. Please see the table in the link.
>
> Q1: We additionally provide a formal theoretical explanation for why fragmentation helps in this setting. Specifically, we show that input fragmentation reduces the probability of escaping the surrogate-gradient corridor, which supports the observed mitigation of gradient suppression. For a detailed derivation, please refer to Section A.5 of “theoretical_basis.pdf” in the link (https://anonymous.4open.science/r/ICML-2026-Theory-878D/theoretical_basis.pdf). We will include this derivation in Appendix D.5 of the revised manuscript.
>
> Q2: The experimental results in the tables of the link show that the setting with one fragment per time step achieves the highest accuracy. This is because, with one fragment per time step, the neuron receives the input in a small portion instead of one big input, which helps keep its membrane potential in the useful range where learning can still happen. The number of fragments per time step can be manually set, and we measure the classification accuracy of MLP  and VGG-7 by changing the number and fault ratio.
>
> Q5: We apply the Time-To-First Spike (TTFS) encoder to the SAF-injected MLP and VGG-7 with our mechanism and compare the accuracy to the case with the Poisson encoder. As shown in the tables (see the link), the SNN models with our mechanism do not exhibit significant accuracy changes despite the different encoder types.
>
> L-1D conversion: We adopt a 1D-to-2D conversion that preserves the order information of 1D samples with minimal zero-padding [2]. Through this conversion, our mechanism enhances the fault tolerance of SNNs with 1D datasets.
>
> ---
>
> # References
>
> [1] Eric Jang et al., "Categorical Reparameterization with Gumbel-Softmax," in Proc. of ICLR, 2017.
>
> [2] Junlu Wang et al., "A T-CNN time series classification method based on Gram matrix," Scientific Reports, vol. 12, Art. no. 15731, 2022.

---

> > ### Author Rebuttal · Reviewer_VoyZ · 2026-04-03
> >
> > Thanks for addressing my concerns. I have raised the score to 4. Good Luck!

---

> > > ### Author Response · Authors · 2026-04-03
> > >
> > > Thank you for raising the recommendation score! Your comments are really helpful! Thanks again.

---

### Official Review · Reviewer_WiQe · 2026-03-11

**Soundness:** 2
**Presentation:** 3
**Significance:** 3
**Originality:** 3
**Overall Recommendation:** 4
**Confidence:** 4

**Summary:**

This paper studies the robustness of hardware-implemented spiking neural networks (SNNs) under synaptic fault conditions. The paper is motivated by the practical limitation that many existing fault-tolerance methods require access to internal circuits, weight modification, or neuron-state monitoring, which may introduce substantial deployment overhead on real neuromorphic hardware. To address this issue, this paper proposes an input-controlled fault-tolerance mechanism, which uses a learnable input fragmentation strategy to partition each input sample into multiple fragments while jointly learning the division boundaries and the number of fragments. By doing so, it aims to redistribute the input burden and mitigate the mismatch between degraded learning capability and input load caused by hardware faults. Experiments on multiple SNN models, several datasets, and an FPGA platform suggest that the proposed mechanism achieves good accuracy and shows certain potential for hardware deployment.

**Compliance With Llm Reviewing Policy:**

Affirmed.

**Final Justification:**

The paper is generally solid and has some merit from an application and systems perspective. While the methodological side could be further strengthened, I would lean toward a weak accept.

**Key Questions For Authors:**

1. In scenarios with a relatively low fault ratio or strict latency constraints, could the additional time-step expansion introduced by the proposed method lead to non-negligible overhead?
2. The current method adopts linear division boundaries. Have the authors considered nonlinear or content-aware partitioning strategies that might offer stronger expressive power on more complex data?
3. The number of fragments is currently selected from a predefined candidate set. Have the authors considered learning fragment numbers beyond the candidate set, or adopting a more continuous formulation?
4. In the FPGA setting, have the time, energy, and storage costs introduced by the additional control processor been fully accounted for in the evaluation?

**Limitations:**

The fragmentation mechanism may introduce extra spatial and storage overhead, and a more detailed quantification of its effect in real-time tasks would make the evaluation more complete. Also, the applicability boundary of the mechanism in highly latency-sensitive scenarios, such as autonomous driving or medical applications, may also deserve further discussion.

**Strengths And Weaknesses:**

Strengths: This work addresses a practically meaningful problem by improving fault tolerance without accessing the internal circuits of the SNN, which distinguishes it from prior methods based on internal reconfiguration or state monitoring. Starting from the observation that faults may lead to membrane potential shifts and gradient degradation, the paper proposes to redistribute the input burden through fragmentation, and the method is reasonably aligned with this motivation. In addition, the experimental evaluation covers multiple models, multiple datasets, and an FPGA platform, while also reporting metrics such as accuracy, training time, latency, and energy consumption.

Weaknesses: Compared with the relatively comprehensive experimental evaluation, the theoretical analysis in the paper seems to be more of an explanatory support for the motivation than a fully rigorous justification, and there may still be room to strengthen. Also, the contribution of the paper seems to be clear at the application and system-design level for faulty SNNs, while its significance as a more general machine learning methodology could be articulated more explicitly. Although FPGA verification is persuasive, its actual deployment assumptions and related costs are worth further explanation as it still relies on additional control processors to complete input construction.

---

> ### Author Rebuttal · Authors · 2026-03-29
>
> # Thanks to Reviewer WiQe
>
> We appreciate your comments on the theory, the significance for general machine learning, and the deployment costs of our mechanism.
>
> **Link for additional experimental results:** https://anonymous.4open.science/r/ICML-2026-rebuttal-1277/README.md
>
> ---
>
> # Responses to Weaknesses, Questions, and Limitations
>
> W-theory: We theoretically explain the motivation in Appendix D, but as noted, it seems to provide explanatory support rather than a full theoretical derivation. To improve the theoretical motivation, in addition to the theoretical explanation of how faults affect learning capability, we also provide a theoretical explanation for why fragmentation helps to mitigate the adverse effects of faults. For a detailed derivation, please refer to Section A.5 of “theoretical_basis.pdf” in the link (https://anonymous.4open.science/r/ICML-2026-Theory-878D/theoretical_basis.pdf). We will include this derivation in Appendix D.5 of the revised manuscript.
>
> W-general machine learning: From a general machine learning perspective, we suggest our mechanism as a black-box load-balancing approach that redistributes input load across fragments, thereby improving robustness and controllability without requiring internal model access. Due to this advantage, our mechanism makes SNNs achieve high performance regardless of their internal settings, and our experimental results for W3 of Reviewer ciSN support this claim.
>
> W-additional costs in FPGA/Q4: We construct our FPGA-based testbed consisting of FPGA-based core SNN models and a control unit of the F2 instance. The control unit generates fragments according to our mechanism and feeds them to the FPGA-implemented SNNs. In the evaluation, we assume that the device is in constant use without external physical impact. We evaluate the processing time, energy consumption, and peak memory of our mechanism on our control unit. Compared to  SNNs in Appendix C.3 of the paper, the system-level overhead of our mechanism is negligible. Please refer to the table in the link.
>
> Q1: In the scenarios mentioned by the reviewer, the increase in the number of time steps is not significant because the SNN retains sufficient usable learning capacity. Thus, excessive fragmentation provides little benefit, leading the model to favor fewer fragments (time steps). For this reason, the additional temporal overhead caused by our mechanism is minimal under low fault ratios. We show the selected numbers for time steps with SAF-injected MLP (MNIST) and VGG-7 (CIFAR-10) under the low fault ratio. As shown in the table, the number of time steps does not increase, although the fault ratio increases when the fault ratio is low. Please refer to the table in the link.
>
> Q2: We use linear fragmentation for our mechanism since the training overhead of linear fragmentation is smaller than that of nonlinear fragmentation, while it achieves comparable fault tolerance. This advantage makes linear fragmentation suitable for SNNs, which exhibit high energy efficiency. To compare the complex partitioning to the linear fragmentation, we adopt a nonlinear (quadratic function) fragmentation for our mechanism and compare the top accuracy and training epochs to reach the top accuracy with MLP (MNIST) and VGG-7 (CIFAR-10) under SAF injection.  As shown in the tables of the link, models with linear fragmentation achieve comparable fault tolerance to models with nonlinear fragmentation using fewer training epochs.
>
> Q3: As noted by the reviewer, we consider different candidate sets for the number of fragments and measure the accuracy of models with different sets. Please refer to the tables in our comment for W3 of Reviewer ciSN. Continuously adjusting the number requires large epochs to converge on a specific number. Therefore, continuous formulation significantly increases energy consumption, ruining the high energy efficiency of SNNs.
>
> L: As commented, our mechanism increases the time overhead because it requires additional time steps. However, our mechanism does not incur a time overhead large enough to hinder its adoption in real-time applications. We measure the processing time (ms) of SNN models with our mechanism, using the KITTI object detection dataset and Chest X-ray14, which are widely used for autonomous driving and medical image processing. As shown in the table, the processing time is sufficiently short for the applications that you mentioned [2, 3]. Please refer to the table in the link.
>
> # References
>
> [1] Ji Lin et al., "Memory-efficient Patch-based Inference for Tiny Deep Learning," in Proc. of NeurIPS, 2021.
>
> [2] Y. Luo. "Time Constraints and Fault Tolerance in Autonomous Driving Systems," Technical Report No. UCB/EECS-2019-39, EECS Department, University of California, Berkeley, 2019.
>
> [3] M. Yamada et al., "Development of a real-time endoscopic image diagnosis support system using deep learning technology in colonoscopy," Scientific Reports, 9(1):14465, 2019.

---

> > ### Author Rebuttal · Reviewer_WiQe · 2026-04-02
> >
> > The authors have responded to my concerns. Thank you.

---

> > > ### Author Response · Authors · 2026-04-02
> > >
> > > Thank you for responding to our rebuttal! Your comments are really helpful! Thanks again.

---

### Official Review · Reviewer_ciSN · 2026-03-13

**Soundness:** 2
**Presentation:** 3
**Significance:** 2
**Originality:** 2
**Overall Recommendation:** 4
**Confidence:** 2

**Summary:**

This paper studies fault tolerance in hardware-implemented spiking neural networks (SNNs) under permanent synaptic faults, focusing on the practically important setting where one cannot modify internal circuits or synapses directly. The paper argues that such faults reduce the network’s usable learning capacity and create a bottleneck in which membrane potentials are pushed away from the surrogate-gradient corridor, causing gradient vanishing and poor learning. To address this, the authors propose a learnable input-fragmentation mechanism that temporally splits each input into multiple fragments, with trainable division boundaries and a learnable fragment count, so that the effective input load better matches the degraded capacity of the faulty SNN. The method is evaluated across multiple SNN architectures, datasets, and fault settings, with additional hardware-oriented validation.

**Compliance With Llm Reviewing Policy:**

Affirmed.

**Key Questions For Authors:**

Please refer to the weakness.

**Limitations:**

Yes

**Strengths And Weaknesses:**

[Strengths]:
- The paper addresses a meaningful and practically motivated problem: improving fault tolerance in SNN hardware without requiring internal circuit access or weight-level reconfiguration, which is a realistic constraint for non-reconfigurable neuromorphic systems.
- The motivation is stronger than a purely heuristic story: the paper empirically connects faults to larger membrane-potential deviations and much higher near-zero-gradient probability, then shows that fragmentation reduces both and improves accuracy. This gives the method a reasonably interpretable mechanism.
- The empirical evaluation appears fairly broad, covering multiple models, datasets, and fault settings, and the paper also includes hardware-oriented analysis and FPGA experiments, which strengthens the practical relevance of the work.

[Weakness]
- Some of the high-level claims feel slightly overstated, especially around being a particularly “practical” or uniquely motivated solution; these claims would be stronger with a more careful discussion of assumptions about deployment faults and retraining conditions.
-  The paper presents a plausible mechanistic explanation for the proposed method, supported by both analysis and measurements, but I found the claimed “concrete theoretical basis” somewhat overstated. The current evidence mainly shows that synaptic faults perturb membrane potentials, increase the probability of escaping the surrogate-gradient corridor, and correlate with gradient suppression, while fragmentation empirically mitigates these effects. This is useful and strengthens the intuition behind the method, but it remains closer to an empirical and mechanism-based diagnosis than to a predictive theory that clearly specifies when and why fragmentation should work more generally.
- While the results are promising, the paper would benefit from a clearer discussion of generality: for example, sensitivity to surrogate gradient choice, encoding scheme, threshold/reset design, and the selected fragment candidate set is not fully resolved in the main narrative.

---

> ### Author Rebuttal · Authors · 2026-03-29
>
> # Thanks to Reviewer ciSN
> We appreciate your comments on the claim, theory, and the generalizability of our mechanism.
>
> ---
>
> # Responses to Weaknesses
>
> W1: We replace ‘practicality’ and ‘uniquely motivated’ with ‘simply deployable’ and ‘thoroughly motivated’ to avoid overstatement. To provide a discussion on the assumption, we explain the scenarios in which neuromorphic devices with our mechanism operate. After initial training, faults occur in the neuromorphic device during inference, severely damaging its data processing ability. In this situation, retraining the neuromorphic device is essential to overcome the faults [1]. Current mechanisms for fault-tolerant neuromorphic devices based on retraining demand internal circuit accesses, such as a fault map and weight adjustment, which are unavailable in neuromorphic devices [2]. However, our mechanism only controls input flows of data samples and does not require internal accesses. Therefore, our mechanism improves the fault tolerance without intrusive approaches, making it deployable in neuromorphic devices.
>
> W2: We agree that ‘concrete theoretical basis’ was too strong, and we will revise it to ‘supportive theoretical basis’. We additionally provide a formal analysis supporting why fragmentation can mitigate the adverse effects of synaptic faults, thereby supporting the observed mitigation of gradient vanishing. Specifically, we show that input fragmentation reduces the probability of escaping the surrogate-gradient corridor, which supports the observed mitigation of gradient suppression. For a detailed derivation, please refer to Section A.5 of “theoretical_basis.pdf” in the anonymous link; we will include this derivation in Appendix D.5 of the revised manuscript. Due to the 5000-character restriction, we upload the theoretical derivation to an anonymous link (https://anonymous.4open.science/r/ICML-2026-Theory-878D/theoretical_basis.pdf).
>
> W3: To show the generalizability of our mechanism, we measure the classification accuracy of MLP (MNIST) and VGG-7 (CIFAR-10) under SAF injection of 0-90% fault ratio by changing the surrogate gradient, encoding scheme, threshold, and fragment candidate sets.
>
> **Surrogate gradient**
>
> MLP
> ||0|10|20|30|40|50|60|70|80|90|
> |---|---|---|---|---|---|---|---|---|---|---|
> |Arctan|98.15|97.79|97.03|95.89|95.15|93.52|10.75|9.8|9.8|9.8|
> |Sigmoid|98.04|97.62|96.89|95.51|94.63|93.12|9.8|9.8|9.8|9.8|
>
> VGG-7
> ||0|10|20|30|40|50|60|70|80|90|
> |---|---|---|---|---|---|---|---|---|---|---|
> |Arctan|77.09|73.19|70.08|69.36|63.14|51.46|38.48|23.25|10|10|
> |Sigmoid|76.84|72.75|69.86|69.01|62.38|50.29|37.34|21.98|10|10|
>
> **Encoding scheme**
>
> Please refer to our response to Q5 of Reviewer VoyZ to check the results by changing the encoding scheme (https://anonymous.4open.science/r/ICML-2026-rebuttal-21E0/README.md).
>
> **Threshold**
>
> MLP
> ||0|10|20|30|40|50|60|70|80|90|
> |---|---|---|---|---|---|---|---|---|---|---|
> |1.0|98.15|97.79|97.03|95.89|95.15|93.52|10.75|9.8|9.8|9.8|
> |2.0|98.11|97.62|96.59|95.91|95.38|94.14|12.73|10.6|9.8|9.8|
>
> VGG-7
> ||0|10|20|30|40|50|60|70|80|90|
> |---|---|---|---|---|---|---|---|---|---|---|
> |1.0|77.09|73.19|70.08|69.36|63.14|51.46|38.48|23.25|10|10|
> |2.0|76.58|71.93|69.71|68.84|64.37|54.29|41.47|26.38|11.41|10|
>
> **Reset setting**
>
> MLP
> ||0|10|20|30|40|50|60|70|80|90|
> |---|---|---|---|---|---|---|---|---|---|---|
> |Hard reset|98.15|97.79|97.03|95.89|95.15|93.52|10.75|9.8|9.8|9.8|
> |Soft reset|97.62|96.95|96.18|94.73|93.91|92.69|9.8|9.8|9.8|9.8|
>
> VGG-7
> ||0|10|20|30|40|50|60|70|80|90|
> |---|---|---|---|---|---|---|---|---|---|---|
> |Hard reset|77.09|73.19|70.08|69.36|63.14|51.46|38.48|23.25|10|10|
> |Soft reset|75.68|72.02|68.83|68.11|61.87|50.75|36.93|21.67|10|10|
>
> **Fragment candidate sets**
>
> MLP
> ||0|10|20|30|40|50|60|70|80|90|
> |---|---|---|---|---|---|---|---|---|---|---|
> |{2, 4, 8}|98.15|97.79|97.03|95.89|95.15|93.52|10.75|9.8|9.8|9.8|
> |{2, 4, 8, 16}|97.42|97.06|96.89|95.13|94.24|92.68|9.8|9.8|9.8|9.8|
>
> VGG-7
> ||0|10|20|30|40|50|60|70|80|90|
> |---|---|---|---|---|---|---|---|---|---|---|
> |{2, 4, 8}|77.09|73.19|70.08|69.36|63.14|51.46|38.48|23.25|10|10|
> |{2, 4, 8, 16}|77.32|73.95|71.12|69.8|64.28|53.52|40.76|25.91|13.16|10|
>
> As shown in the tables, our mechanism successfully improves the fault tolerance of SNNs regardless of various settings.
>
> ---
>
> # References
>
> [1] Ayesha Siddique et al., "Improving Reliability of Spiking Neural Networks through Fault Aware Threshold Voltage Optimization," in Proc. of DATE, pp. 1-6, 2023.
>
> [2] Giju Jung et al., "Cost- and Dataset-free Stuck-at Fault Mitigation for ReRAM-based Deep Learning Accelerators," in Proc. of DATE, pp. 1733-1738, 2021.

---

> > ### Author Rebuttal · Reviewer_ciSN · 2026-04-03
> >
> > Thanks for addressing my concerns.

---

> > > ### Author Response · Authors · 2026-04-03
> > >
> > > Thank you for responding to our rebuttal! Your comments are really helpful! Thanks again.

---

### Official Review · Reviewer_Ners · 2026-03-19

**Soundness:** 2
**Presentation:** 2
**Significance:** 2
**Originality:** 3
**Overall Recommendation:** 4
**Confidence:** 3

**Summary:**

This paper improves the model’s fault tolerance only from the input level, improving the overall performance by dividing each input sample into multiple fragments and redistributing the input load via a learnable fragmentation strategy. The whole storyline and writing style look interesting to me, but it seems to fit better in venues like iccad or dac. To me, if this paper aims at a machine learning venue, I would prefer to see whether it works on more SoTA attention-based models instead of small and old models like VGG and ResNet running on small datasets.

**Compliance With Llm Reviewing Policy:**

Affirmed.

**Final Justification:**

I recommend weak accept.

**Key Questions For Authors:**

1.This paper uses old and small network architectures like MLP, VGG, and ResNet. Does the proposed method still work with more sota spiking transformer structures?

2.In Section B.1.2, the accuracy on tiny-ImageNet is only 8–10%. Could the authors train the network better and then compare the accuracy? It looks like, no matter whether with or without the proposed method, the network fails to perform correct classification. Further, in the same tables, does the proposed method show better noise robustness because it has a higher starting accuracy compared with other methods? I recommend training the network well and letting all methods have a similar starting accuracy without injecting noise, and then starting the noise-level comparison.

Small questions:
1. Table 1: Is the neuron result here the average over all SNN neurons, or only the statistics of the faulty SNN neurons? Could the authors separately show the deviation results of normal and faulty neurons?

2. Figure 1: The lines are clustered together, so it is quite hard to see clearly. The authors may use a small zoomed-in figure.

**Limitations:**

See questions.

**Strengths And Weaknesses:**

Strengths:
1. This paper focuses on the fault tolerance of hardware-implemented SNNs, which is often overlooked by software papers.

2. Modification of synaptic weights or circumventing faulty synapses is not needed.

3. it provides an FPGA implementation of the SNN.

Weakness:
See questions.

---

> ### Author Rebuttal · Authors · 2026-03-25
>
> # Thanks to Reviewer Ners
> We appreciate your thorough comments on the scalability of our mechanism and the clarity of the figures.
>
> ---
>
> # Responses to Questions
> Q1: Our mechanism works well with the SOTA spiking transformer structure. We apply our mechanism and the benchmark methods to Spikformer and compare their fault tolerance under the same SAF-injection setting [1].
>
> ||Baseline|ECOC|SoftSNN|Routing|Astrocyte|FalVolt|LIFA|Proposed|
> |---|---|---|---|---|---|---|---|---|
> |0|71.92|72.47|72.01|71.96|71.33|72.29|71.37|72.94|
> |10|65.42|68.11|65.38|66.29|63.44|69.81|63.83|70.15|
> |20|60.19|62.74|60.84|64.83|57.01|66.46|58.19|67.56|
> |30|51.34|53.48|51.15|55.29|48.98|56.61|50.17|58.38|
> |40|35.61|37.08|29.48|36.02|33.12|38.13|33.96|40.24|
> |50|20.45|21.33|8.59|21.91|11.29|22.57|12.75|25.55|
> |60|11.6|11.54|0.1|12.35|1.48|13.05|2.36|17.21|
> |70|0.1|0.1|0.1|0.1|0.1|0.1|0.1|11.42|
> |80|0.1|0.1|0.1|0.1|0.1|0.1|0.1|4.64|
> |90|0.1|0.1|0.1|0.1|0.1|0.1|0.1|0.1|
>
> The table above reports the classification accuracy (%) of SAF-injected Spikformer under fault ratios ranging from 0% to 90%. As shown in the table, Spikformer equipped with our mechanism achieves the highest classification accuracy for most fault ratios under SAF injection. This result shows that our mechanism exhibits stronger fault robustness than the benchmark methods with a transformer-based SNN model and a complex dataset.
>
> Q2: As suggested, to train the network well, we have increased the number of epochs to 200 when training the model with Tiny-ImageNet. To make all methods have a similar starting accuracy, we do not use the time step adjustment of our mechanism, which improves the classification accuracy of models in a clean scenario (0% fault ratio), and compare the classification accuracy of the model using our mechanism with the benchmarks
>
> ||Baseline|ECOC|SoftSNN|Routing|Astrocyte|FalVolt|LIFA|Proposed|
> |---|---|---|---|---|---|---|---|---|
> |0|50.17|50.96|49.14|50.25|50.14|50.22|50.19|50.24|
> |10|42.35|43.27|42.06|43.68|38.52|44.27|40.02|45.89|
> |20|19.77|23.65|20.35|24.52|18.14|25.36|18.89|27.41|
> |30|5.58|8.16|6.74|8.94|6.02|10.15|6.38|16.4|
> |40|0.5|0.5|0.5|0.5|0.5|0.5|0.5|7.79|
> |50|0.5|0.5|0.5|0.5|0.5|0.5|0.5|3.32|
> |60-90|0.5|0.5|0.5|0.5|0.5|0.5|0.5|0.5|
>
> The table above shows the classification accuracy of ResNet-34 with the Tiny-ImageNet dataset using 200 epochs for training. The overall classification accuracy improves since we use sufficient training epochs for proper classification. The model with our mechanism achieves smaller accuracy drops than the benchmarks with the same fault ratio. This result indicates that, with  Tiny-ImageNet, our method does not have a starting accuracy advantage and improves the fault tolerance of SNNs without this advantage.
>
> Small Q1: We obtain the results by averaging over all neurons in nominal and faulty SNNs. To address your question, we additionally compute the absolute deviation separately for nominal neurons and faulty neurons within the faulty SNNs.
>
> ||MLP|VGG-7|ResNet-18|
> |---|---|---|---|
> |Nominal neurons|12.16±2.32|10.96±2.48|1.54±0.99|
> |Faulty neurons|148.28±14.81|49.72±10.35|4.39±1.56|
>
> As shown in the table, we observe that the membrane potential of nominal neurons in the faulty SNNs also increases, resulting in the escape of the potential from the surrogate corridor. This suggests that synaptic faults perturb network-wide membrane dynamics, not only the directly faulty neurons
>
> Small Q2: To improve readability, we add a zoomed-in subfigure that separates the overlapping curves. We will insert a small subfigure showing the lines near 0 in Figure 1 of Subsection 4.2 for the manuscript if the paper is accepted. Please refer to the figure in the link (https://anonymous.4open.science/r/ICML-2026-rebuttal-28EC/atan_membrane.png).
>
> ---
>
> # References
>
> [1] Zhou et al., "Spikformer: When Spiking Neural Network Meets Transformer," in Proc. of ICLR, 2023.

---

> > ### Author Rebuttal · Reviewer_Ners · 2026-04-01
> >
> > My concerns have been solved. I raise my score to 4. Good luck!

---

> > > ### Author Response · Authors · 2026-04-01
> > >
> > > Thank you for raising the recommendation score! Your comments are really helpful! Thanks again.

---

### Decision · Program_Chairs · 2026-04-30

**Decision:**

Accept (regular)

**Comment:**

This paper focuses on improving the hardware-level fault tolerance of spiking neural networks (SNNs) without requiring access to the internal SNN circuits or relying on hardware reconfigurability. It introduces a learnable input fragmentation mechanism that divides each input sample into multiple segments to better match degraded network capacity under hardware faults. The method jointly learns how to partition inputs and how many fragments to use by parameterizing these components, which helps in effectively redistributing the computational load. By operating purely at the input level, it avoids intrusive hardware modifications while maintaining performance. The paper demonstrates improved robustness on standard SNNs as well as physical FPGA platform implementations.

The paper went through productive discussions during the rebuttal phase, and the reviewers unanimously acknowledged the value of the paper's contribution. The central concerns were primarily empirical, such as the lack of benchmarking on larger-scale architectures. These were addressed with additional experiments on Spikformer and SEW-ResNet-34, as well as ImageNet-scale evaluations, and analyses with varying surrogate gradients and neuron encoding schemes, all of which the authors committed to incorporating with discussions in the revised manuscript. Another concern clarified during the discussions was notational inconsistencies, which the authors similarly agreed to revise for the camera-ready version.

The AC believes that there are some stylistic presentation issues in the submitted manuscript, including floating figures and tables between paragraphs or right after subsection titles, equations/figures/tables extending outside single-column margins, and scattered typos, all of which the authors should carefully address before submitting the final camera-ready version (please check the formatting guidelines). The AC also strongly recommends including some condensed version of the key results from Appendix B (the ones with FPGA implementations), in the main manuscript.

In general, the rebuttal was effective and the authors successfully demonstrated the novelty and rigor of their work. Overall, the paper presents an interesting contribution to the field of SNN robustness, and the AC recommends acceptance, with the expectation that the authors carefully address the presentation issues and incorporate the additional experiments and discussions from the rebuttal into the camera-ready version.